# Learning with Operator-valued Kernels in Reproducing Kernel Krein Spaces

**Akash Saha**
IEOR, IIT Bombay
Mumbai, India
akashsaha@iitb.ac.in

**P. Balamurugan**
IEOR, IIT Bombay
Mumbai, India
balamurugan.palaniappan@iitb.ac.in

## Abstract

Operator-valued kernels have shown promise in supervised learning problems with functional inputs and functional outputs. The crucial (and possibly restrictive) assumption of positive definiteness of operator-valued kernels has been instrumental in developing efficient algorithms. In this work, we consider operator-valued kernels which might not be necessarily positive definite. To tackle the indefiniteness of operator-valued kernels, we harness the machinery of Reproducing Kernel Krein Spaces (RKKS) of function-valued functions. A representer theorem is illustrated which yields a suitable loss stabilization problem for supervised learning with function-valued inputs and outputs. Analysis of generalization properties of the proposed framework is given. An iterative Operator based Minimum Residual (OpMINRES) algorithm is proposed for solving the loss stabilization problem. Experiments with indefinite operator-valued kernels on synthetic and real data sets demonstrate the utility of the proposed approach.

## 1 Introduction

We consider the problem of learning a function-valued function $F : \mathcal{X} \to \mathcal{Y}$ between an input space $\mathcal{X}$ and an output space $\mathcal{Y}$ of functions. Sometimes this problem is called *functional regression* (Morris, 2015). Several applications (*e.g.* audio-visual apps, weather forecasting) motivate the need for considering data as functions. Though practical data is typically discrete, the need to consider inherent time-based correlations and its potential smoothness might be fruitful (Ramsay and Silverman, 2007; Kokoszka and Reimherr, 2018). Among the machine learning methods to solve the functional regression problem, we are interested in the functional reproducing kernel Hilbert space (functional RKHS) idea introduced in (Lian, 2007) and substantially developed in (Kadri et al., 2016). Functional RKHS extends the RKHS framework popularly used for multivariate data (Schölkopf et al., 1999) to functional data. Similar to RKHS which is associated with a non-negative (or positive) scalar-valued kernel with the so-called reproducing property, a representer theorem for functional RKHS allows it to be associated with a corresponding non-negative (or positive definite) operator-valued kernel with reproducing property (see (Lian, 2007) and Appendix A). However construction of non-negative or positive definite operator-valued kernels is not straightforward and particular examples with separable structure are provided in (Lian, 2007; Kadri et al., 2016). The positive definiteness of operator-valued kernels is crucial for establishing technical results associated with functional RKHS and also helps in designing efficient algorithms (Lian, 2007; Kadri et al., 2016).

Note that demonstrating the positive definiteness property of operator-valued kernels (even for particular cases) might be a difficult exercise in itself. Demanding the non-negativeness or positive definiteness of operator-valued kernels effectively restricts practitioners from trying other useful operator-valued kernels which might be indefinite, yet potentially useful for some applications (*e.g.* similarity computation between function-valued data can involve indefinite operator-valued kernels).

Similar concerns previously raised in the case of scalar-valued kernels (*e.g.* see (Ong et al., 2004)), have led to interesting theory establishing a counterpart of RKHS, namely the reproducing kernel Krein space (RKKS) suitable for non-positive kernels of certain type (Ong et al., 2004; Oglic and Gärtner, 2018). Here, we embark on a similar pursuit to develop the necessary theoretical tools which would help construct a function-valued RKKS for generalized operator-valued kernels which might not be non-negative. The structure of generalized operator-valued kernels may seem as an extension of generalized scalar-valued kernels considered in (Ong et al., 2004), however dealing with operator-valued nature of the kernels brings in challenges. Designing a suitable algorithmic scheme to make the framework of generalized operator-valued kernels useful for practical applications is also challenging. Therefore, a systematic development and study of generalized operator-valued kernels and related algorithms become imperative. We aim to address these objectives in this work and outline our major contributions below.

**Contributions:** We introduce the concepts of generalized operator-valued kernel (which might be indefinite) and function-valued RKKS. We show the relevant properties required to associate function-valued RKKS with generalized operator-valued kernels. We remark that demonstrating the existence of an associated RKKS for a generalized operator-valued kernel (more specifically, deriving Lemma 2.3 and Corollary 2.3.1 leading to the proof of Theorem 2.4 below) is mathematically challenging. We then cast the functional regression problem over function-valued RKKS in an appropriate learning setup using a regularized empirical loss stabilization formulation. We further prove a representer theorem for the function-valued RKKS which yields a tractable solution of the loss stabilization problem. To make the theoretical framework useful for practical scenarios, we devise an iterative Krylov subspace method called **Op**erator **MIN**imum **RES**idual method (**OpMINRES**) to solve the loss stabilization problem. Further, using an appropriate Rademacher average, we provide technical results on generalization properties of the proposed learning setup. To the best of our knowledge, the technical results connecting the framework of generalized operator-valued kernel and its associated function-valued RKKS, and the proposed OpMINRES algorithmic scheme are new. An extensive empirical evaluation on real data and comparison with benchmark methods demonstrate that the proposed learning framework is competitive, while allowing for the flexibility of using indefinite operator-valued kernels in functional data settings.

**Paper organization:** Generalized operator-valued kernels and function-valued RKKS are introduced and their properties are discussed in Section 2. We formulate a regularized loss stabilization learning problem and furnish a representer theorem for function-valued RKKS in Section 3. The iterative OpMINRES algorithm used to solve the loss stabilization problem is illustrated in Section 4. Bounds on the generalization error are established in Section 5. Related work is summarized briefly in Section 6. Experiments using the OpMINRES algorithm and comparative results are provided in Section 7. Section 8 concludes the paper.

## 2  Generalized Operator-valued Kernels and Function-valued Reproducing Kernel Krein Spaces

To appreciate the results introduced in this Section, it would be useful to recall the fundamentals of Krein spaces, Reproducing Kernel Krein Spaces (RKKS) and generalized scalar-valued kernels. We discuss them in Appendix B, where a scalar-valued RKKS with its associated generalized reproducing kernel is shown to help in learning real-valued functions of the form $f : X \to \mathbb{R}$, $X$ being an appropriate input space. Here, we consider their extensions to learn functions of the form $F : \mathcal{X} \to \mathcal{Y}$, where $\mathcal{X}$ is a suitable input space and $\mathcal{Y}$ is an output space of functions. A relevant framework of operator-valued kernels (Kadri et al., 2016) has been particularly useful in this context. We note that operator-valued kernels have been proposed for infinite dimensional spaces in other previous works (see *e.g.* (Caponnetto et al., 2008; Carmeli et al., 2010)) and also for finite dimensional spaces (Micchelli and Pontil, 2005). We make the following assumption on $\mathcal{X}$ and $\mathcal{Y}$, which would help us to avoid difficulties arising due to functional analysis considerations.

**Assumption 2.1.** $\mathcal{X}, \mathcal{Y}$ are Hilbert spaces of square integrable functions defined on compact sets.

For a compact $\Omega \subset \mathbb{R}$, it is well-known that $\mathcal{X} = \mathcal{Y} = L^2(\Omega)$, the space of equivalence classes of square integrable functions on $\Omega$ satisfy Assumption 2.1. To define an operator-valued kernel, we require the set $\mathcal{L}(\mathcal{Y})$ of bounded linear operators over $\mathcal{Y}$ of the form $\mathfrak{f} : \mathcal{Y} \to \mathcal{Y}$ (for discussion on linear operators and their properties, see *e.g.* (Kreyszig, 1989, Chapter 2)). Recall that in Appendix B.2, scalar-valued kernels $k : X \times X \to \mathbb{R}$ mapped a pair $(x, x') \in X \times X$ to $k(x, x') \in \mathbb{R}$. This

notion can be extended to the functional setting enabling us to devise an operator-valued kernel to map the elements of $\mathcal{X} \times \mathcal{X}$ to $\mathcal{L}(\mathcal{Y})$, as follows.

**Definition 2.1. Operator-valued Kernel.** (Kadri et al., 2016) An $\mathcal{L}(\mathcal{Y})$-valued kernel $K$ on $\mathcal{X}^2$ is a function $K(\cdot, \cdot) : \mathcal{X} \times \mathcal{X} \to \mathcal{L}(\mathcal{Y})$, with the following properties:

1. $K$ is Hermitian if $\forall w, z \in \mathcal{X}, K(w, z) = K(z, w)^*$, (where $^*$ denotes the adjoint operator),
2. $K$ is non-negative (or positive definite) on $\mathcal{X}^2$ if it is Hermitian and for every natural number $r$ and all $\{(w_i, u_i)_{i=1,2,...,r}\} \in \mathcal{X} \times \mathcal{Y}$, the matrix with $(i, j)$-th entry given by $\langle K(w_i, w_j)u_i, u_j \rangle_{\mathcal{Y}}$ is non-negative (or positive definite).

For an operator-valued kernel $K$, and for a set $\{z_i\}_{i=1}^n \subset \mathcal{X}$, we can define a corresponding matrix $\mathbf{K} \in \mathcal{L}(\mathcal{Y}^n)$ called the block operator kernel matrix whose entries are $\mathbf{K}_{ij} = K(z_i, z_j) \in \mathcal{L}(\mathcal{Y})$. Then the trace $\text{Tr}(K(z_i, z_j))$ of operator $K(z_i, z_j)$ can be defined as the trace $\text{Tr}(\mathbf{K}_{ij})$ of the corresponding matrix $\mathbf{K}_{ij}$. Note that verifying the Hermitian and non-negativity properties in Definition 2.1 is not straightforward and we need to consider specific forms which would satisfy both these properties (Kadri et al., 2016). We now discuss a construction from (Kadri et al., 2016) which would help us appreciate the structure of an operator-valued kernel. Suitable extensions of this example will be considered later when we discuss the generalized operator-valued kernel case. Note also that a similar construction is available in (Lian, 2007) and is used in other settings as well (Micchelli and Pontil, 2005; Caponnetto et al., 2008; Alvarez et al., 2012). Consider now the following operator-valued kernel with a separable structure (Kadri et al., 2016):

$$K(x_i, x_j) = k(x_i, x_j)T, \tag{1}$$

where $x_i, x_j \in \mathcal{X}, T$ is a bounded linear operator on $\mathcal{Y}$ and $k$ is a positive scalar-valued kernel on $\mathcal{X}^2$. Notice that the operator-valued kernel $K(\cdot, \cdot)$ construction in Eq. (1) assumes a positive scalar-valued kernel $k : \mathcal{X} \times \mathcal{X} \to \mathbb{R}$ which is then used to scale an operator $T \in \mathcal{L}(\mathcal{Y})$. A concrete example for $K$ of the form in Eq. (1) can be given as:

$$(K(x_i, x_j)y)(t) = k(x_i, x_j) \int_{\Omega_y} h(s, t)y(s)ds, \tag{2}$$

where, $\Omega_x = \Omega_y = [0, 1], \mathcal{X} = L^2(\Omega_x), \mathcal{Y} = L^2(\Omega_y), k$ is a positive scalar-valued kernel on $\mathcal{X}^2$ and $h : \Omega_y \times \Omega_y \to \mathbb{R}$ is a kernel on $(\Omega_y)^2$. The linear integral operator used in Eq. (2) is especially useful in applications involving data that can be well-approximated using continuous functions (Ramsay and Silverman, 2007). The form of $K$ considered in Eq. (2) is called a Hilbert-Schmidt integral operator and is known to be non-negative (Kadri et al., 2016).

Significant impetus has been given in the literature to construct *non-negative* operator-valued kernels which can be associated with a suitable functional RKHS (Lian, 2007; Carmeli et al., 2010; Kadri et al., 2016). For an operator-valued kernel to be qualified as a Mercer kernel, Carmeli et al. (2010) provide a characterization that the associated RKHS (whose elements are continuous functions) be a subspace of the vector space $\mathcal{C}(\mathcal{X}, \mathcal{Y})$ of continuous functions from $\mathcal{X}$ to $\mathcal{Y}$. Moreover operator-valued kernels which are Mercer, are locally bounded and strongly continuous (Carmeli et al., 2006, 2010). Henceforth we will restrict our attention to only those function-valued RKHS whose associated operator-valued kernel can be qualified as Mercer in the sense of Carmeli et al. (2010). Analogous to the bijection between scalar-valued RKHS and Mercer kernels, there exists a bijection between the space of operator-valued kernels and the space of function-valued RKHS (Kadri et al., 2016).

We now move on to accomplish one of the major goals of our current work here, which is to develop suitable generalized operator-valued kernels (that might not be non-negative), which can then be appropriately associated with a function-valued RKKS.

**Definition 2.2. Generalized Operator-valued Kernel**: A generalized $\mathcal{L}(\mathcal{Y})$-valued kernel $\breve{K}$ on $\mathcal{X}^2$ is a function $\breve{K}(\cdot, \cdot) : \mathcal{X} \times \mathcal{X} \to \mathcal{L}(\mathcal{Y})$ which can be written as $\breve{K} = K_1 - K_2$, where $K_1, K_2 : \mathcal{X} \times \mathcal{X} \to \mathcal{L}(\mathcal{Y})$ are non-negative operator-valued kernels.

Similar to the non-negative operator-valued kernel case, it is possible to define a block operator kernel matrix $\breve{\mathbf{K}}$ for a generalized operator-valued kernel. The definition of a generalized operator-valued kernel is motivated by the generalized scalar-valued kernel $\breve{k}$ in Theorem B.1 (see Appendix B), where $\breve{k}$ is represented as a difference of two positive scalar-valued kernels $k_1$ and $k_2$. The next immediate goal is to establish a connection between the generalized operator-valued kernel and an

appropriate RKKS, analogous to the result in Theorem B.2, where a generalized scalar-valued kernel $\breve{k}$ is associated with a scalar-valued RKKS. The following definition will help us to define the required RKKS.

**Definition 2.3. Function-valued RKKS**: A Krein space $\mathcal{F}$ of functions from $\mathcal{X}$ to $\mathcal{Y}$ is called a reproducing kernel Krein space if there is a $\mathcal{L}(\mathcal{Y})$-valued kernel $\breve{K}$ on $\mathcal{X}^2$, such that:

1. the function $z \mapsto \breve{K}(w, z)g$ belongs to $\mathcal{F}, \forall z, w, \in \mathcal{X}$ and $\forall g \in \mathcal{Y}$,
2. **(reproducing property)** $\langle F, \breve{K}(w, .)g \rangle_{\mathcal{F}} = \langle F(w), g \rangle_{\mathcal{Y}}$, for every $F \in \mathcal{F}, w \in \mathcal{X}, g \in \mathcal{Y}$.

Note that we have defined a function-valued RKKS by extending the definition provided for function-valued RKHS in (Kadri et al., 2016). To establish a correspondence between generalized operator-valued kernel and a function-valued RKKS, the following results are essential. Lemma 2.2 provides a RKHS characterization of the intersection of the function-valued RKHS associated with two non-negative operator-valued kernels on $\mathcal{X}^2$. Lemma 2.3 helps to construct partially ordered set $I(K_1, K_2)$ which is also inductive (see (Bourbaki, 2004, Chapter III) for a definition of inductive set).

**Lemma 2.2.** Let $K_1$ and $K_2$ be two $\mathcal{L}(\mathcal{Y})$-valued non-negative kernels on $\mathcal{X}^2$ with corresponding function-valued RKHS $\mathcal{H}_1$ and $\mathcal{H}_2$ respectively. Then the intersection $\mathcal{H}_1 \cap \mathcal{H}_2$ with the inner product

$$\langle f, f \rangle_{\mathcal{H}_1 \cap \mathcal{H}_2} = \langle f, f \rangle_{\mathcal{H}_1} + \langle f, f \rangle_{\mathcal{H}_2}$$

is a RKHS contractively included in $\mathcal{H}_1$ and $\mathcal{H}_2$.

Note that for two $\mathcal{L}(\mathcal{Y})$-valued non-negative kernels $K_1, K_2$, we let $K_1 \leq K_2$ if $\langle K_1(x, x)y, y \rangle_{\mathcal{Y}} \leq \langle K_2(x, x)y, y \rangle_{\mathcal{Y}}, \forall x \in \mathcal{X}, \forall y \in \mathcal{Y}$. This notation is used in the next Lemma.

**Lemma 2.3.** Let $K_1$ and $K_2$ be two $\mathcal{L}(\mathcal{Y})$-valued non-negative kernels on $\mathcal{X}^2$ and let $I(K_1, K_2)$ denote the set of all functions $K$ non-negative on $\mathcal{X}^2$ and such that $K \leq K_1$ and $K \leq K_2$. Then $I(K_1, K_2)$ is inductive.

We now have the following corollary.

**Corollary 2.3.1.** Let $K$ be a difference of two non-negative $\mathcal{L}(\mathcal{Y})$-valued kernels on $\mathcal{X}^2$, $K = K_1 - K_2$. Then, without loss of generality, one can choose $K_1$ and $K_2$ with corresponding reproducing kernel Hilbert spaces $\mathcal{H}_1$ and $\mathcal{H}_2$, respectively, such that $\mathcal{H}_1 \cap \mathcal{H}_2 = \{0\}$.

The results stated in Lemma 2.2, Lemma 2.3 and Corollary 2.3.1 are extensions of similar results proved in (Alpay, 1991) for the set $\mathbb{C}^{m \times m}$ of all $m \times m$ matrices over field $\mathbb{C}$ of complex numbers. We give their proofs in Appendix C. Corollary 2.3.1 especially helps in the construction of an appropriate function valued RKKS for a generalized operator-valued kernel in the following result.

**Theorem 2.4.** Let $\breve{K}$ be a $\mathcal{L}(\mathcal{Y})$-valued kernel on $\mathcal{X}^2$. Then there is an associated reproducing kernel Krein space if and only if $\breve{K}$ is a generalized $\mathcal{L}(\mathcal{Y})$-valued kernel, that is, $\breve{K} = K_1 - K_2$, where $K_1$ and $K_2$ are non-negative $\mathcal{L}(\mathcal{Y})$-valued kernels on $\mathcal{X}^2$.

The proof of Theorem 2.4 follows the arguments in (Alpay, 1991, Theorem 2.1); details are given in Appendix C.

**An example for generalized operator-valued kernel:** Having established the correspondence between a generalized operator-valued kernel and function-valued RKKS, we consider an extension of $K$ in Eq. (1) as

$$\breve{K}(x_i, x_j) = (g(x_i, x_j))(T_1 - T_2) \text{ or } \breve{K}(x_i, x_j) = (g_1(x_i, x_j) - g_2(x_i, x_j))T,$$

where $x_i, x_j \in \mathcal{X}, T, T_1, T_2$ are bounded linear operators on $\mathcal{Y}$ and $g, g_1, g_2$ are positive scalar-valued kernels on $\mathcal{X}^2$. As a concrete example, consider a generalized operator-valued kernel analogous to the one in Equation (2) as

$$(\breve{K}(x_i, x_j)y)(t) = g(x_i, x_j) \int_{\Omega_y} h(s, t)y(s)ds, \tag{3}$$

where, $\Omega_x = \Omega_y = [0, 1], \mathcal{X} = L^2(\Omega_x), \mathcal{Y} = L^2(\Omega_y), g$ is a scalar-valued kernel on $\mathcal{X}^2$ and $h$ is an output kernel on $(\Omega_y)^2$, and either $g$ or $h$ is indefinite. We illustrate in Appendix D that the operator-valued kernel constructed in Eq. (3) satisfies the properties in Definition (2.3).

We now move on to define a suitable learning problem involving generalized operator-valued kernels and function-valued RKKS.

# 3 Learning Problem Formulation

Let $\mathcal{X} = L^2([a,b]), a < b$ and $\mathcal{Y} = L^2([c,d]), c < d$, thus satisfying Assumption 2.1. Consider the supervised setting of learning a function $F$, such that $F(x_i) = y_i$, for the training data $((x_i(s), y_i(t)))_{i=1}^n \in (\mathcal{X} \times \mathcal{Y})^n$, where $s \in [a,b]$, $t \in [c,d]$. We consider a Krein space $\mathcal{K}$ of operators from $\mathcal{X}$ to $\mathcal{Y}$. Inspired by Ong et al. (2004), we now formulate the learning problem as a regularized empirical loss stabilization problem over the functions in $\mathcal{K}$, as follows.

$$\widetilde{F}_\lambda = \arg\operatorname*{stabilize}_{F \in \mathcal{K}} \sum_{i=1}^n \|y_i - F(x_i)\|_{\mathcal{Y}}^2 + \lambda\langle F, F\rangle_{\mathcal{K}}, \tag{4}$$

where $\lambda > 0$ is the regularization parameter. Note that problem (4) considers risk stabilization (to find a stationary point) instead of the usual risk minimization, as the regularization term $\langle F, F\rangle_{\mathcal{K}}$ can be negative, which makes the problem non-convex. We now furnish a representer theorem which provides a representation of the solution of problem (4) using the generalized operator-valued kernel $\breve{K}$ associated with the Krein space $\mathcal{K}$.

**Theorem 3.1** (**Representer theorem**). Let $\breve{K}$ be a generalized operator-valued kernel and $\mathcal{K}(= \mathcal{K}_1 \oplus \mathcal{K}_2 = \{F_1 + F_2 | F_1 \in \mathcal{K}_1, F_2 \in \mathcal{K}_2\})$ its corresponding function-valued RKKS. The solution $\widetilde{F}_\lambda \in \mathcal{K}$ of the regularized stabilization problem $\Theta(F) = \operatorname{stabilize}_{F \in \mathcal{K}} \sum_{i=1}^n \|y_i - F(x_i)\|_{\mathcal{Y}}^2 + \lambda\langle F, F\rangle_{\mathcal{K}}$, where $\lambda > 0, F(= F_1 + F_2) \in \mathcal{K}$, has the following form: $\widetilde{F}_\lambda(.) = \sum_{i=1}^n \breve{K}(x_i, .)u_i$, where $u_i \in \mathcal{Y}$.

Theorem 3.1 can be proved by finding the Gateaux derivative of the optimization function $\Theta(F)$ and equating it to zero. Proof details are given in Appendix E. Using Theorem 3.1, we can cast optimization problem (4) over $\mathcal{Y}^n$ as

$$\tilde{\mathbf{u}}_\lambda = \arg\operatorname*{stabilize}_{\mathbf{u} \in \mathcal{Y}^n} \sum_{i=1}^n \left\|y_i - \sum_{j=1}^n \breve{K}(x_i, x_j)u_j\right\|_{\mathcal{Y}}^2 + \lambda\left\langle \sum_{i=1}^n \breve{K}(x_i, .)u_i, \sum_{j=1}^n \breve{K}(x_j, .)u_j\right\rangle_{\mathcal{K}},$$

which can be simplified to the following equivalent problem using the reproducing property of $\breve{K}$:

$$\tilde{\mathbf{u}}_\lambda = \arg\operatorname*{stabilize}_{\mathbf{u} \in \mathcal{Y}^n} \sum_{i=1}^n \left\|y_i - \sum_{j=1}^n \breve{K}(x_i, x_j)u_j\right\|_{\mathcal{Y}}^2 + \lambda \sum_{i=1, j=1}^n \langle \breve{K}(x_i, x_j)u_i, u_j\rangle_{\mathcal{Y}}. \tag{5}$$

The optimization problem (5) needs to be solved in order to determine the vectors $u_i$, $i = 1, \ldots, n$, to learn the function-valued function $F$ using Theorem 3.1. By using the conditions for finding stationary points of problem (5) (see Appendix F), we obtain

$$(\breve{\mathbf{K}} + \lambda I)\mathbf{u} = \mathbf{y}, \tag{6}$$

where $\breve{\mathbf{K}}$ is a block operator kernel matrix, $\mathbf{y}$ is a column vector of output functions corresponding to inputs $x_i$'s, $i = 1, \ldots, n$. The $\mathbf{u}$ computed from Equation (6) consists of a column vector of operators $u_i \in \mathcal{L}(\mathcal{Y})$ using which the prediction for an unseen example $\hat{x}$ is obtained as: $F(\hat{x}) = \sum_{i=1}^n \breve{K}(x_i, \hat{x})u_i$. The final operator matrix relation in Eq. (6) closely resembles the one obtained in (Kadri et al., 2016); however a simple inversion of $(\breve{\mathbf{K}} + \lambda I)$ might no longer possible in Eq. (6). To tackle this difficulty, we propose in the next section, an iterative algorithm which can be used to solve Eq. (6).

# 4 Operator Minimum Residual (OpMINRES) Algorithm to solve (6)

To solve for $\mathbf{u}$ in problem (6), we follow Ong et al. (2004) and adapt the minimum residual (MINRES) algorithm used for solving a system of linear equations (Paige and Saunders, 1975). MINRES is a Krylov subspace method (see *e.g.* (Barrett et al., 1994; Choi, 2006) and Appendix G) and is well-suited for problems of the type given in Eq. (6), since the matrix of operators $(\breve{\mathbf{K}} + \lambda I)$ is Hermitian (or symmetric), may be indefinite, and more importantly, MINRES would help us in approximating the problem in an infinite dimensional setting to a problem in $\mathbb{R}^k$ for some suitable $k \geq 1$ (as described below). We call the adapted version Operator minimum residual (OpMINRES).

The norm used in conventional MINRES is the usual vector $\ell_2$-norm (Paige and Saunders, 1975), however for OpMINRES we need to consider the norm of a vector of functions. Let $\mathbf{v} = [\mathbf{v}_1, \mathbf{v}_2, \ldots, \mathbf{v}_n]^\top \in \mathcal{Y}^n$, where $\mathcal{Y} = L^2([0,1])$ is assumed for simplicity; note however that the norm can be suitably modified for any $\mathcal{Y}$ satisfying Assumption 2.1. One possible definition of norm of $\mathbf{v}$ is given by $\|\mathbf{v}\|_{\mathcal{Y}^n} = \sqrt{\sum_{i=1}^n \int_0^1 \mathbf{v}_i^2(t)dt}$. Now letting $\mathbf{A} := (\check{\mathbf{K}} + \lambda I)$ in Eq. (6), we have the equivalent form $\mathbf{Au} = \mathbf{y}$ and we see that $\mathbf{A}$ is a symmetric $n \times n$ matrix of self-adjoint linear bounded operators on $\mathcal{Y}(= L^2([0,1]))$ and $\mathbf{u}, \mathbf{y} \in \mathcal{Y}^n$. To solve $\mathbf{Au} = \mathbf{y}$, OpMINRES minimizes the norm $\|\mathbf{y} - \mathbf{Au}\|_{\mathcal{Y}^n}$, and at each iteration OpMINRES is composed of the following major steps:

1. A scheme for transforming the linear operator system into a linear system in $\mathbb{R}^k$ using a Lanczos-based method (Lanczos, 1950), which we call **OpLanczos**.
2. Using QR decomposition to solve the linear system obtained in the previous step.
3. A transformation to obtain back the solution in $\mathcal{Y}^n$.

We provide a summary of these steps here, relegating all details to Appendix H. OpMINRES attempts to find a solution in the Krylov subspace obtained at the $k$-th iteration, denoted by $\mathcal{K}_k(\mathbf{A}, \mathbf{y}) = \text{span}\{\mathbf{y}, \mathbf{Ay}, \mathbf{A}^2\mathbf{y}, \ldots, \mathbf{A}^{k-1}\mathbf{y}\}$, using

$$\mathbf{u}^k = \underset{\mathbf{x} \in \mathcal{K}_k(\mathbf{A}, \mathbf{y})}{\arg \min} \|\mathbf{y} - \mathbf{Ax}\|_{\mathcal{Y}^n}. \tag{7}$$

The OpLanczos method helps to transform problem (7) into a problem in $\mathbb{R}^k$. The OpLanczos method at the $k$-th iteration, tridiagonalizes $\mathbf{A}$ to get $\mathbf{A}V_k = V_k T_k$, where $T_k$ has a tridiagonal structure and $V_k = [v_1\ v_2\ \ldots\ v_k]$, where the $v_i$'s belonging to $\mathcal{Y}^n$ are orthonormal and $v_1$ is generally assumed to be $\mathbf{y}/\|\mathbf{y}\|_{\mathcal{Y}^n}$. Further, the relation $\mathbf{A}V_k = V_{k+1}\overline{T}_k$ is also satisfied for a suitably defined $\overline{T}_k$. Using $V_k$, $\mathbf{x} \in \mathcal{Y}^n$ can be written as $\mathbf{x} = V_k x$. Hence we have:

$$\min_{\mathbf{x} \in \mathcal{K}_k(\mathbf{A}, \mathbf{y})} \|\mathbf{y} - \mathbf{Ax}\|_{\mathcal{Y}^n} = \min_{x \in \mathbb{R}^k} \|\mathbf{y} - \mathbf{A}V_k x\|_{\mathcal{Y}^n} = \min_{x \in \mathbb{R}^k} \|\mathbf{y} - V_{k+1}\overline{T}_k x\|_{\mathcal{Y}^n}$$

$$= \min_{x \in \mathbb{R}^k} \|V_{k+1}(\beta_1 e_1 - \overline{T}_k x)\|_{\mathcal{Y}^n},$$

$$\text{(where } \beta_1 = \|\mathbf{y}\|_{\mathcal{Y}^n}, e_1 = [1\ 0\ \ldots 0]^\top \text{ and } v_1 = \mathbf{y}/\|\mathbf{y}\|_{\mathcal{Y}^n})$$

$$= \min_{x \in \mathbb{R}^k} \|\beta_1 e_1 - \overline{T}_k x\|_2. \quad (\|.\|_2 \text{ is the standard Euclidean norm.})$$

Solving for $x_k = \arg\min_{x \in \mathbb{R}^k} \|\beta_1 e_1 - \bar{T}_k x\|_2$ can be done using QR decomposition (Choi, 2006). Now, the transformation from $\mathbb{R}^k$ back to $\mathcal{Y}^n$ to obtain $\mathbf{u}^k$ is achieved using by the following: $\mathbf{u}^k = V_k x_k = V_k \left(\arg\min_{x \in \mathbb{R}^k} \|\beta_1 e_1 - \bar{T}_k x\|_2\right)$.

## 5 Bounds on Generalization Error

Let $\check{K}$ be a generalized $\mathcal{L}(\mathcal{Y})$-valued kernel on $\mathcal{X}^2$ associated with the function-valued RKKS $\mathcal{K}$. Let $\check{K} = K_1 - K_2$, where $K_1, K_2$ are non-negative $\mathcal{L}(\mathcal{Y})$-valued kernels. From the discussion in Appendix B, an associated function-valued RKHS $\mathcal{H}_\mathcal{K}$ can be obtained for the decomposition $\check{K} = K_1 - K_2$ with the non-negative $\mathcal{L}(\mathcal{Y})$-valued kernel $K = K_1 + K_2$ whose Hilbertian topology defines the strong topology of the Krein space $\mathcal{K}$. We follow Ong et al. (2004) and consider the set $\mathcal{B}_\mathcal{K}$ defined as $\mathcal{B}_\mathcal{K} = \left\{F \in \mathcal{K} \big| \|F_1\|_1^2 + \|F_2\|_2^2 = \|F\|_{\mathcal{H}_\mathcal{K}}^2 \leq 1\right\}$. Consider training data $S = \{(x_i, y_i)\}_{i=1}^n \subset (\mathcal{X} \times \mathcal{Y})$ drawn i.i.d. from an unknown distribution $\mu$. The loss $\ell_y : \mathcal{Y} \to [0, +\infty)$ is defined for every $y \in \mathcal{Y}$ and $F \in \mathcal{K}$ acting on an input $x \in \mathcal{X}$ as $\ell_y(F(x))$. The generalization error (or risk) is defined as $R(F) = \int \ell_y(F(x))d\mu(x, y)$. The empirical error of $F \in \mathcal{B}_\mathcal{K}$ over the training set $S$ is given by $R_e(S, F) = \frac{1}{n}\sum_{i=1}^n \ell_{y_i}(F(x_i))$. We make the following assumptions.

**Assumption 5.1.** $\exists 0 < \kappa < +\infty$ such that $\forall x \in \mathcal{X}, \text{Tr}(K(x, x)) < \kappa$.

**Assumption 5.2.** The loss $\ell_y$ is Lipschitz continuous for every $y \in \mathcal{Y}$ with a Lipschitz constant $\sigma > 0$.

**Assumption 5.3.** $\exists \beta > 0$ such that $\|y\|_\mathcal{Y} < \beta, \forall y \in \mathcal{Y}$.

Assumption 5.1 requires the non-negative $\mathcal{L}(\mathcal{Y})$ kernel $K$ of the associated RKHS $\mathcal{H}_\mathcal{K}$ to be trace class. A similar assumption is also used in (Caponnetto and De Vito, 2006).

Define the *Rademacher average* of $\mathcal{B}_\mathcal{K}$ on a sample $(x_1, \ldots, x_n) \in \mathcal{X}^n$ to be $\mathscr{R}_n(\mathcal{B}_\mathcal{K}) = \mathbb{E}_\mu \mathbb{E}_\varepsilon \sup_{F \in \mathcal{B}_\mathcal{K}} \sum_{i=1}^n \varepsilon_i \ell_{y_i}(F(x_i))$, where $\varepsilon_i$'s are independent Rademacher random variables

uniformly distributed over $\{+1, -1\}$. Now from Assumptions 5.1-5.3 and from (Maurer, 2016, Section 4.3), we have the following bound on the Rademacher average: $\mathscr{R}_n(\mathcal{B}_{\mathcal{K}}) \leq \sqrt{2}\sigma\beta\sqrt{\sum_{i=1}^n \text{Tr}(K(x_i, x_i))}$. The bound on the Rademacher complexity can now be used in (Mendelson, 2003, Corollary 3) to obtain the following result: there is an absolute constant $C$ such that if $n \geq \frac{C}{\epsilon^2} \max\{\mathscr{R}_n^2(\mathcal{B}_{\mathcal{K}}), \log \frac{1}{\delta}\}$, then it holds

$$Pr\{\sup_{F \in \mathcal{B}_{\mathcal{K}}} |R_e(S, F) - R(F)| \geq \epsilon\} \leq \delta. \tag{8}$$

However as noted in (Kadri et al., 2016, Remark 2, page 32), Assumption 5.1 is not always satisfied for all non-negative operator-valued kernels, in which case establishing a bound on the Rademacher average becomes difficult.

The stabilization problem (4) in Section 3, inspired from (Ong et al., 2004) helps in deriving the result in Representer Theorem 3.1. On the other hand, when the stabilizer $\tilde{F}_\lambda$ from Eq. (4) belongs to the ball $\mathcal{B}_{\mathcal{K}}$ of fixed radius $r$ (defined with $r = 1$), it enjoys the generalization bounds in Eq. (8). It is not clear how the stabilizer would behave when it does not belong to $\mathcal{B}_{\mathcal{K}}$. Note that adapting the minimization problem formulation in (Oglic and Gärtner, 2018) would not help here since it leads to complicated variance constraints involving integrals. Further, using a Gateaux derivative approach for the constrained or unconstrained minimization problem similar to that in (Oglic and Gärtner, 2018), leads to difficulties in obtaining the Representer Theorem 3.1 in our work. As a consequence of these facts, we can only resort to an empirical cross-validation approach which we have used in our experiments to ensure that the stabilizer of problem (4) is not far away from $\mathcal{B}_{\mathcal{K}}$.

# 6 Related Work

Since the pioneering works of Ramsay (1982) and Ramsay and Dalzell (1991) on functional data analysis (FDA), there have been significant developments in developing FDA techniques (see *e.g.* non-parametric FDA (Ferraty and Vieu, 2006) and wavelets based FDA (Morettin et al., 2017)). Kernels have been extensively used in machine learning for scalar-valued data (Schölkopf et al., 1999), vector-valued data (Micchelli and Pontil, 2005) and function-valued data (Kadri et al., 2016). Theoretical study on understanding properties of different types of kernels has also been extensive (see *e.g.* (Alpay, 1991, 2001; Carmeli et al., 2006; Caponnetto and De Vito, 2006)). Machine learning with non-positive kernels and scalar-valued RKKS were first proposed for scalar-valued settings in (Ong et al., 2004) and efficient algorithms have been developed in (Oglic and Gärtner, 2018, 2019).

In the context of operator-valued kernels, a prior work by (Zhang et al., 2012) investigates the construction of a positive definite operator-valued kernel $K_r$ called the refinement kernel for a different but fixed positive definite operator-valued kernel $K$, particularly used in multi-task learning. In (Kadri et al., 2012), a finite (positive) linear combination of positive definite operator-valued kernels has been considered, which leads to another positive definite operator-valued kernel. A similar approach can also be found in (Audiffren and Kadri, 2013), where online learning is accomplished using multiple operator-valued kernels.

Among other works on learning using function-valued data, Oliva et al. (2015) approximate function-valued data using projections onto a custom orthogonal basis (called 3BE). This yields a regression problem where the basis coefficients associated with input functional data are used to estimate the basis coefficients of output functional data. A related projection-based approach KPL in (Bouche et al., 2020) approximates the output space $\mathcal{Y}$ by a finite-dimensional Euclidean space $\mathbb{Y} \subset \mathbb{R}^D$, assumed to be the linear span of a suitable (not necessarily orthogonal) basis. Thus Bouche et al. (2020) propose to learn the function $h : \mathcal{X} \to \mathbb{Y}$, by optimizing a suitable regularized functional loss. Empirical loss minimization in purely functional setup for additive function-on-function regression is considered in (Reimherr and Sriperumbudur, 2017). A Bayesian approach considered in (Shi and Choi, 2011), imposes a data-driven Gaussian process prior for estimating a function-valued function.

# 7 Experiments

We consider the functional regression problem for our experiments. Let $\mathcal{X} = L^2(\Omega_x)$, $\mathcal{Y} = L^2(\Omega_y)$ for some suitable $\Omega_x$ and $\Omega_y$. The aim is to learn a function-valued function $F : \mathcal{X} \to \mathcal{Y}$. However as noted in Section 1, in practical applications, $x(s) \in \mathcal{X}$ and $y(t) \in \mathcal{Y}$ are not available $\forall s \in \Omega_x$ and $\forall t \in \Omega_y$. Instead only discrete observations $\{x_p\}_{p=1}^P \subset \Omega_x$ and $\{y_q\}_{q=1}^Q \subset \Omega_y$ are observed. However we can approximate these discrete observations as functions using FDA techniques like

B-splines or Fourier bases, so that the generalized operator-valued framework introduced in the previous sections can be used. The error metric used for evaluating output functions is residual sum of squares error (RSSE) defined as $RSSE = \int \sum_i \{y_i(t) - \hat{y}_i(t)\}^2 dt$ (Kadri et al., 2016), where $y_i$ is the actual output and $\hat{y}_i$ is the predicted output function. We use RSSE since it is suitable for the functional nature of the outputs in a functional regression problem. Numerical integration techniques (Hamming, 2012) were used to compute the integrals.

**Speech Inversion.** We consider the application of speech inversion, where based on input audio signals, the Vocal Tract (VT) variables (*e.g.* Lip Aperture (LA), Lip Protrusion (LP), Jaw Angle (JA)) are approximated in order to understand the movements of human body parts which create particular sounds. Speech inversion finds use in applications like lip reading and speech understanding. We use the Haskins IEEE Rate Comparison DB dataset available at `https://yale.app.box.com/s/cfn8hj2puveo65fq54rp1ml2mk7moj3h` (Tiede et al., 2017). The dataset details are given in Appendix I. The data was pre-processed to trim the samples to the smallest speech recording ($\approx$1.73 seconds). Recordings where complete data was not available were excluded. The input sounds were used to create 13 mel cepstral coefficients (MFCCs) acquired each 12 milliseconds with a window duration of 46 milliseconds. For each input audio sample, the MFCCs are available as 13 vectors each of size 149. Each output function of Lip Aperture (LA) VT variable is sampled at 174 points. The functional output data corresponding to LA was constructed using an orthonormal trigonometric basis of $n_b$ elements.

**Experimental Setting:** All methods were coded in Python 3.6 and the codes are made public[1]. All experiments were run on a Linux box with 182 Gigabytes main memory and 28 CPU cores. The experiments performed used 320 samples for training and 80 samples for testing. For hyperparameter tuning, we used 3-fold multi-grid cross validation for all the methods. For the encoding of LA functions, we cross-validated the $n_b$ parameter from the set $\{10, 20, 30, 40, 50\}$ for all methods. The following methods are considered for comparison.

**OpMINRES.** We considered the generalized operator-valued kernel in Eq. (3), where we used the following choices for output kernel $h(s,t)$: $e^{-\gamma|t-s|}$ (ABS), $e^{-\gamma(t-s)^2}$ (SQ), $e^{-\gamma_1|t-s|} - e^{-\gamma_2|t-s|}$ (DIFFABS), $e^{-\gamma_1(t-s)^2} - e^{-\gamma_2(t-s)^2}$ (DIFFSQ), $e^{-\gamma_1|t-s|} - e^{-\gamma_2(t-s)^2}$ (DIFFABSSQ) and $e^{-\gamma_1(t-s)^2} - e^{-\gamma_2|t-s|}$ (DIFFSQABS). The following choices for the input kernel $g(x,z)$ were used: $e^{-\eta\|x-z\|^2}$ (RBF), $e^{-\eta_1\|x-z\|^2} - e^{-\eta_2\|x-z\|^2}$ (DIFFGAUSS) and $\max(0, 1 - \eta\|x - z\|^2)$ (EPAN). $\lambda$ was chosen from $\{10^{-3}, 10^{-2}, 0.1, 1, 10, 100\}$. $\gamma, \gamma_1, \gamma_2, \eta, \eta_1, \eta_2$ were chosen from $\{0.001, 0.002, \ldots, 0.009, 0.01, 0.02, \ldots, 0.09, 0.1, 0.2, \ldots, 0.9, 1, 2, \ldots, 10, 20, \ldots, 100\}$. The per-iteration complexity for OpMINRES is $O(nQ^3 + nQ^2 n_b + n^3 Q n_b)$, where $n$ is number training samples, $Q$ is the discretization size in each LA output and $n_b$ is the cardinality of the basis considered.

**3BE.** (Oliva et al., 2015) Here, the encoding was done only for the output functions using a trigonometric basis of $n_b$ elements and the input MFCCs were considered in their vector form. An RBF kernel $e^{-\eta\|x-z\|^2}$ for inputs was considered and range for $\eta$ was chosen similar to OpMINRES. The regularization parameter $\lambda$ of 3BE was chosen from $\{10^{-3}, 10^{-2}, 0.1, 1, 10, 100\}$.

**KPL.** (Bouche et al., 2020) The dictionary for LA outputs was an orthonormal basis of $n_b$ trigonometric functions. A separable kernel of the type $K(x_i, x_j) = g(x_i, x_j)B$ was chosen where $B$ is a $n \times n$ diagonal matrix with $B_{ii} = 1/b^{n-i}$. An RBF kernel $e^{-\eta\|x-z\|^2}$ for the inputs was chosen where $\eta$ was chosen similar to OpMINRES. For matrix $B$, the value of $b$ was chosen from $\{0.1, 1, 10, 20, 50, 100\}$. Computing the $\boldsymbol{\eta}^k$ parameter using sample average did not yield good results, hence we chose $\boldsymbol{\eta}^k = \Phi_{(n)}^{\#}\mathbf{y}$ (Bouche et al., 2020). The regularization parameter $\lambda$ of KPL was chosen from $\{10^{-3}, 10^{-2}, 0.1, 1, 10, 100\}$.

**Non-negative Operator-valued kernel approach (NOVK).** (Kadri et al., 2016) Note that the resultant matrix operator equation in (Kadri et al., 2016) is similar to Eq. (6). Hence OpMINRES was used for obtaining the solution. ABS and SQ were used as output kernels. RBF was used as input kernel. All parameters were cross-validated similar to OpMINRES.

The results given in Table 1 show that OpMINRES for the proposed generalized operator-valued kernel and function valued RKKS approach attains comparable performance, while allowing for more

choices and flexibility in choosing the input and output kernels. In terms of runtime, 3BE was faster than all methods. The time taken for KPL, OpMINRES for NOVK and OpMINRES for our approach were comparable. Experiments on other data sets are provided in Appendix I.

| Method | Input Kernel | Output kernel | Best Test RSSE |
|---|---|---|---|
| NOVK | RBF | ABS | 5.4031 |
| NOVK | RBF | SQ | 5.4836 |
| 3BE | RBF | – | 5.4314 |
| KPL | RBF | – | 5.3566 |
| OpMINRES | RBF | DIFFABS | 5.4897 |
| | RBF | DIFFSQ | 5.5169 |
| | RBF | DIFFABSSQ | 5.4905 |
| | RBF | DIFFSQABS | 5.5167 |
| | DIFFGAUSS | ABS | 5.3956 |
| | DIFFGAUSS | SQ | 5.4007 |
| | EPAN | ABS | 5.3494 |
| | EPAN | SQ | 5.4086 |

Table 1: Test RSSE Comparison Results

# 8   Conclusion

In this paper, we have developed theoretical tools useful for generalized operator-valued kernels, which are not necessarily non-negative, and have discussed results establishing the association between generalized operator-valued kernel and its associated function-valued reproducing kernel Krein space (RKKS). We formulated a learning problem and provided a representer theorem, and analyzed the generalization error bounds. We proposed an iterative operator minimum residual algorithm for solving an operator matrix equation resulting from the learning problem, which has been implemented on practical data sets. Experiments show the usefulness of the proposed theoretical framework, allowing for flexible choices of indefinite kernels in functional regression problems.

## Broader Impact

The theoretical tools introduced in the paper for generalized operator-valued kernels and function-valued Reproducing Kernel Krein Spaces (RKKS) are new and will promote research in investigating more sophisticated techniques for handling function data and other data with complicated structures. The proposed methods and algorithms have been applied on a speech inversion problem and accurate predictions of function-valued outputs in such applications might be useful for improving the current understanding of the speech generation process in humans. To the best of our knowledge, our work does not have any negative impact.

## Acknowledgments and Disclosure of Funding

The authors thank the anonymous reviewers for their useful comments. The work of the second author is partially supported by the starting SEED grant from IIT Bombay. The authors declare no competing interests.

## Footnotes

[1]The codes used for experiments can be found at `https://github.com/akashsaha06/NeurIPS-2020/`

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
