[Supplementary Material 1]

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

# A Primer on Hilbert spaces and RKHS

In this primer, we cover certain formal definitions in order to lay the framework for establishing the properties of reproducing kernel Hilbert spaces (RKHS). A detailed account on RKHS and scalar-valued kernels can be found in (Schölkopf et al., 1999) and (Shawe-Taylor et al., 2004).

## A.1 Hilbert Spaces

Let $\mathcal{H}$ be a vector space defined on the field $\mathbb{R}$ of real numbers (arbitrary fields can be considered). An *inner product* on $\mathcal{H}$ is a function $\langle \cdot, \cdot \rangle_{\mathcal{H}} : \mathcal{H} \times \mathcal{H} \to \mathbb{R}$, such that $\forall f, g, h \in \mathcal{H}$ and scalars $\alpha, \beta \in \mathbb{R}$, it satisfies:

1. $\langle f + g, h \rangle_{\mathcal{H}} = \langle f, h \rangle_{\mathcal{H}} + \langle g, h \rangle_{\mathcal{H}}$
2. $\langle \alpha f, g \rangle_{\mathcal{H}} = \alpha \langle f, g \rangle_{\mathcal{H}}$
3. $\langle f, g \rangle_{\mathcal{H}} = \langle g, f \rangle_{\mathcal{H}}$
4. $\langle f, f \rangle_{\mathcal{H}} \geq 0$ and equality holds if and only if $f = 0$.

**Definition A.1. Hilbert Space:** A Hilbert space is a vector space $\mathcal{H}$ on $\mathbb{R}$ (arbitrary fields can be considered) with an inner product $\langle \cdot, \cdot \rangle_{\mathcal{H}}$ such that the norm defined by $\|f\|_{\mathcal{H}} = \sqrt{\langle f, f \rangle_{\mathcal{H}}}$ turns $\mathcal{H}$ into a complete metric space. By completeness, we denote that for every Cauchy sequence $\{f_n\}_{n=1,2,\dots} \in \mathcal{H}$, there exists an element $f \in \mathcal{H}$ such that $\lim_{n \to \infty} \|f_n - f\|_{\mathcal{H}} = 0$.

When the context of $\mathcal{H}$ is clear, we use $\|f\|$ instead of $\|f\|_{\mathcal{H}}$.

## A.2 Scalar-valued RKHS

Now, having introduced inner product and Hilbert spaces, we consider learning functions $f : X \to Y$, where $X$ is a suitable input space (typically $X = \mathbb{R}^n$ for some $n \in \mathbb{N}$, the set of natural numbers) and $Y = \mathbb{R}$ is the output space. Let us assume that $L^2(X)$ denotes the space of equivalence classes of square integrable functions from $X$ to $\mathbb{R}$ for some measurable $X$. Now, we can define Evaluation functional which can be used to characterize scalar-valued reproducing kernel Hilbert spaces (RKHS).

**Definition A.2. Evaluation functional:** Let $X$ be a suitable space of inputs. Consider a Hilbert space $\mathcal{H} \subset \mathbb{R}^X$. The evaluation functional $\Xi_x$ associated with $\mathcal{H}$ valuates a function $f \in \mathcal{H}$ at $x \in X$, and is defined as

$$\Xi_x : \mathcal{H} \to \mathbb{R}, \text{ where } \mathcal{H} \ni f \mapsto \Xi_x f = f(x) \in \mathbb{R}.$$

**Definition A.3. Scalar-valued RKHS:** A Hilbert space $\mathcal{H}$ is a scalar-valued reproducing kernel Hilbert space if $\mathcal{H} \subset \mathbb{R}^X$ and the associated evaluation functional is bounded: $\forall x \in X, \exists \lambda_x \geq 0$ such that $\forall f \in \mathcal{H}$,

$$|f(x)| = |\Xi_x f| \leq \lambda_x \|f\|_{\mathcal{H}}.$$

It is clear that evaluation functionals are always linear. For $f, g \in \mathcal{H}$ and $\alpha, \beta \in \mathbb{R}$, $\Xi_x(\alpha f + \beta g) = (\alpha f + \beta g)(x) = \alpha f(x) + \beta g(x) = \alpha \Xi_x(f) + \beta \Xi_x(g)$. A natural way to define scalar-valued RKHS can be the continuity of evaluation functional (*e.g.* Definition 4.18 (ii) in (Steinwart and Christmann, 2008)).

Having provided the definitions of scalar-valued RKHS, we can proceed to understanding reproducing kernels.

## A.3 Reproducing Kernels

**Definition A.4. Reproducing Kernel:** (Berlinet and Thomas-Agnan, 2011, Definition 1.) Let $\mathcal{H}$ be a Hilbert space of scalar-valued functions defined on an input space $X$. A function $k : X \times X \to \mathbb{R}$ is called a reproducing kernel of $\mathcal{H}$ if it satisfies:

1. $\forall x \in X, k(x, \cdot) \in \mathcal{H}$,
2. **(reproducing property)** $\forall x \in X, \forall f \in \mathcal{H}, \langle f, k(x, \cdot) \rangle_{\mathcal{H}} = f(x)$.

In particular, for any $x, y \in X$,

$$k(x, y) = \langle k(x, \cdot), k(y, \cdot) \rangle_{\mathcal{H}}.$$

Now, having defined reproducing kernels with respect to a Hilbert space, we provide a more general definition of kernels.

**Definition A.5. Kernels:** A function $k : X \times X \to \mathbb{R}$ is called a kernel on $X^2$ if there exists a Hilbert space (not necessarily RKHS) $\mathcal{G}$ and a map $\phi : X \to \mathcal{G}$, such that $k(x, y) = \langle \phi(x), \phi(y) \rangle_{\mathcal{G}}$.

In the above definition, the map $\phi$ is called a feature map and $\mathcal{G}$ is a feature space. It is straightforward that every reproducing kernel is a kernel with $\phi : x \mapsto k(x, .), k(x, y) = \langle k(x, .), k(y, .) \rangle_{\mathcal{H}}$. The following property helps us in obtaining a characterization of scalar-valued RKHS and kernels.

**Definition A.6. Positive Definiteness:** A symmetric function $k$ on $X^2$ is positive definite (or non-negative) if for any $f \in L^2(X)$,

$$\int \int f(x) k(x, x') f(x') dx dx' \geq 0.$$

In literature, the words positive, positive semi-definite, positive definite and non-negative have been used equivalently. In order to remove ambiguity, we use positive definiteness with the above mentioned definition. Additionally, kernels satisfying the positive definiteness property are known as Mercer kernels. The above definition generalizes the definition for matrices since for any finite subset of $X$, we obtain that the Gram matrix $(K)_{i,j} = k(x_i, x_j)$ is positive definite. The following lemma provides a relation between reproducing kernels and positive definiteness.

**Lemma A.1.** Let $X$ be an input space and let $\mathcal{H}$ be an RKHS on $X$ with reproducing kernel $k$, then $k$ is positive definite.

The following result provides a characterization for positive definite kernels and RKHS. The result relates a positive definite kernel with a corresponding RKHS (see (Moore, 1935; Aronszajn, 1950)).

**Theorem A.2. (Moore-Aronszajn Theorem)** Let $k : X \times X \to \mathbb{R}$ be positive-definite. There exists a unique RKHS $\mathcal{H} \subset \mathbb{R}^X$ with reproducing kernel $k$. The subspace $\mathcal{H}_0$ of $\mathcal{H}$ spanned by the functions $(k(x, .)_{x \in X})$ is dense in $\mathcal{H}$ and $\mathcal{H}$ is the set of functionals on $X$ which are pointwise limits of Cauchy sequences in $\mathcal{H}_0$ with the inner product

$$\langle f, g \rangle_{\mathcal{H}_0} = \sum_{i=1}^{n} \sum_{j=1}^{n} \alpha_i \beta_j k(x_i, x_j), \text{ where } f = \sum_{i=1}^{n} \alpha_i k(x_i, .) \text{ and } \sum_{j=1}^{n} \beta_j k(x_j, .).$$

The above theorem associates a scalar-valued RKHS with any positive definite kernel. Therefore, there is a bijection between the set of scalar-valued RKHS and the set of positive definite kernels.

# B Reproducing Kernel Krein Spaces (RKKS)

We start this section by providing a brief introduction to Krein spaces and then provide characterization of scalar-valued reproducing kernel Krein spaces (RKKS) and recall some of their properties. A more thorough introduction of Krein spaces can be found in (Bognar, 1974) and (Azizov and Iokhvidov, 1989) and results related to scalar-valued RKKS can be found in (Alpay, 2001; Ong et al., 2004).

## B.1 Krein Spaces

Let $\mathcal{K}$ be a vector space defined on the field $\mathbb{R}$ of real numbers (we restrict our attention to $\mathbb{R}$ for simplicity, noting that arbitrary fields can be considered). A *bilinear form* on $\mathcal{K}$ is a function $\langle \cdot, \cdot \rangle_{\mathcal{K}} : \mathcal{K} \times \mathcal{K} \to \mathbb{R}$ such that, $\forall f, g, h \in \mathcal{K}$ and scalars $\alpha, \beta \in \mathbb{R}$, it satisfies:

1. $\langle \alpha f + \beta g, h \rangle_{\mathcal{K}} = \alpha \langle f, h \rangle_{\mathcal{K}} + \beta \langle f, h \rangle_{\mathcal{K}}$, and
2. $\langle f, \alpha g + \beta f \rangle_{\mathcal{K}} = \alpha \langle f, g \rangle_{\mathcal{K}} + \beta \langle f, h \rangle_{\mathcal{K}}$.

For $f \in \mathcal{K}$, if $\langle f, g \rangle_{\mathcal{K}} = 0$, $\forall g \in \mathcal{K}$ implies $f = 0$, then the bilinear form is called non-degenerate. The bilinear form $\langle \cdot, \cdot \rangle_{\mathcal{K}}$ is symmetric if, $\langle f, g \rangle_{\mathcal{K}} = \langle g, f \rangle_{\mathcal{K}}$, $\forall f, g \in \mathcal{K}$. The form is called indefinite if there exists $f, g \in \mathcal{K}$ such that $\langle f, f \rangle_{\mathcal{K}} > 0$ and $\langle g, g \rangle_{\mathcal{K}} < 0$. If $\langle f, f \rangle_{\mathcal{K}} \geq 0$, $\forall f \in \mathcal{K}$, then the form is called positive. A non-degenerate, symmetric and positive bilinear form on $\mathcal{K}$ is called *inner product*.

Any two elements $f, g \in \mathcal{K}$ that satisfy $\langle f, g \rangle_{\mathcal{K}} = 0$ are $\langle \cdot, \cdot \rangle_{\mathcal{K}}$-orthogonal. Similarly, any two subspaces $\mathcal{K}_1, \mathcal{K}_2 \subset \mathcal{K}$ that satisfy $\langle f_1, f_2 \rangle_{\mathcal{K}} = 0, \forall f_1 \in \mathcal{K}_1$ and $\forall f_2 \in \mathcal{K}_2$ are called $\langle \cdot, \cdot \rangle_{\mathcal{K}}$-orthogonal. Krein spaces can now be defined based on the notion of a bilinear form.

**Definition B.1. Krein Space.** A vector space $\mathcal{K}$ endowed with a non-degenerate symmetric bilinear form $\langle \cdot, \cdot \rangle_{\mathcal{K}}$ is called a Krein space if it admits a decomposition into a direct sum $\mathcal{K} = \mathcal{H}_1 \oplus \mathcal{H}_2$ of $\langle \cdot, \cdot \rangle_{\mathcal{K}}$-orthogonal Hilbert spaces $\mathcal{H}_1, \mathcal{H}_2$, endowed with inner products $\langle \cdot, \cdot \rangle_{\mathcal{H}_1}, \langle \cdot, \cdot \rangle_{\mathcal{H}_2}$, such that the bilinear form can be written as

$$\langle f, g \rangle_{\mathcal{K}} = \langle f_1, g_1 \rangle_{\mathcal{H}_1} - \langle f_2, g_2 \rangle_{\mathcal{H}_2},$$

where $f_1, g_1 \in \mathcal{H}_1, f_2, g_2 \in \mathcal{H}_2$ and $f = f_1 + f_2, g = g_1 + g_2$.

Notice that, despite the non-negativity of inner products $\langle \cdot, \cdot \rangle_{\mathcal{H}_1}$ and $\langle \cdot, \cdot \rangle_{\mathcal{H}_2}$, the bilinear form $\langle \cdot, \cdot \rangle_{\mathcal{K}}$ might be indefinite. As we describe later, this property of Krein spaces is particularly useful in developing reproducing kernel Krein spaces, which can be suitably identified with the space of indefinite reproducing kernels.

Now we define an associated Hilbert space of the Krein space, where the Hilbertian inner product structure is preserved.

**Definition B.2. Associated Hilbert Space.** Let $\mathcal{K}$ be a Krein space admitting a decomposition into Hilbert spaces $\mathcal{H}_1$ and $\mathcal{H}_2$. Then the associated Hilbert space is defined by $\mathcal{H}_{\mathcal{K}} = \mathcal{H}_1 \oplus \mathcal{H}_2$, endowed with the inner product: $\langle f, g \rangle_{\mathcal{H}_{\mathcal{K}}} = \langle f_1, g_1 \rangle_{\mathcal{H}_1} + \langle f_2, g_2 \rangle_{\mathcal{H}_2}$.

The decomposition of a Krein space $\mathcal{K} = \mathcal{H}_1 \oplus \mathcal{H}_2$ in not necessarily unique. Therefore, a Krein space in general, can be associated with infinitely many Hilbert spaces. But, for any such associated Hilbert space $\mathcal{H}_{\mathcal{K}}$, the topology introduced on $\mathcal{K}$ via the norm $\|f\|_{\mathcal{H}_{\mathcal{K}}} = \sqrt{\langle f, f \rangle_{\mathcal{H}_{\mathcal{K}}}}$ is independent of the decomposition and the associated Hilbert space. The topology on $\mathcal{K}$ defined by the norm of an associated Hilbert space is known as the strong topology on $\mathcal{K}$. The notions of continuity and convergence in a Krein space are defined with respect to the strong topology.

## B.2 Scalar-valued RKKS

Having introduced Krein spaces, we can now adapt them to aid in predictive machine learning applications which aim at learning functions of the form $f : X \to Y$, where $X$ is a suitable input space and $Y = \mathbb{R}$ is the output space. Accordingly, we define a scalar-valued reproducing kernel Krein space (RKKS) and discuss few relevant results.

**Definition B.3. Evaluation functional.** Let $X$ be a suitable space of inputs. Consider a Krein space $\mathcal{K} \subset \mathbb{R}^X$. The evaluation functional $\Xi_x$ that evaluates a function $f \in \mathcal{K}$ at $x \in X$, is defined as

$$\Xi_x : \mathcal{K} \to \mathbb{R}, \text{ where } \mathcal{K} \ni f \mapsto \Xi_x f = f(x) \in \mathbb{R}.$$

**Definition B.4. Scalar-valued RKKS.** (Alpay, 2001) A Krein space $(\mathcal{K}, \langle \cdot, \cdot \rangle_{\mathcal{K}})$ is a scalar-valued reproducing kernel Krein space if $\mathcal{K} \subset \mathbb{R}^X$ and the evaluation functional is continuous on $\mathcal{K}$ with respect to its strong topology.

By restricting the functions $f : X \to \mathbb{R}$ to be such that $f \in \mathcal{K}$, where $\mathcal{K}$ is a scalar-valued RKKS, the next result on the reproducing property of a generalized kernel $\breve{k}$ (which might be indefinite), associated with the scalar-valued RKKS, allows us to learn $f$ using $\breve{k}$.

**Theorem B.1. (Reproducing Kernel)** (Ong et al., 2004) Let $\mathcal{K}$ be a scalar-valued RKKS with decomposition into Hilbert spaces $\mathcal{H}_1$ and $\mathcal{H}_2$. Then

- $\mathcal{H}_1$ and $\mathcal{H}_2$ are scalar-valued RKHS (with kernels $k_1$ and $k_2$).
- **(Reproducing property)** There is a unique symmetric $\breve{k}(x, x')$ with $\breve{k}(x, .) \in \mathcal{K}$, such that for all $f \in \mathcal{K}$, $\langle f, \breve{k}(x, .) \rangle_{\mathcal{K}} = f(x)$.
- $\breve{k} = k_1 - k_2$.

Indeed, analogous to a scalar-valued RKHS where the availability of a reproducing positive kernel is guaranteed (Schölkopf et al., 1999), at least one generalized kernel $\breve{k}$ can be associated with a scalar-valued RKKS as described in the next result.

**Theorem B.2.** (Mary, 2003) Let $\breve{k}$ be a symmetric real valued function on $X^2$, where $X$ is the input space. Then the following are equivalent:

- There exists (at least) one scalar-valued RKKS with kernel $\breve{k}$.

- $\check{k}$ admits a positive decomposition, that is there exists two positive kernels $k_1$ and $k_2$, such that $\check{k} = k_1 - k_2$.
- $\check{k}$ is dominated by some positive kernel $p$ (i.e., $p - \check{k}$ is a positive kernel).

Notice however that unlike the bijection between the set of scalar-valued RKHS and the set of Mercer kernels, there is only a surjection between the set of scalar-valued RKKS and the set of generalized kernels defined in the vector space generated by the set of all Mercer kernels over $X$ (Ong et al., 2004).

## C    Proofs of Lemmas and Theorem in Section 2

We recall the results in Section 2 and discuss the proofs here.

**Lemma C.1.** Let $K_1$ and $K_2$ be two $\mathcal{L}(\mathcal{Y})$-valued non-negative kernels on $\mathcal{X}^2$ with corresponding Hilbert spaces $\mathcal{H}_1$ and $\mathcal{H}_2$ respectively. Then the intersection $\mathcal{H}_1 \cap \mathcal{H}_2$ with the inner product

$$\langle f, f \rangle_{\mathcal{H}_1 \cap \mathcal{H}_2} = \langle f, f \rangle_{\mathcal{H}_1} + \langle f, f \rangle_{\mathcal{H}_2} \tag{9}$$

is a reproducing kernel Hilbert space contractively included in $\mathcal{H}_1$ and $\mathcal{H}_2$.

*Proof.* The intersection $\mathcal{H} = \mathcal{H}_1 \cap \mathcal{H}_2$ endowed with the inner product $\langle \cdot, \cdot \rangle$ in (9) is a pre-Hilbert space. Let $F_n(.)$ be a Cauchy sequence in $\mathcal{H}$. Then it is also a Cauchy sequence in $\mathcal{H}_1$ and $\mathcal{H}_2$, and thus there exists $F(.)$ in $\mathcal{H}_1$ and $G(.)$ in $\mathcal{H}_2$ such that

$$\lim_{n \to \infty} F_n(.) = F(.)$$

in the $\mathcal{H}_1$ norm, and

$$\lim_{n \to \infty} F_n(.) = G(.)$$

in the $\mathcal{H}_2$ norm. For $w \in \mathcal{X}, u \in \mathcal{Y}$,

$$\lim_{n \to \infty} \langle F_n(w), u \rangle_{\mathcal{Y}} = \langle \lim_{n \to \infty} F_n(w), u \rangle_{\mathcal{Y}} = \langle F(w), u \rangle_{\mathcal{Y}}$$
$$= \langle \lim_{n \to \infty} F_n(w), u \rangle_{\mathcal{Y}} = \langle G(w), u \rangle_{\mathcal{Y}}$$
$$\langle F(w), u \rangle_{\mathcal{Y}} = \langle G(w), u \rangle_{\mathcal{Y}}$$
$$\implies F(w) = G(w).$$

Hence, $F(.) = G(.)$ and $F \in \mathcal{H}$. For a Cauchy sequence $F_n(.)$ in $\mathcal{H}$, $\lim_{n \to \infty} F_n(.) = F(.) \in \mathcal{H}$, which proves $\mathcal{H}$ is a Hilbert space. In order to prove that $\mathcal{H}$ is a reproducing kernel Hilbert space, based on (Carmeli et al., 2006, Definition 2.1) and (Carmeli et al., 2010) we use the fact that for $\mathcal{H}_1$ and $\mathcal{H}_2$ which are RKHS, for every $w \in \mathcal{X}$ there exist positive constants $M_w, G_w$ such that

$$\|F(w)\|_{\mathcal{Y}} \leq M_w \|F(.)\|_{\mathcal{H}_1}, \|F(w)\|_{\mathcal{Y}} \leq G_w \|F(.)\|_{\mathcal{H}_2}, \ \forall w \in \mathcal{X}$$
$$\implies \|F(w)\|_{\mathcal{Y}}^2 \leq M_w^2 \|F(.)\|_{\mathcal{H}_1}^2, \|F(w)\|_{\mathcal{Y}}^2 \leq G_w^2 \|F(.)\|_{\mathcal{H}_2}^2, \ \forall w \in \mathcal{X}$$
$$\implies \|F(w)\|_{\mathcal{Y}}^2 \leq P_w^2 (\|F(.)\|_{\mathcal{H}_1}^2 + \|F(.)\|_{\mathcal{H}_2}^2), \ \forall w \in \mathcal{X}, \text{ where } P_w = \min\{M_w, G_w\} \tag{10}$$
$$\implies \|F(w)\|_{\mathcal{Y}} \leq P_w \|F(.)\|_{\mathcal{H}}, \ \forall w \in \mathcal{X}. \tag{11}$$

We obtain inequality in (11) from the inequality in (10) using $\|F(.)\|_{\mathcal{H}_1 \cap \mathcal{H}_2}^2 = \|F(.)\|_{\mathcal{H}_1}^2 + \|F(.)\|_{\mathcal{H}_2}^2$ from (9). Hence, $\mathcal{H}$ is a reproducing kernel Hilbert space. $\qquad \square$

In the above proof, $\mathcal{H}_1 \cap \mathcal{H}_2$ being contractively included in $\mathcal{H}_1$ and $\mathcal{H}_2$ follows based on the definition of $\langle \cdot, \cdot \rangle_{\mathcal{H}_1 \cap \mathcal{H}_2}$ in (9) as $\|F\|_{\mathcal{H}_1} \leq \|F\|_{\mathcal{H}_1 \cap \mathcal{H}_2}$ and $\|F\|_{\mathcal{H}_2} \leq \|F\|_{\mathcal{H}_1 \cap \mathcal{H}_2}, \forall F \in \mathcal{H}_1 \cap \mathcal{H}_2$. Before proceeding to the next lemma, we recall the definition of an inductive set.

**Definition C.1. Inductive Set:** An ordered set $S$ is said to be inductive if every totally ordered subset of $S$ has an upper bound in $S$.

Notice that Lemma C.1 helps us to create a RKHS using intersection of the function-valued RKHS associated with two non-negative operator-valued kernels on $\mathcal{X}^2$. We proceed to obtain a partial order on $I(K_1, K_2)$ defined in Lemma C.2, which is also shown to be inductive.

**Lemma C.2.** Let $K_1$ and $K_2$ be two $\mathcal{L}(\mathcal{Y})$-valued non-negative kernels on $\mathcal{X}^2$ and let $I(K_1, K_2)$ denote the set of all functions $K$ non-negative on $\mathcal{X}^2$ and such that $K \leq K_1$ and $K \leq K_2$. Then $I(K_1, K_2)$ is inductive.

*Proof.* In this proof, we first consider an ordered subset of $I(K_1, K_2)$. Then, we proceed to find the limit of the ordered subset. Finally, we establish that the limit belongs to $I(K_1, K_2)$.

Let $(K_j)_{j \in J}$ be an ordered subset of $I(K_1, K_2)$, where $J$ is a suitable index set. Then, for each $x \in \mathcal{X}$ and $y \in \mathcal{Y}$,

$$\{\langle K_j(x,x)y, y\rangle_{\mathcal{Y}}\}_{j \in J}$$

is an increasing bounded sequence of non-negative numbers.

Let $x \in \mathcal{X}, y \in \mathcal{Y}$ and $i \leq j$ $(i, j \in J)$. Let $\mathcal{H}_i, \mathcal{H}_j$ be the RKHS corresponding to $K_i, K_j$ respectively. Let $K = K_j - K_i$, we see that $K$ is a non-negative $\mathcal{L}(\mathcal{Y})$-valued kernel on $\mathcal{X}^2$ using the fact that $K_i \leq K_j$. Consider $\mathcal{H}$ to be the RKHS corresponding to $K$. Now, using $K_j = K + K_i$ and the definition of Function-valued RKHS in (Kadri et al., 2016), we obtain

$$\mathcal{H}_j = \{K(x,.)y + K_i(x,.)y | x \in \mathcal{X}, y \in \mathcal{Y}\}.$$

In particular, when $K(x,.)y = 0$, then $\mathcal{H}_i \subset \mathcal{H}_j$. Now, based on Lemma C.1, we obtain $\mathcal{H}_i$ is contractively included in $\mathcal{H}_j$ with a new norm $\|.\|_{\mathcal{H}_i}$ based on (9) and the reproducing property of $K_i$ gives us

$$\|K_i(x,.)y\|^2_{\mathcal{H}_j} \leq \|K_i(x,.)y\|^2_{\mathcal{H}_i} = \langle K_i(x,.)y, K_i(x,.)y\rangle_{\mathcal{H}_i} = \langle K_i(x,x)y, y\rangle_{\mathcal{Y}}$$

For $x, w \in \mathcal{X}, y, g \in \mathcal{Y}$,

$$\langle K_i(x,w)g - K_j(x,w)g, y\rangle_{\mathcal{Y}} = \langle K_i(x,w)g, y\rangle_{\mathcal{Y}} - \langle K_j(x,w)g, y\rangle_{\mathcal{Y}}$$
$$= \langle K_i(x,.)g, K_j(w,.)y\rangle_{\mathcal{H}_j} - \langle K_j(x,.)g, K_j(w,.)y\rangle_{\mathcal{H}_j} \quad (12)$$
$$= \langle K_i(x,.)g - K_j(x,.)g, K_j(w,.)y\rangle_{\mathcal{H}_j} \quad (13)$$

Equations (12) and (13) follow from the reproducing property of $K_j$ and properties of $\langle \cdot, \cdot \rangle_{\mathcal{Y}}$. Using Cauchy-Schwartz inequality, we obtain from Eq. (13),

$$|\langle K_i(x,w)g - K_j(x,w)g, y\rangle_{\mathcal{Y}}|^2 \leq \|K_j(w,.)y\|^2_{\mathcal{H}_j} \|K_i(x,.)g - K_j(x,.)g\|^2_{\mathcal{H}_j}.$$

Now using the reproducibility of $K_i$, $\langle K_i(x,.)g, K_j(x,.)g\rangle_{\mathcal{H}_j} = \langle K_j(x,x)g, g\rangle_{\mathcal{Y}}$, and we have

$$\|K_i(x,.)g - K_j(x,.)g\|^2_{\mathcal{H}_j} = \|K_i(x,.)g\|^2_{\mathcal{H}_j} + \|K_j(x,.)g\|^2_{\mathcal{H}_j} - 2\langle K_i(x,.)g, K_j(x,.)g\rangle_{\mathcal{H}_j}$$
$$= \|K_i(x,.)g\|^2_{\mathcal{H}_j} + \|K_j(x,.)g\|^2_{\mathcal{H}_j} - 2\langle K_j(x,x)g, g\rangle_{\mathcal{Y}}$$
$$= \langle K_i(x,x)g, g\rangle_{\mathcal{Y}} + \langle K_j(x,x)g, g\rangle_{\mathcal{Y}} - 2\langle K_j(x,x)g, g\rangle_{\mathcal{Y}}$$
$$= \langle K_i(x,x)g, g\rangle_{\mathcal{Y}} - \langle K_j(x,x)g, g\rangle_{\mathcal{Y}}$$
$$= \langle K_i(x,x)g - K_j(x,x)g, g\rangle_{\mathcal{Y}}.$$

Therefore,

$$|\langle K_i(x,w)g - K_j(x,w)g, y\rangle_{\mathcal{Y}}|^2 \leq \langle K_j(w,w)y, y\rangle_{\mathcal{Y}}\langle K_i(x,x)g - K_j(x,x)g, g\rangle_{\mathcal{Y}}. \quad (14)$$

Now, using inequality in (14), when $i \to \infty$ then $\langle K_i(x,x)g - K_j(x,x)g, g\rangle_{\mathcal{Y}} \to 0$ as $\{\langle K_j(x,x)g, g\rangle_{\mathcal{Y}}\}_{j \in J}$ is an increasing bounded sequence of non-negative numbers. Thus $|\langle K_i(x,w)g - K_j(x,w)g, y\rangle_{\mathcal{Y}}| \to 0$, as $i \to \infty$. Therefore, $\exists \bar{K}(\cdot, \cdot) : \mathcal{X} \times \mathcal{X} \to \mathcal{L}(\mathcal{Y})$ such that,

$$\lim_{i \to \infty} \langle K_i(x,w)g, y\rangle_{\mathcal{Y}} = \langle \lim_{i \to \infty} K_i(x,w)g, y\rangle_{\mathcal{Y}} = \langle \bar{K}(x,w)g, y\rangle_{\mathcal{Y}}, \quad \forall x, w \in \mathcal{X}, g, y \in \mathcal{Y}.$$

Hence, $\bar{K}(x,w) = \lim_i K_i(x,w)$ exists for any $x, w \in \mathcal{X}$.

Let $y \in \mathcal{Y}, x \in \mathcal{X}$. Now we show that $\bar{K}$ is non-negative.

$$\{\langle K_j(x,x)y, y\rangle_{\mathcal{Y}}\}_{j \in J} \text{ is an increasing bounded sequence of non-negative numbers.}$$
$$\implies \lim_{j \to \infty} \langle K_j(x,x)g, g\rangle_{\mathcal{Y}} \geq 0$$
$$\implies \langle \lim_{j \to \infty} K_j(x,x)g, g\rangle_{\mathcal{Y}} \geq 0$$
$$\implies \langle \bar{K}(x,x)g, g\rangle_{\mathcal{Y}} \geq 0$$
$$\implies \bar{K} \text{ is non-negative.}$$

Now, since $K_j \in I(K_1, K_2), j \in J$ by our assumption,

$$\langle K_j(x,x)g, g\rangle_{\mathcal{Y}} \le \langle K_1(x,x)g, g\rangle_{\mathcal{Y}}$$
$$\implies \lim_{j \to \infty} \langle K_j(x,x)g, g\rangle_{\mathcal{Y}} \le \langle K_1(x,x)g, g\rangle_{\mathcal{Y}}$$
$$\implies \langle \lim_{j \to \infty} K_j(x,x)g, g\rangle_{\mathcal{Y}} \le \langle K_1(x,x)g, g\rangle_{\mathcal{Y}}$$
$$\implies \langle \bar{K}(x,x)g, g\rangle_{\mathcal{Y}} \le \langle K_1(x,x)g, g\rangle_{\mathcal{Y}}$$

Similarly,

$$\langle \bar{K}(x,x)g, g\rangle_{\mathcal{Y}} \le \langle K_2(x,x)g, g\rangle_{\mathcal{Y}}.$$

Therefore, $\bar{K} \le K_1, K \le K_2$. Hence, $\bar{K}(x,w) = \lim_i K_i(x,w)$ exists for any $x, w \in \mathcal{X}$. $\bar{K}$ is non-negative and is in $I(K_1, K_2)$. $\qquad\square$

**Corollary C.2.1.** Let $K$ be a difference of two non-negative $\mathcal{L}(\mathcal{Y})$-valued kernels on $\mathcal{X}^2$, $K = K_1 - K_2$. Then, without loss of generality, one can choose $K_1$ and $K_2$ with corresponding Hilbert spaces $\mathcal{H}_1$ and $\mathcal{H}_2$, respectively, such that $\mathcal{H}_1 \cap \mathcal{H}_2 = \{0\}$.

*Proof.* By Zorn's lemma, the set $I(K_1, K_2)$ admits a maximum element $K_{\max}$. Based on the proof in Lemma C.2, we can ensure $K_{\max} \le K_1, K_{\max} \le K_2$ i.e., $K_1 - K_{\max}$ and $K_2 - K_{\max}$ are non-negative kernels on $\mathcal{X}^2$. Suppose that $\mathcal{H}_1^{\max}$ and $\mathcal{H}_2^{\max}$ be the corresponding RKHS with respect to $K_1 - K_{\max}$ and $K_2 - K_{\max}$, respectively. Suppose $\mathcal{H}_1^{\max} \cap \mathcal{H}_2^{\max} \ne \{0\}$. By Lemma C.1, the intersection is then an RKHS with a reproducing kernel $K$. Now, let $x \in \mathcal{X}, y \in \mathcal{Y}$, based on the contractive inclusion in Lemma C.1 we obtain

$$\|K(x,.)y\|_{\mathcal{H}_1^{\max}} \le \|K(x,.)y\|_{\mathcal{H}_1^{\max} \cap \mathcal{H}_2^{\max}}$$
$$\implies \|K(x,.)y\|_{\mathcal{H}_1^{\max}} \le \|(K_1 - K_{\max})(x,.)y\|_{\mathcal{H}_1^{\max} \cap \mathcal{H}_2^{\max}} \tag{15}$$
$$\implies \langle K(x,.)y, K(x,.)y\rangle_{\mathcal{H}_1^{\max}} \le \langle (K_1 - K_{\max})(x,.)y, (K_1 - K_{\max})(x,.)y\rangle_{\mathcal{H}_1^{\max} \cap \mathcal{H}_2^{\max}} \tag{16}$$
$$\implies \langle K(x,x)y, y\rangle_{\mathcal{Y}} \le \langle (K_1 - K_{\max})(x,x)y, y\rangle_{\mathcal{Y}} \tag{17}$$

The inequality in (15) can be obtained from the first inequality, since any function $K(x,.)y$ in $\mathcal{H}_1^{\max} \cap \mathcal{H}_2^{\max}$ is also a member of $\mathcal{H}_1^{\max}$ and hence can be equivalently represented as $(K_1 - K_{\max})(x,.)y$. We obtain inequality in (17) from inequality (16) using reproducing property of $\mathcal{L}(\mathcal{Y})$-valued kernels $K_1 - K_{\max}$ and $K$. From inequality (17), we can deduce that $K \le K_1 - K_{\max}$. Similarly, we can argue that $K \le K_2 - K_{\max}$. As $\mathcal{H}_1^{\max} \cap \mathcal{H}_2^{\max} \ne \{0\}$, $K$ is a non-zero reproducing kernel which contradicts the maximality of $K_{\max}$. This leads to $\bar{K}_1 = K_1 - K_{\max}$ and $\bar{K}_2 = K_2 - K_{\max}$ having corresponding Hilbert spaces $\mathcal{H}_1$ and $\mathcal{H}_2$, respectively, satisfying $\mathcal{H}_1 \cap \mathcal{H}_2 = \{0\}$. $\qquad\square$

The following theorem provides a characterization between a generalized $\mathcal{L}(\mathcal{Y})$-valued kernel on $\mathcal{X}^2$ and an associated function-valued RKKS.

**Theorem C.3.** Let $\breve{K}$ be a $\mathcal{L}(\mathcal{Y})$-valued kernel on $\mathcal{X}^2$. Then there is an associated reproducing kernel Krein space if and only if $\breve{K}$ is a generalized $\mathcal{L}(\mathcal{Y})$-valued kernel, that is, $\breve{K} = K_1 - K_2$, where $K_1$ and $K_2$ are non-negative $\mathcal{L}(\mathcal{Y})$-valued kernels on $\mathcal{X}^2$

*Proof.* Suppose that $\breve{K}$ is the reproducing kernel of some RKKS $(\mathcal{K}, [.,.])$ and let $\mathcal{K} = \mathcal{K}_1 \oplus \mathcal{K}_2$ be a decomposition of $\mathcal{K}$, where $(\mathcal{K}_1, \langle.,.\rangle_1)$ and $(\mathcal{K}_2, \langle.,.\rangle_2)$ are orthogonal Hilbert subspaces. Let $P_1$ (respectively, $P_2$) be the orthogonal projection from $\mathcal{K}$ onto $\mathcal{K}_1$ (respectively, $\mathcal{K}_2$). Using reproducibility property in Definition 2.3, we get

$$\langle \breve{K}(z,w)g, y\rangle_{\mathcal{Y}} = [\breve{K}(z,.)g, \breve{K}(w,.)y]$$
$$= \langle P_1 \breve{K}(z,.)g, P_1 \breve{K}(w,.)y\rangle_1 - \langle P_2 \breve{K}(z,.)g, P_2 \breve{K}(w,.)y\rangle_2$$

which exhibits $\breve{K}$ as a difference of two positive functions, that is, $\breve{K}$ is a generalized $\mathcal{L}(\mathcal{Y})$-valued kernel.

Conversely, by definition, a generalized $\mathcal{L}(\mathcal{Y})$-valued kernel $\breve{K}$ is associated with two non-negative operator-valued kernels $K_1, K_2$ such that $\breve{K} = K_1 - K_2$. Using Corollary (C.2.1), we can obtain

Hilbert spaces $\mathcal{H}_1, \mathcal{H}_2$ corresponding to $K_1, K_2$ respectively such that $\mathcal{H}_1 \cap \mathcal{H}_2 = \{0\}$. Then the space

$$\mathcal{K} = \{F = F_1 + F_2, \ \ F_1 \in \mathcal{H}_1, F_2 \in \mathcal{H}_2\}$$

with the inner product

$$\langle F, F \rangle_\mathcal{K} = \langle F_1, F_1 \rangle_{\mathcal{H}_1} + \langle F_2, F_2 \rangle_{\mathcal{H}_2}$$

is a Hilbert space. Moreover, the map $\sigma$ defined by

$$\sigma F = F_1 - F_2$$

is self-adjoint and unitary from $\mathcal{K}$ to $\mathcal{K}$, $K(x, .)y$ belongs to $\mathcal{K}$ for any $x \in \mathcal{X}$ and $y \in \mathcal{Y}$, with

$$[F, F] = \langle F, \sigma F \rangle_\mathcal{K},$$

whence we obtain

$$
\begin{aligned}
[F, \check{K}(x, .)y] &= \langle F_1, K_1(x, .)y \rangle_{\mathcal{H}_1} + \langle F_2, K_2(x, .)y \rangle_{\mathcal{H}_2} \\
&= \langle F_1(x), y \rangle_\mathcal{Y} + \langle F_2(x), y \rangle_\mathcal{Y} \\
&= \langle F_1(x) + F_2(x), y \rangle_\mathcal{Y} \\
&= \langle F(x), y \rangle_\mathcal{Y}.
\end{aligned}
$$

Therefore, $(\mathcal{K}, [., .])$ is a reproducing kernel Krein space with the reproducing kernel $\check{K}$, where $\mathcal{K} = \mathcal{H}_1 \oplus \mathcal{H}_2$. $\qquad\square$

Theorem C.3 ensures that for a generalized operator-valued kernel there exists an associated function-valued RKKS, which is a compromise on the bijection that exists between positive definite operator-valued kernels and associated function-valued RKHS (Kadri et al., 2016).

## D  Example (Eq. 3) in Section 2 revisited

Recall the generalized operator-valued kernel in Eq. (3):

$$(\check{K}(x_i, x_j)y)(t) = g(x_i, x_j) \int_{\Omega_y} h(s, t)y(s)ds, \qquad (18)$$

where, $\Omega_x = \Omega_y = [0, 1], \mathcal{X} = L^2(\Omega_x), \mathcal{Y} = L^2(\Omega_y), g$ is a scalar-valued kernel on $\mathcal{X}^2$ and $h$ is an output kernel on $(\Omega_y)^2$, and either $g$ or $h$ is indefinite. We illustrate here that the indefinite operator-valued kernel constructed in Eq. (18) satisfies the properties in Definition (2.3).

From the definition of $\check{K} = K_1 - K_2$, where $K_1, K_2$ are defined as

$$(K_1(x_i, x_j)y)(t) = g_1(x_i, x_j) \int_{\Omega_y} h_1(s, t)y(s)ds$$

$$(K_2(x_i, x_j)y)(t) = g_2(x_i, x_j) \int_{\Omega_y} h_2(s, t)y(s)ds$$

with $x_i, x_j \in \mathcal{X}, y \in \mathcal{Y}, g_1, g_2$ are scalar-valued positive kernels on $\mathcal{X}^2, h_1, h_2$ are scalar-valued kernels on $(\Omega_y)^2$.

For the Krein space $\mathcal{K}$ of functions from $\mathcal{X}$ to $\mathcal{Y}$, we can obtain $\mathcal{K} = \mathcal{H}_1 \oplus \mathcal{H}_2$, where $\mathcal{H}_1$ and $\mathcal{H}_2$ are function-valued RKHS for operator-valued kernels $K_1$ and $K_2$ respectively. Now we have

$$
\begin{aligned}
\langle F, \check{K}(w, .)y \rangle_{\mathcal{K}} &= \langle F_1, K_1(w, .)y \rangle_{\mathcal{H}_1} - \langle F_2, -K_2(w, .)y \rangle_{\mathcal{H}_2} \\
&= \langle F_1, K_1(w, .)y \rangle_{\mathcal{H}_1} + \langle F_2, K_2(w, .)y \rangle_{\mathcal{H}_2} \\
&= \left\langle F_1, g_1(w, .) \int_{\Omega_y} h_1(s, t)y(s)ds \right\rangle_{\mathcal{F}} + \left\langle F_2, g_2(w, .) \int_{\Omega_y} h_2(s, t)y(s)ds \right\rangle_{\mathcal{F}} \\
&= \int_{\Omega_y} \int_{\Omega_x} [F_1(z)](t) \left[ g_1(w, z) \int_{\Omega_y} h_1(s, t)y(s)ds \right] dzdt + \\
&\qquad \int_{\Omega_y} \int_{\Omega_x} [F_2(z)](t) \left[ g_2(w, z) \int_{\Omega_y} h_2(s, t)y(s)ds \right] dzdt \qquad\qquad (19) \\
&= \int_{\Omega_y} \int_{\Omega_x} [g_1(w, z)F_1(z)](t) \left[ \int_{\Omega_y} h_1(s, t)y(s)ds \right] dzdt + \\
&\qquad \int_{\Omega_y} \int_{\Omega_x} [g_2(w, z)F_2(z)](t) \left[ \int_{\Omega_y} h_2(s, t)y(s)ds \right] dzdt \qquad\qquad (20) \\
&= \int_{\Omega_y} [F_1(w)](t)y(t)dt + \int_{\Omega_y} [F_2(w)](t)y(t)dt \qquad\qquad (21) \\
&= \langle F_1(w), y \rangle_{\mathcal{Y}} + \langle F_2(w), y \rangle_{\mathcal{Y}} \\
&= \langle F(w), y \rangle_{\mathcal{Y}}.
\end{aligned}
$$

Equations (19), (20) and (21) are a result of the reproducibility property of scalar-valued kernels $g_1, g_2, h_1$ and $h_2$.

## E  Proof of Representer Theorem

In this section, we provide a proof for the Representer theorem stated in Section 3. We recall the result here. In the proof we use the Gateaux derivative in an associated function-valued reproducing kernel Krein space for a generalized operator-valued kernel, which is an extension of the Gateaux derivative in a Hilbert space.

**Theorem E.1 (Representer theorem).** Let $\check{K}$ be an indefinite operator-valued kernel and $\mathcal{K}(= \mathcal{K}_1 \oplus \mathcal{K}_2)$ be its corresponding function-valued reproducing kernel Krein space. The solution $\widetilde{F}_\lambda \in \mathcal{K}$ of the regularized optimization problem.

$$
\widetilde{F}_\lambda = \arg \underset{F \in \mathcal{K}}{\text{stabilize}} \sum_{i=1}^{n} \|y_i - F(x_i)\|_{\mathcal{Y}}^2 + \lambda \langle F, F \rangle_{\mathcal{K}}, \qquad\qquad (22)
$$

where $\lambda > 0, F(= F_1 + F_2) \in \mathcal{K}$, has the following form

$$
\widetilde{F}_\lambda(.) = \sum_{i=1}^{n} \check{K}(x_i, .)u_i, \text{ where } u_i \in \mathcal{Y}. \qquad\qquad (23)
$$

*Proof.* We use the Gateaux derivative to obtain the condition for stationary points which stabilize the functional $J_\lambda(F)$, given by

$$
J_\lambda(F) = \sum_{i=1}^{n} \|y_i - F(x_i)\|_{\mathcal{Y}}^2 + \lambda \langle F, F \rangle_{\mathcal{K}}, \quad \forall F \in \mathcal{K}.
$$

In order to find the critical points in $\mathcal{K}$, we use Gateaux derivative $D_G$ of $J_\lambda$ with respect to $F$ in the direction $H$, which is defined by

$$
D_G J_\lambda(F, H) = \lim_{\tau \to 0} \frac{J_\lambda(F + \tau H) - J_\lambda(F)}{\tau}.
$$

Let $\widetilde{F}$ be the operator in $\mathcal{K}$ such that

$$\widetilde{F} = \arg \operatorname*{stabilize}_{F \in \mathcal{K}} J_\lambda(F) \implies D_G J_\lambda(F, H) = 0, \ \ \forall H \in \mathcal{K}.$$

$J_\lambda$ can be written as

$$J_\lambda(F) = \sum_{i=1}^{n} G_i(F) + \lambda L(F)$$

and as $D_G J_\lambda(F, H) = \langle D_G J_\lambda(F), H \rangle_\mathcal{K}, \ \forall F, H \in \mathcal{K}$, we obtain the following.

1. $L(F) = \langle F, F \rangle_\mathcal{K}$. Therefore we have

$$\lim_{\tau \to 0} \frac{\langle F + \tau H, F + \tau H \rangle_\mathcal{K} - \langle F, F \rangle_\mathcal{K}}{\tau} = 2 \langle F, H \rangle_\mathcal{K}$$

$$\implies D_G L(F) = 2F.$$

2. $G_i(F) = \|y_i - F(x_i)\|_\mathcal{Y}^2$. Then we have

$$\lim_{\tau \to 0} \frac{\|y_i - F(x_i) - \tau H(x_i)\|_\mathcal{Y}^2 - \|y_i - F(x_i)\|_\mathcal{Y}^2}{\tau} = -2 \langle y_i - F(x_i), H(x_i) \rangle_\mathcal{Y} \quad (24)$$

$$= -2 \langle \check{K}(x_i, .)(y_i - F(x_i)), H \rangle_\mathcal{K} \quad (25)$$

$$= -2 \langle \check{K}(x_i, .)u_i, H \rangle_\mathcal{K}, \quad (26)$$

$$\implies D_G G_i(F) = -2 \check{K}(x_i, .)u_i.$$

We obtain Eq. (25) from Eq. (24) using the reproducibility property in Definition 2.3. In Eq. (25), we use $u_i = y_i - F(x_i)$ to get Eq. (26). Using 1, 2, and $D_G J_\lambda(\widetilde{F}) = 0$, we obtain, $\widetilde{F}(.) = \frac{1}{\lambda} \sum_{i=1}^{n} \check{K}(x_i, .)u_i$. The constant $\frac{1}{\lambda}$ can be absorbed in functions $u_i$'s, such that $\widetilde{F}(.) = \sum_{i=1}^{n} \check{K}(x_i, .)u_i$. $\qquad \square$

## F   Condition for Stationary Points of Problem (4)

We obtain a condition for stationary points of the optimization problem in Equation (4).

Using the representer theorem, the problem (4) can be equivalently formulated as the following problem:

$$\tilde{\mathbf{u}}_\lambda = \arg \operatorname*{stabilize}_{\mathbf{u} \in \mathcal{Y}^n} \sum_{i=1}^{n} \left\| y_i - \sum_{j=1}^{n} \check{K}(x_i, x_j)u_j \right\|_\mathcal{Y}^2 + \lambda \left\langle \sum_{i=1}^{n} \check{K}(x_i, .)u_i, \sum_{j=1}^{n} \check{K}(x_j, .)u_j \right\rangle_\mathcal{K}. \quad (27)$$

We have the following simplification of the term $\left\langle \sum_{i=1}^{n} \check{K}(x_i, .)u_i, \sum_{j=1}^{n} \check{K}(x_j, .)u_j \right\rangle_\mathcal{K}$ in problem (27). We have

$$\left\langle \sum_{i=1}^{n} \check{K}(x_i, .)u_i, \sum_{j=1}^{n} \check{K}(x_j, .)u_j \right\rangle_\mathcal{K} = \sum_{i=1}^{n} \left\langle \check{K}(x_i, .)u_i, \sum_{j=1}^{n} \check{K}(x_j, .)u_j \right\rangle_\mathcal{K} \quad (28)$$

$$= \sum_{i=1}^{n} \sum_{j=1}^{n} \left\langle \check{K}(x_i, .)u_i, \check{K}(x_j, .)u_j \right\rangle_\mathcal{K} \quad (29)$$

$$= \sum_{i=1}^{n} \sum_{j=1}^{n} \left\langle \check{K}(x_i, x_j)u_i, u_j \right\rangle_\mathcal{Y}. \quad (30)$$

Note that Eq. (28) and Eq. (29) follow from the property of bilinear forms and Eq. (30) follows from the reproducing property of $\check{K}$. Thus we have the following simplified formulation:

$$\tilde{\mathbf{u}}_\lambda = \arg \operatorname*{stabilize}_{\mathbf{u} \in \mathcal{Y}^n} \sum_{i=1}^{n} \left\| y_i - \sum_{j=1}^{n} \check{K}(x_i, x_j)u_j \right\|_\mathcal{Y}^2 + \lambda \sum_{i=1, j=1}^{n} \langle \check{K}(x_i, x_j)u_i, u_j \rangle_\mathcal{Y},$$

To solve this problem, we use the directional derivative of the objective function $J_\lambda(\mathbf{u})$, given by

$$J_\lambda(\mathbf{u}) = \sum_{i=1}^{n} \left\| y_i - \sum_{j=1}^{n} \breve{K}(x_i, x_j) u_j \right\|_{\mathcal{Y}}^2 + \lambda \sum_{i=1, j=1}^{n} \langle \breve{K}(x_i, x_j) u_i, u_j \rangle_{\mathcal{Y}}, \quad \mathbf{u} \in \mathcal{Y}^n.$$

Letting $J_\lambda(\mathbf{u}) = \sum_{i=1}^{n} G_i(\mathbf{u}) + \lambda L(\mathbf{u})$, we can find the directional derivative of $J_\lambda(\mathbf{u})$ with respect to the direction $\mathbf{v}$ as $D_{\mathbf{v}} J_\lambda(\mathbf{u})$.

$$D_{\mathbf{v}} G_i(\mathbf{u}) = \lim_{\tau \to 0} \frac{G_i(u + \tau v) - G_i(u)}{\tau}$$

$$= -2 \left\langle y_i - \sum_{j=1}^{n} \breve{K}(x_i, x_j) u_j, \sum_{j=1}^{n} \breve{K}(x_i, x_j) v_j \right\rangle.$$

$$D_{\mathbf{v}} L(\mathbf{u}) = \lim_{\tau \to 0} \frac{L(u + \tau v) - L(u)}{\tau}$$

$$= \lambda \sum_{i,j}^{n} \langle \breve{K}(x_i, x_j) u_i, v_j \rangle + \lambda \sum_{i,j}^{n} \langle \breve{K}(x_i, x_j) v_i, u_j \rangle.$$

As $\breve{K}$ is Hermitian from the definition of operator-valued kernel, we obtain

$$\langle \breve{K}(x_i, x_j) u_i, v_j \rangle = \langle u_i, \breve{K}(x_i, x_j) v_j \rangle, \quad \forall i, j = 1, \dots, n. \tag{31}$$

Therefore,

$$D_{\mathbf{v}} L(\mathbf{u}) = \lambda \sum_{i,j}^{n} \langle \breve{K}(x_i, x_j) u_i, v_j \rangle + \lambda \sum_{i,j}^{n} \langle \breve{K}(x_i, x_j) v_i, u_j \rangle$$

$$= \lambda \sum_{i,j}^{n} \langle u_i, \breve{K}(x_i, x_j) v_j \rangle + \lambda \sum_{i,j}^{n} \langle \breve{K}(x_i, x_j) v_i, u_j \rangle \tag{32}$$

$$= \lambda \sum_{i,j}^{n} \langle u_i, \breve{K}(x_i, x_j) v_j \rangle + \lambda \sum_{i,j}^{n} \langle u_j, \breve{K}(x_j, x_i) v_i \rangle \tag{33}$$

$$= 2\lambda \sum_{i,j}^{n} \langle u_i, \breve{K}(x_i, x_j) v_j \rangle \tag{34}$$

Eq. (32) follows from Eq. (31) and in Eq. (32), we use symmetry of $\langle \cdot, \cdot \rangle$ to obtain Eq. (34). In order to stabilize $J_\lambda(\mathbf{u})$, its directional derivative $D_{\mathbf{v}} J_\lambda(\mathbf{u}) = 0$, $\forall v \in \mathcal{Y}^n$.

$$D_{\mathbf{v}} J_\lambda(\mathbf{u}) = 0$$

$$\implies \sum_{i=1}^{n} D_{\mathbf{v}} G_i(\mathbf{u}) + \lambda D_{\mathbf{v}} L(\mathbf{u}) = 0$$

$$\implies -2 \sum_{i=1}^{n} \left\langle y_i - \sum_{j=1}^{n} \breve{K}(x_i, x_j) u_j, \sum_{j=1}^{n} \breve{K}(x_i, x_j) v_j \right\rangle + 2\lambda \sum_{i,j}^{n} \langle u_i, K(x_i, x_j) v_j \rangle = 0$$

$$\implies \sum_{i=1}^{n} \left\langle \sum_{j=1}^{n} \breve{K}(x_i, x_j) u_j - y_i, \sum_{j=1}^{n} \breve{K}(x_i, x_j) v_j \right\rangle + \sum_{i,j}^{n} \langle \lambda u_i, K(x_i, x_j) v_j \rangle = 0$$

$$\implies \sum_{i=1}^{n} \left\langle \sum_{j=1}^{n} \breve{K}(x_i, x_j) u_j - y_i, \sum_{j=1}^{n} \breve{K}(x_i, x_j) v_j \right\rangle + \sum_{i=1}^{n} \left\langle \lambda u_i, \sum_{j=1}^{n} \breve{K}(x_i, x_j) v_j \right\rangle = 0$$

$$\implies \sum_{i=1}^{n} \left\langle \sum_{j=1}^{n} \breve{K}(x_i, x_j) u_j - y_i + \lambda u_i, \sum_{j=1}^{n} \breve{K}(x_i, x_j) v_j \right\rangle = 0, \forall v \in \mathcal{Y}^n.$$

The above condition can be reduced to

$$(\breve{\mathbf{K}} + \lambda I) \mathbf{u} = \mathbf{y}, \tag{35}$$

where $\check{\mathbf{K}}$ is a matrix of operators formed by using $\check{K}$. For the example considered in Appendix D), we have

$$
\check{\mathbf{K}} = \begin{bmatrix} K_1(x_1,x_1) - K_2(x_1,x_1) & \ldots & K_1(x_1,x_n) - K_2(x_1,x_n) \\ \vdots & \ddots & \vdots \\ K_1(x_n,x_1) - K_2(x_n,x_1) & \ldots & K_1(x_n,x_n) - K_2(x_n,x_n) \end{bmatrix}
$$

$$
= \begin{bmatrix} g_1(x_1,x_1)T_1 - g_2(x_1,x_1)T_2 & \ldots & g_1(x_1,x_n)T_1 - g_2(x_1,x_n)T_2 \\ \vdots & \ddots & \vdots \\ g_1(x_n,x_1)T_1 - g_2(x_n,x_1)T_2 & \ldots & g_1(x_n,x_n)T_1 - g_2(x_n,x_n)T_2 \end{bmatrix}.
$$

Note that in Eq. (35), $\mathbf{y}$ is a column vector of output functions corresponding to the inputs $x_i$'s, for $i = 1, 2, \ldots, n$. The $\mathbf{u}$ computed from Eq. (35) consists of a column vector of operators in $\mathcal{L}(\mathcal{Y})$ which act as basis functions for predictions made for an unseen example.

Equation (35) provides a sufficient condition for obtaining the stationary points of the stabilization problem 27.

## G  Krylov Subspace Methods

There are a number of Krylov subspace methods for solving system of a linear system of equations. For solving a linear system

$$
Ax = b, A \in \mathbb{R}^{n \times n}, A^\top = A, x, b \in \mathbb{R}^n, \tag{36}
$$

a Krylov subspace method is based on iteratively computing an approximation of the solution $x$. Consider the $m$-th Krylov subspace,

$$
\mathcal{K}_m(A, r_0) = span\{r_0, Ar_0, \ldots, A^m r_0\}, \text{ where } r_0 = b - Ax_0,
$$

and $x_0$ is an initial approximation (or guess) of $x$. The solution $x$ of $Ax = b$ is obtained in $\mathcal{K}_m(A, r_0)$ for $m \leq n$, without explicitly computing $A^{-1}$.

A popular variant is the Minimal residual method (MINRES), first proposed in (Paige and Saunders, 1975). MINRES algorithm is based on solving for $x$ in Eq. (36), with a symmetric (Hermitian) matrix (possibly indefinite) $A$ by minimizing the norm residual $\|r_i\| = \|b - Ax_i\|, x_i \in \mathcal{K}_i(A, b)$ in the $i$-th iteration ($\|.\|$ is the 2-norm). MINRES is based on tridiagonalization using orthonormal vectors obtained from Lanczos algorithm (Lanczos, 1950). A detailed account of MINRES and other Krylov subspace methods can be found in (Barrett et al., 1994) and (Choi, 2006).

## H  Details of OpMINRES Algorithm

In order to solve for $\mathbf{u}$ in Eq. (35), we use an operator based Krylov subspace method, inspired by a similar construction in (Ong et al., 2004). As the matrix of operators $(\check{\mathbf{K}} + \lambda I)$ in Eq. (35) is symmetric and possibly indefinite, we based our algorithm on the minimal residual method (MINRES). The proposed OpMINRES is designed for a matrix of operators acting on a column of functions from $\mathcal{L}(\mathcal{Y})$. We illustrate the algorithm by solving for $\mathbf{u}$ in $\mathbf{Au} = \mathbf{y}$, with $\mathbf{A} = (\check{\mathbf{K}} + \lambda I)$.

### H.1  OpLanczos Step

The Lanczos method used in MINRES helps to tridiagonalize $A$ in Eq. (36). Similarly, OpLanczos in OpMINRES is used to trigiagonalize the operator matrix $\mathbf{A}$. The vectors obtained from OpLanczos form an orthonormal set. Using the OpLanczosStep Algorithm 1, we can obtain,

$$
\mathbf{A}V_k = V_k T_k, \quad \text{where } T_k = \begin{bmatrix} \alpha_1 & \beta_2 & & & & & 0 \\ \beta_2 & \alpha_2 & \beta_3 & & & & \\ & \beta_3 & \alpha_3 & \ddots & & & \\ & & \ddots & \ddots & \beta_{k-2} & & \\ & & & \beta_{k-1} & \alpha_{k-1} & \beta_k \\ 0 & & & & \beta_k & \alpha_k \end{bmatrix},
$$

**Algorithm 1 OpLanczosStep**$(A, v_k, v_{k-1}, \beta_k)$

> **Input:** $A, v_k, v_{k-1}, \beta_k$
> **Output:** $\alpha_k, \beta_{k+1}, v_{k+1}$
> $\bar{v}_{k+1} = A v_k - \beta_k v_{k-1}$
> $\alpha_k = \langle \bar{v}_{k+1}, q_k \rangle_{\mathcal{Y}^n}$
> $\bar{v}_{k+1} \leftarrow \bar{v}_{k+1} - \alpha_k v_k$
> $\beta_{k+1} = \|\bar{v}_{k+1}\|_{\mathcal{Y}^n}$
> $v_{k+1} = \frac{1}{\beta_{k+1}} \bar{v}_{k+1}$

and $V_k = [v_1 \, v_2 \, \ldots \, v_k]$, where $v_i$'s are obtained using OpLanczosStep Algorithm. The columns of $V_k$ belonging to $\mathcal{Y}^n$ are orthonormal and the following equation is satisfied

$$\mathbf{A} V_k = V_{k+1} \overline{T}_k, \quad \text{where } \overline{T}_k = \begin{bmatrix} \alpha_1 & \beta_2 & & & & & 0 \\ \beta_2 & \alpha_2 & \beta_3 & & & & \\ & \beta_3 & \alpha_3 & \ddots & & & \\ & & \ddots & \ddots & \beta_{k-2} & & \\ & & & \beta_{k-1} & \alpha_{k-1} & \beta_k & \\ & & & & \beta_k & \alpha_k & \\ 0 & & & & & \beta_{k+1} \end{bmatrix}.$$

We intend to solve $\mathbf{A}\mathbf{u} = \mathbf{y}$ by obtaining a solution in the Krylov space $\mathcal{K}_k(\mathbf{A}, \mathbf{y}) = \text{span}\{\mathbf{y}, \mathbf{A}\mathbf{y}, \mathbf{A}^2\mathbf{y}, \ldots, \mathbf{A}^{k-1}\mathbf{y}\}$. For each iteration $k$, we obtain the following equations using the transformation $\mathbf{x} = V_k x$, where $\mathbf{x} \in \mathcal{Y}^n, x \in \mathbb{R}^k$.

$$\min_{\mathbf{x} \in \mathcal{K}_k(\mathbf{A}, \mathbf{y})} \|\mathbf{y} - \mathbf{A}\mathbf{x}\|_{\mathcal{Y}^n} = \min_{x \in \mathbb{R}^k} \|\mathbf{y} - \mathbf{A} V_k x\|_{\mathcal{Y}^n} = \min_{x \in \mathbb{R}^k} \|\mathbf{y} - V_{k+1}\overline{T}_k x\|_{\mathcal{Y}^n}$$

$$= \min_{x \in \mathbb{R}^k} \|V_{k+1}(\beta_1 e_1 - \overline{T}_k x)\|_{\mathcal{Y}^n}, \tag{37}$$

$$(\text{where } \beta_1 = \|\mathbf{y}\|_{\mathcal{Y}^n}, e_1 = [1 \, 0 \, \ldots \, 0]^\top \text{ and } v_1 = \mathbf{y}/\|\mathbf{y}\|_{\mathcal{Y}^n})$$

$$= \min_{x \in \mathbb{R}^k} \|\beta_1 e_1 - \overline{T}_k x\|_2. \tag{38}$$

The change in norms $\|.\|_{\mathcal{Y}^n}$ in (37) to $\|.\|_2$ is obtained based on the following arguments. Let $z = [z_1, z_2, \ldots, z_{k+1}]^\top \in \mathbb{R}^{k+1}$ and $V_{k+1} = [v_1 \, v_2 \, \ldots v_{k+1}]$, where $v_i \in \mathcal{Y}^n$, for $i = 1, 2, \ldots, k+1$, then we have

$$\|V_{k+1} z_{k+1}\|_{\mathcal{Y}^n} = \|z_1 v_1 + z_2 v_2 + \cdots + z_{k+1} v_{k+1}\|_{\mathcal{Y}^n}$$

$$= \sqrt{z_1^2 \int_{\Omega_y} v_1^2(t) dt + z_2^2 \int_{\Omega_y} v_2^2(t) dt + \cdots + z_{k+1}^2 \int_{\Omega_y} v_{k+1}^2(t) dt} \tag{39}$$

$$= \sqrt{z_1^2 + z_2^2 + \cdots + z_{k+1}^2} \tag{40}$$

$$= \|z\|_2$$

Equation (39) reduces to (40) as the $v_i$'s are orthonormal in $\mathcal{Y}^n$. Solving for $x_k = \arg\min_{x \in \mathbb{R}^k} \|\beta_1 e_1 - \overline{T}_k x\|_2$ can be done using QR decomposition (Choi, 2006) which has been discussed in the next section. Now, the transformation from $\mathbb{R}^k$ back to $\mathcal{Y}^n$ to obtain $\mathbf{u}^k$ is achieved using the following:

$$\mathbf{u}^k = V_k x_k = V_k \left( \arg\min_{x \in \mathbb{R}^k} \|\beta_1 e_1 - \overline{T}_k x\|_2 \right).$$

**Algorithm 2 SymOrtho**$(a, b)$

---

   **Input:** $a, b$
   **Output:** $c, s, r$
   **if** $b == 0$ **then**
      $s = 0$
      $r = |a|$
      **if** $a == 0$ **then**
         $c = 1$
      **else**
         $c = \text{sgn}(a)$
      **end if**
   **else if** $a == 0$ **then**
      $c = 0$
      $s = \text{sgn}(b)$
      $r = |b|$
   **else if** $|b| > |a|$ **then**
      $\tau = a/b$
      $s = \text{sgn}(b)/\sqrt{1 + \tau^2}$
      $c = s\tau$
      $r = b/s$
   **else if** $|a| > |b|$ **then**
      $\tau = b/a$
      $c = \text{sgn}(a)/\sqrt{1 + \tau^2}$
      $s = c\tau$
      $r = a/c$
   **end if**

---

### H.1.1 QR Decomposition

In order to apply QR decomposition on symmetric $\overline{T}_k$, we use Givens rotation $Q_k$ to obtain a upper-triangular system.

$$
Q_k \overline{T}_k = \begin{bmatrix} R_k \\ 0 \end{bmatrix} = \begin{bmatrix} \gamma_1^{(1)} & \delta_2^{(1)} & \epsilon_3^{(1)} & & & & 0 \\ & \gamma_2^{(2)} & \delta_3^{(2)} & \epsilon_4^{(1)} & & & \\ & & \ddots & \ddots & \ddots & & \\ & & & \gamma_{k-2}^{(2)} & \delta_{k-1}^{(2)} & \epsilon_k^{(1)} & \\ & & & & \gamma_{k-1}^{(2)} & \delta_k^{(2)} & \\ & & & & & \gamma_k^{(2)} & \\ 0 & & & & & & 0 \end{bmatrix}, \qquad Q_k(\beta_1 e_1) = \begin{bmatrix} t_k \\ \phi_k \end{bmatrix},
$$

where $Q_k = Q_{k,k+1} \ldots Q_{2,3} Q_{1,2}, Q_{i,i+1}$ are Givens rotations created to annihilate the $\beta_i$'s in sub-diagonal of $\overline{T}_k$. The $Q_{i,i+1}$'s involved in the product to obtain $Q_k$ are given by,

$$
Q_{i,i+1} = \begin{bmatrix} I_{i-1} & & & \\ & c_i & s_i & \\ & s_i & -c_i & \\ & & & I_{k-i} \end{bmatrix}.
$$

The matrices $Q_{i,i+1}$ are obtained using the SymOrtho Algorithm 2. The sub-problem can be rewritten with $x_k = \arg\min_{x \in \mathbb{R}^k} \|\beta_1 e_1 - \overline{T}_k x\|_2$ as

$$
x_k = \arg\min_{x \in \mathbb{R}^k} \left\| \begin{bmatrix} t_k \\ \phi_k \end{bmatrix} - \begin{bmatrix} R_k \\ 0 \end{bmatrix} x \right\|_2, \text{ where } t_k = [\tau_1 \ \ \tau_2 \ \ldots \ \tau_k]^\top \text{ and}
$$

**Algorithm 3 OpMINRES**$(A, b, maxiter)$

**Input:** $A, b, maxiter$
**Output:** $x, \phi, \psi, \chi$
$\beta_1 = \|b\|_{\mathcal{Y}^n}$
$v_0 = 0$
$v_1 = \frac{1}{\beta_1} b$
$\phi_0 = \tau_0 = \beta_1$
$\chi_0 = 0$
$\delta_1^{(1)} = 0$
$c_0 = -1$
$s_0 = 0$
$d_0 = d_{-1} = x_0 = 0$
$k = 1$
**while** stopping criteria not satisfied **do**
    **OpLanczosStep**$(A, v_k, v_{k-1}, \beta_k) \to \alpha_k, \beta_{k+1}, v_{k+1}$
    //last left orthogonalization on middle two entries in last column of $T_{k+1,k}$
    $\delta_k^{(2)} = c_{k-1}\delta_k^{(1)} + s_{k-1}\alpha_k$
    $\gamma_k^{(1)} = s_{k-1}\delta_k^{(1)} - c_{k-1}\alpha_k$
    //last left orthogonalization to produce first two entries of $T_{k+2,k+1}e_{k+1}$
    $\epsilon_{k+1}^{(1)} = s_{k-1}\beta_{k+1}$
    $\delta_{k+1}^{(1)} = -c_{k-1}\beta_{k+1}$
    //current left orthogonalization to zero out $\beta_{k+1}$
    **SymOrtho**$(\gamma_k^{(1)}, \beta_{k+1}) \to c_k, s_k, \gamma_k^{(2)}$
    //right-hand side, residual norms
    $\tau_k = c_k\phi_{k-1}$
    $\phi_k = s_k\phi_{k-1}$
    $\psi_{k-1} = \phi_{k-1}\sqrt{(\gamma_k^{(1)})^2 + (\delta_{k+1}^{(1)})^2}$
    //update solution
    $d_k = \frac{1}{\gamma_k^{(2)}}\left(v_k - \delta_k^{(2)}d_{k-1} - \epsilon_k^{(1)}d_{k-2}\right)$
    $x_k = x_{k-1} + \tau_k d_k$
    $\chi_k = \|x_k\|_{\mathcal{Y}^n}$
    $k \leftarrow k + 1$
**end while**
$x = x_k, \phi = \phi_k, \psi = \phi_k\sqrt{(\gamma_{k+1}^{(1)})^2 + (\delta_{k+2}^{(1)})^2}, \chi = \chi_k$

$$
\begin{bmatrix} t_k \\ \phi_k \end{bmatrix} = \beta_1 Q_{k,k+1}\ldots Q_{2,3}\begin{bmatrix} c_1 \\ s_1 \\ 0_{k-1} \end{bmatrix} = \beta_1 Q_{k,k+1}\ldots Q_{3,4}\begin{bmatrix} c_1 \\ s_1 c_2 \\ s_1 s_2 \\ 0_{k-2} \end{bmatrix} = \beta_1 \begin{bmatrix} c_1 \\ s_1 c_2 \\ \vdots \\ s_1\ldots s_{k-1}c_k \\ s_1\ldots s_{k-1}s_k \end{bmatrix}.
$$

A shorthand way to represent the action of $Q_{k,k+1}$ can be described as

$$
\begin{bmatrix} c_k & s_k \\ s_k & -c_k \end{bmatrix}\left[\begin{array}{ccc|c} \gamma_k^{(1)} & \delta_{k+1}^{(1)} & 0 & \phi_{k-1} \\ \beta_{k+1} & \alpha_{k+1} & \beta_{k+2} & 0 \end{array}\right] = \left[\begin{array}{ccc|c} \gamma_k^{(2)} & \delta_{k+1}^{(2)} & \epsilon_{k+2}^{(1)} & \tau_k \\ 0 & \gamma_{k+1}^{(1)} & \delta_{k+2}^{(1)} & \phi_k \end{array}\right].
$$

OpMINRES computes $\mathbf{u}^k$ in $\mathcal{K}_k(\mathbf{A}, \mathbf{y})$ as an approximate solution to the problem $\mathbf{A}\mathbf{u} = \mathbf{y}$:

$$
\mathbf{u}^k = V_k x_k = V_k R_k^{-1} t_k = D_k \begin{bmatrix} t_{k-1} \\ \tau_k \end{bmatrix} = [D_{k-1} \quad d_k]\begin{bmatrix} t_{k-1} \\ \tau_k \end{bmatrix}
$$
$$
= \mathbf{u}^{k-1} + \tau_k d_k.
$$

The relation satisfied by $d_k$ is given by,

$$d_k = \frac{1}{\gamma_k^{(2)}} \left( v_k - \delta_k^{(2)} d_{k-1} - \epsilon_k^{(1)} d_{k-2} \right).$$

The details are provided in OpMINRES Algorithm 3. As OpMINRES Algorithm 3 is based on reducing the problem in Eq. (35) from an infinite-dimensional optimization problem to a finite-dimensional problem in Eq. (38), the convergence of OpMINRES follows from the convergence of MINRES (Choi, 2006). The construction of OpMINRES ensures the monotonicity of the residual norms. The stopping criteria for OpMINRES could be based on the value of relative residual norms $\phi_k/\phi_0$. Traditionally, MINRES suffers from loss of orthogonalization but the effect is not usually observed in practical applications (Choi, 2006). In our experiments, we observed that OpMINRES does not suffer from the issue of loss of orthogonalization and no extra steps were taken to ensure the orthogonality of the intermediate systems.

# I   Details on Experiments with OpMINRES

In addition to the experiments described in Section 7, we report in this section the details on two more experiments conducted using a real data set and a synthetic data set. We also provide the data set details of speech inversion data set in Section I.3.

In the following experiments two different functional regression problems have been considered. Let $\mathcal{X} = L^2(\Omega_x)$, $\mathcal{Y} = L^2(\Omega_y)$ for suitable $\Omega_x$ and $\Omega_y$ based on the datasets used. We intend to learn a function-valued function $F : \mathcal{X} \rightarrow \mathcal{Y}$. However as noted in Section 1, in practical applications, $x(s) \in \mathcal{X}$ and $y(t) \in \mathcal{Y}$ are not available $\forall s \in \Omega_x$ and $\forall t \in \Omega_y$. Instead only discrete observations $\{x_p\}_{p=1}^P \subset \Omega_x$ and $\{y_q\}_{q=1}^Q \subset \Omega_y$ are observed. However we can approximate these discrete observations as functions using FDA techniques like B-splines or Fourier bases, so that the generalized operator-valued framework introduced in the previous sections can be used. The error metric used for evaluating output functions is residual sum of squares error (RSSE) defined as $RSSE = \int \sum_i \{y_i(t) - \hat{y}_i(t)\}^2 dt$ (Kadri et al., 2016), where $y_i$ is the actual output and $\hat{y}_i$ is the predicted output function. We use total RSSE since it is suitable for the functional nature of the outputs in a functional regression problem. Numerical integration techniques (Hamming, 2012) were used to compute the integrals. For all the experiments, we used OpMINRES with maximum iteration as $10^5$ and tolerance as $10^{-3}$.

## I.1   Additional Experiments on Diffusion Tensor Imaging Data

Multiple sclerosis (MS) is a potentially long-term illness in which the immune system attacks the protective sheath (myelin) that covers nerve fibers affecting the brain and spinal cord (central nervous system) that disrupts the flow of information within the brain, and between the brain and body. Eventually, the disease can cause permanent damage or deterioration of the nerves. As fractional anisotropy (FA) tract profiles for corpus callosum (CCA) and the right corticospinal (RCS) are major indicators of demyelification, we intend to predict the FA profiles along the RCS tract from the FA profiles along the CCA. This would help us having a broader understanding of the relationship between the two for both the healthy as well as MS subjects.

**Dataset Description.**   The Diffusion Tensor Imaging (DTI) dataset available at `https://www.rdocumentation.org/packages/refund/versions/0.1-21/topics/DTI` contains the FA tract profiles along CCA and RCS inferred from DTI scans for 382 profiles from 142 subjects, where 100 subjects are found to suffer from MS and 42 are healthy controls. DTI dataset is available in Refund R package as well. The DTI data were collected at Johns Hopkins University and the Kennedy-Krieger Institute. The dataset also includes subject ID numbers, visit number, total number of scans, multiple sclerosis case status and Paced Auditory Serial Addition Test (pasat) score.

**Data Pre-processing.** As the DTI dataset contains 382 profiles from 142 subjects, we focus on the scans from first visits of all the patients in order to avoid interdependencies. The FA tract values along the CCA and RCS are taken at 93 locations and 54 locations, respectively. There are a lot of missing data with NA values especially in the FA tract values along RCS with a big chunk of the data missing in the initial block of locations. We ignore the missing blocks and refrain from using interpolation or approximations for the missing values for medical record data. Extrapolation and approximation of missing values are not performed in our experiments, considering the significance

of medical attributes and taking into account the possible implications of filling missing data with arbitrary quantities. This pre-processing results in working with 141 pairs of functions. The functions has samples from 93 locations along the CCA tract and 43 along the RCS tract (positions $12 - 54$).

We assume the locations are equally spaced in $[0, 1]$ for both CCA and RCS tract data. Both the functions are normalized to be varying in between $[0, 1]$ by scaling them with their respective maximum absolute quantities.

**Experimental Setting.** All methods were coded in Python 3.6 and all experiments were run on a Linux box with 182 Gigabytes main memory and 28 CPU cores. The experiments performed used 112 samples for training and 29 samples for testing. For hyperparameter tuning, we used 3-fold multi-grid cross validation for all the methods. For encoding of the output functions, we cross-validated the $n_b$ parameter from the set $\{10, 20, 30, 40, 50\}$ for all methods except 3BE with random kitchen sink features.

We consider the following methods for comparison.

**OpMINRES.** We considered the generalized operator-valued kernel in Eq. (3), where we used the following choices for output kernel $h(s, t)$: $e^{-\gamma|t-s|}$ (ABS), $e^{-\gamma(t-s)^2}$ (SQ), $e^{-\gamma_1|t-s|} - e^{-\gamma_2|t-s|}$ (DIFFABS), $e^{-\gamma_1(t-s)^2} - e^{-\gamma_2(t-s)^2}$ (DIFFSQ), $e^{-\gamma_1|t-s|} - e^{-\gamma_2(t-s)^2}$ (DIFFABSSQ) and $e^{-\gamma_1(t-s)^2} - e^{-\gamma_2|t-s|}$ (DIFFSQABS). The following choices for the input kernel $g(x, z)$ were used: $e^{-\eta\|x-z\|^2}$ (RBF), $e^{-\eta_1\|x-z\|^2} - e^{-\eta_2\|x-z\|^2}$ (DIFF-GAUSS) and $\max(0, 1 - \eta\|x - z\|^2)$ (EPAN), where EPAN denotes the Epanechnikov kernel. $\lambda$ was chosen from $\{10^{-3}, 10^{-2}, 0.1, 1, 10, 100\}$. $\gamma, \gamma_1, \gamma_2, \eta, \eta_1, \eta_2$ were chosen from $\{0.001, 0.002, \ldots, 0.009, 0.01, 0.02, \ldots, 0.09, 0.1, 0.2, \ldots, 0.9, 1, 2, \ldots, 10, 20, \ldots, 100\}$.

**3BE.** (Oliva et al., 2015) For this approach, we used two different encodings for the inputs. In the first case, the data set of random kitchen sink features was generated using the input and output bases to be orthogonal trigonometric bases each of size 150, and by setting $\sigma = 0.1$, $D = 3000$ Oliva et al. (2015). Hence the input kernel is computed in this case using the projection coefficients of the inputs onto the bases and then using a transformation $z$ onto a $D$-dimensional space. We denote the input kernel as RKS-DOTPROD in Table 2.

In the second case, the encoding was done only for the output functions using a trigonometric basis of $n_b$ elements and the input functions were considered in their vector form. An RBF kernel $e^{-\eta\|x-z\|^2}$ for inputs was considered and range for $\eta$ was chosen similar to OpMINRES. The regularization parameter $\lambda$ of 3BE was chosen from $\{10^{-3}, 10^{-2}, 0.1, 1, 10, 100\}$.

**KPL.** (Bouche et al., 2020) The dictionary for output functions was an orthonormal basis of $n_b$ trigonometric functions. A separable kernel of the type $K(x_i, x_j) = g(x_i, x_j)B$ was chosen where $B$ is a $n \times n$ diagonal matrix with $B_{ii} = 1/b^{n-i}$. An RBF kernel $e^{-\eta\|x-z\|^2}$ for the inputs was chosen where $\eta$ was chosen similar to OpMINRES. For matrix $B$, the value of $b$ was chosen from $\{0.1, 1, 10, 20, 50, 100\}$. Computing the $\boldsymbol{\eta}^k$ parameter using sample average did not yield good results, hence we chose $\boldsymbol{\eta}^k = \Phi_{(n)}^{\#}\mathbf{y}$ (Bouche et al., 2020). The regularization parameter $\lambda$ of KPL was chosen from $\{10^{-3}, 10^{-2}, 0.1, 1, 10, 100\}$.

**Non-negative Operator-valued kernel approach (NOVK).** (Kadri et al., 2016) Note that the resultant matrix operator equation in (Kadri et al., 2016) is similar to Eq. (6). Hence OpMINRES was used for obtaining the solution. ABS and SQ were used as output kernels. RBF was used as input kernel. All parameters were cross-validated similar to OpMINRES.

The results given in Table 2 show that some indefinite kernel choices used in OpMINRES achieve comparable performance, while others achieve slightly deteriorated performance, indicating that some applications might benefit from particular choices of kernels. Also, 3BE with random kitchen sink features was comparably worse than all other methods. However considering non-encoded inputs in 3BE gave better performance. In terms of runtime, 3BE with non-encoded inputs was faster than all methods. KPL was slower than 3BE with non-encoded inputs and relatively faster than OpMINRES for our approach and for NOVK and 3BE with random kitchen sink features. The time taken for KBE with random kitchen sink features, OpMINRES for NOVK and OpMINRES for our approach were comparable.

| Method | Input Kernel | Output kernel | Best Test RSSE |
|---|---|---|---|
| NOVK | RBF | ABS | 0.1916 |
| NOVK | RBF | SQ | 0.1916 |
| 3BE | RBF | – | 0.1905 |
| 3BE | RKS-DOTPROD | – | 3.1294 |
| KPL | RBF | – | 0.1924 |
| OpMINRES | RBF | DIFFABS | 0.2032 |
| | RBF | DIFFSQ | 0.2035 |
| | RBF | DIFFABSSQ | 0.2034 |
| | RBF | DIFFSQABS | 0.2035 |
| | DIFFGAUSS | ABS | 0.2164 |
| | DIFFGAUSS | SQ | 0.2414 |
| | EPAN | ABS | 0.1903 |
| | EPAN | SQ | 0.1916 |

Table 2: Test RSSE Comparison Results for DTI data

## I.2  Additional Experiments on Toy Problem

We now discuss a few experiments conducted on a synthetic data set.

**Data Generation.** We generate input functions using weighted cosine function on $[-1, 1]$ and the output functions are weighted sixth order Chebychev polynomials of the first kind. In order to generate the toy dataset, we create the input and output functions with $N = 5$, using $c_n \in U([-1, 1]), w_n \in U([0, 1]), \forall n = 1, 2, \ldots, N$ as

$$x(t) = \sum_{n=1}^{N} c_n \cos(w_n t), \ t \in [0, 2\pi], \ y(t) = \sum_{n=1}^{N} c_n T_6(w_n t), \ t \in [-1, 1].$$

The functions $x$ and $y$ have been sampled at 100 points, with Gaussian noise being introduced for both. In order to illustrate the learning capabilities of OpMINRES algorithm, we consider 80 training samples with $\sigma_x = 0.02$ and 20 test samples with $\sigma_y = 0.02$.

**Experimental Setting.** All methods were coded in Python 3.6 and all experiments were run on a Linux box with 182 Gigabytes main memory and 28 CPU cores. The experiments performed used 160 samples for training and 40 samples for testing. For hyperparameter tuning, we used 3-fold multi-grid cross validation for all the methods. For encoding of the output functions, we cross-validated the $n_b$ parameter from the set $\{10, 20, 30, 40, 50\}$ for all methods. The following results are obtained based on different methods used for comparison.

We consider the following methods for comparison.

**OpMINRES.** We considered the generalized operator-valued kernel in Eq. (3), where we used the following choices for output kernel $h(s, t)$: $e^{-\gamma|t-s|}$ (ABS), $e^{-\gamma(t-s)^2}$ (SQ), $e^{-\gamma_1|t-s|} - e^{-\gamma_2|t-s|}$ (DIFFABS), $e^{-\gamma_1(t-s)^2} - e^{-\gamma_2(t-s)^2}$ (DIFFSQ), $e^{-\gamma_1|t-s|} - e^{-\gamma_2(t-s)^2}$ (DIFFABSSQ) and $e^{-\gamma_1(t-s)^2} - e^{-\gamma_2|t-s|}$ (DIFFSQABS). The following choices for the input kernel $g(x, z)$ were used: $e^{-\eta\|x-z\|^2}$ (RBF), $e^{-\eta_1\|x-z\|^2} - e^{-\eta_2\|x-z\|^2}$ (DIFFGAUSS) and $\max(0, 1 - \eta\|x - z\|^2)$ (EPAN). $\lambda$ was chosen from $\{10^{-3}, 10^{-2}, 0.1, 1, 10, 100\}$. $\gamma, \gamma_1, \gamma_2, \eta, \eta_1, \eta_2$ were chosen from $\{0.001, 0.01, 0.1, 1, 10, 100\}$.

**3BE.** (Oliva et al., 2015) Here, the encoding was done only for the output functions using a trigonometric basis of $n_b$ elements and the input functions were considered in their vector form. An RBF kernel $e^{-\eta\|x-z\|^2}$ for inputs was considered and range for $\eta$ was chosen similar to OpMINRES. The regularization parameter $\lambda$ of 3BE was chosen from $\{10^{-3}, 10^{-2}, 0.1, 1, 10, 100\}$.

**KPL.** (Bouche et al., 2020) The dictionary for output functions was an orthonormal basis of $n_b$ trigonometric functions. A separable kernel of the type $K(x_i, x_j) = g(x_i, x_j)B$ was chosen where $B$ is a $n \times n$ diagonal matrix with $B_{ii} = 1/b^{n-i}$. An RBF kernel $e^{-\eta\|x-z\|^2}$ for the inputs was chosen where $\eta$ was chosen similar to OpMINRES. For matrix $B$, the value of $b$ was chosen from

$\{0.1, 1, 10, 20, 50, 100\}$. Computing the $\boldsymbol{\eta}^k$ parameter using sample average did not yield good results, hence we chose $\boldsymbol{\eta}^k = \Phi_{(n)}^{\#} \mathbf{y}$ (Bouche et al., 2020). The regularization parameter $\lambda$ of KPL was chosen from $\{10^{-3}, 10^{-2}, 0.1, 1, 10, 100\}$.

**Non-negative Operator-valued kernel approach (NOVK).** (Kadri et al., 2016) Since the resultant matrix operator equation in (Kadri et al., 2016) is similar to Eq. (6), we used OpMINRES for obtaining the solution. ABS and SQ were used as output kernels. RBF was used as input kernel. All parameters were cross-validated similar to OpMINRES.

The results obtained were almost similar for all the methods (the differences arose only in the seventh digit after the decimal point). During the cross-validation, we could compare the predictions to the noisy outputs. However at the end we could compute the RSSE against the noiseless outputs as well. Accordingly all methods resulted in RSSE of 30.0512 against the noisy outputs and RSSE of 31.6249 against the noiseless outputs. Through these experiments, we see that the results obtained using indefinite kernels are comparable (almost same in this case) to the existing methods using positive definite kernels and algorithms using other techniques.

### I.3 Additional Information on Speech Inversion Dataset

We use the dataset *Haskins IEEE Rate Comparison DB* available at `https://yale.app.box.com/s/cfn8hj2puveo65fq54rp1ml2mk7moj3h/`. The data set contains recordings from 4 female and 4 male subjects reciting 720 phonetically balanced sentences at normal and fast production rates (Tiede et al., 2017). The recordings were done using an electromagnetic articulometry (EMA) system. Each sentence was first produced at speaker's normal speaking rate and then by producing a *fast* repetition of the same, without making errors. Five sensors were placed on the tongue (tip (TT), body (TB), root (TR)), lips (upper (UL) and lower (LL)) and mandible, together with reference sensors on the left and right mastoids, and upper and lower incisors (UI, LI). These EMA trajectories were obtained at 100 Hz and then were low-pass filtered at 5 Hz for references and 20 Hz for articulator sensors. Synchronized audio was recorded at 44100 Hz. The VT variables (namely Lip Aperture (LA), Lip Protrusion (LP), Jaw Angle (JA)) were computed using the EMA trajectories as in (Seneviratne et al., 2019). The experiments were performed for F01 female speaker at normal speaking rate to estimate LA function.

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

[Supplementary Material 2 · neurips_final_afterconf_supplement.pdf]

# A  Primer on Hilbert spaces and RKHS

In this primer, we cover certain formal definitions in order to lay the framework for establishing the properties of reproducing kernel Hilbert spaces (RKHS). A detailed account on RKHS and scalar-valued kernels can be found in (Schölkopf et al., 1999) and (Shawe-Taylor et al., 2004).

## A.1  Hilbert Spaces

Let $\mathcal{H}$ be a vector space defined on the field $\mathbb{R}$ of real numbers (arbitrary fields can be considered). An *inner product* on $\mathcal{H}$ is a function $\langle \cdot, \cdot \rangle_{\mathcal{H}} : \mathcal{H} \times \mathcal{H} \to \mathbb{R}$, such that $\forall f, g, h \in \mathcal{H}$ and scalars $\alpha, \beta \in \mathbb{R}$, it satisfies:

1. $\langle f + g, h \rangle_{\mathcal{H}} = \langle f, h \rangle_{\mathcal{H}} + \langle g, h \rangle_{\mathcal{H}}$
2. $\langle \alpha f, g \rangle_{\mathcal{H}} = \alpha \langle f, g \rangle_{\mathcal{H}}$
3. $\langle f, g \rangle_{\mathcal{H}} = \langle g, f \rangle_{\mathcal{H}}$
4. $\langle f, f \rangle_{\mathcal{H}} \geq 0$ and equality holds if and only if $f = 0$.

**Definition A.1.  Hilbert Space:** A Hilbert space is a vector space $\mathcal{H}$ on $\mathbb{R}$ (arbitrary fields can be considered) with an inner product $\langle \cdot, \cdot \rangle_{\mathcal{H}}$ such that the norm defined by $\|f\|_{\mathcal{H}} = \sqrt{\langle f, f \rangle_{\mathcal{H}}}$ turns $\mathcal{H}$ into a complete metric space. By completeness, we denote that for every Cauchy sequence $\{f_n\}_{n=1,2,\dots} \in \mathcal{H}$, there exists an element $f \in \mathcal{H}$ such that $\lim_{n \to \infty} \|f_n - f\|_{\mathcal{H}} = 0$.

When the context of $\mathcal{H}$ is clear, we use $\|f\|$ instead of $\|f\|_{\mathcal{H}}$.

## A.2  Scalar-valued RKHS

Now, having introduced inner product and Hilbert spaces, we consider learning functions $f : X \to Y$, where $X$ is a suitable input space (typically $X = \mathbb{R}^n$ for some $n \in \mathbb{N}$, the set of natural numbers) and $Y = \mathbb{R}$ is the output space. Let us assume that $L^2(X)$ denotes the space of equivalence classes of square integrable functions from $X$ to $\mathbb{R}$ for some measurable $X$. Now, we can define Evaluation functional which can be used to characterize scalar-valued reproducing kernel Hilbert spaces (RKHS).

**Definition A.2.  Evaluation functional:** Let $X$ be a suitable space of inputs. Consider a Hilbert space $\mathcal{H} \subset \mathbb{R}^X$. The evaluation functional $\Xi_x$ associated with $\mathcal{H}$ valuates a function $f \in \mathcal{H}$ at $x \in X$, and is defined as

$$\Xi_x : \mathcal{H} \to \mathbb{R}, \text{ where } \mathcal{H} \ni f \mapsto \Xi_x f = f(x) \in \mathbb{R}.$$

**Definition A.3.  Scalar-valued RKHS:** A Hilbert space $\mathcal{H}$ is a scalar-valued reproducing kernel Hilbert space if $\mathcal{H} \subset \mathbb{R}^X$ and the associated evaluation functional is bounded: $\forall x \in X, \exists \lambda_x \geq 0$ such that $\forall f \in \mathcal{H}$,

$$|f(x)| = |\Xi_x f| \leq \lambda_x \|f\|_{\mathcal{H}}.$$

It is clear that evaluation functionals are always linear. For $f, g \in \mathcal{H}$ and $\alpha, \beta \in \mathbb{R}$, $\Xi_x(\alpha f + \beta g) = (\alpha f + \beta g)(x) = \alpha f(x) + \beta g(x) = \alpha \Xi_x(f) + \beta \Xi_x(g)$. A natural way to define scalar-valued RKHS can be the continuity of evaluation functional (*e.g.* Definition 4.18 (ii) in (Steinwart and Christmann, 2008)).

Having provided the definitions of scalar-valued RKHS, we can proceed to understanding reproducing kernels.

## A.3  Reproducing Kernels

**Definition A.4.  Reproducing Kernel:** (Berlinet and Thomas-Agnan, 2011, Definition 1.) Let $\mathcal{H}$ be a Hilbert space of scalar-valued functions defined on an input space $X$. A function $k : X \times X \to \mathbb{R}$ is called a reproducing kernel of $\mathcal{H}$ if it satisfies:

1. $\forall x \in X, k(x, \cdot) \in \mathcal{H}$,
2. **(reproducing property)** $\forall x \in X, \forall f \in \mathcal{H}, \langle f, k(x, \cdot) \rangle_{\mathcal{H}} = f(x)$.

In particular, for any $x, y \in X$,

$$k(x, y) = \langle k(x, \cdot), k(y, \cdot) \rangle_{\mathcal{H}}.$$

Now, having defined reproducing kernels with respect to a Hilbert space, we provide a more general definition of kernels.

**Definition A.5. Kernels:** A function $k : X \times X \to \mathbb{R}$ is called a kernel on $X^2$ if there exists a Hilbert space (not necessarily RKHS) $\mathcal{G}$ and a map $\phi : X \to \mathcal{G}$, such that $k(x, y) = \langle \phi(x), \phi(y) \rangle_{\mathcal{G}}$.

In the above definition, the map $\phi$ is called a feature map and $\mathcal{G}$ is a feature space. It is straightforward that every reproducing kernel is a kernel with $\phi : x \mapsto k(x, .), k(x, y) = \langle k(x, .), k(y, .) \rangle_{\mathcal{H}}$. The following property helps us in obtaining a characterization of scalar-valued RKHS and kernels.

**Definition A.6. Positive Definiteness:** A symmetric function $k$ on $X^2$ is positive definite (or non-negative) if for any $f \in L^2(X)$,

$$\int \int f(x)k(x, x')f(x')dxdx' \geq 0.$$

In literature, the words positive, positive semi-definite, positive definite and non-negative have been used equivalently. In order to remove ambiguity, we use positive definiteness with the above mentioned definition. Additionally, kernels satisfying the positive definiteness property are known as Mercer kernels. The above definition generalizes the definition for matrices since for any finite subset of $X$, we obtain that the Gram matrix $(K)_{i,j} = k(x_i, x_j)$ is positive definite. The following lemma provides a relation between reproducing kernels and positive definiteness.

**Lemma A.1.** Let $X$ be an input space and let $\mathcal{H}$ be an RKHS on $X$ with reproducing kernel $k$, then $k$ is positive definite.

The following result provides a characterization for positive definite kernels and RKHS. The result relates a positive definite kernel with a corresponding RKHS (see (Moore, 1935; Aronszajn, 1950)).

**Theorem A.2. (Moore-Aronszajn Theorem)** Let $k : X \times X \to \mathbb{R}$ be positive-definite. There exists a unique RKHS $\mathcal{H} \subset \mathbb{R}^X$ with reproducing kernel $k$. The subspace $\mathcal{H}_0$ of $\mathcal{H}$ spanned by the functions $(k(x, .)_{x \in X})$ is dense in $\mathcal{H}$ and $\mathcal{H}$ is the set of functionals on $X$ which are pointwise limits of Cauchy sequences in $\mathcal{H}_0$ with the inner product

$$\langle f, g \rangle_{\mathcal{H}_0} = \sum_{i=1}^{n} \sum_{j=1}^{n} \alpha_i \beta_j k(x_i, x_j), \text{ where } f = \sum_{i=1}^{n} \alpha_i k(x_i, .) \text{ and } \sum_{j=1}^{n} \beta_j k(x_j, .).$$

The above theorem associates a scalar-valued RKHS with any positive definite kernel. Therefore, there is a bijection between the set of scalar-valued RKHS and the set of positive definite kernels.

# B    Reproducing Kernel Krein Spaces (RKKS)

We start this section by providing a brief introduction to Krein spaces and then provide characterization of scalar-valued reproducing kernel Krein spaces (RKKS) and recall some of their properties. A more thorough introduction of Krein spaces can be found in (Bognar, 1974) and (Azizov and Iokhvidov, 1989) and results related to scalar-valued RKKS can be found in (Alpay, 2001; Ong et al., 2004).

## B.1    Krein Spaces

Let $\mathcal{K}$ be a vector space defined on the field $\mathbb{R}$ of real numbers (we restrict our attention to $\mathbb{R}$ for simplicity, noting that arbitrary fields can be considered). A *bilinear form* on $\mathcal{K}$ is a function $\langle \cdot, \cdot \rangle_{\mathcal{K}} : \mathcal{K} \times \mathcal{K} \to \mathbb{R}$ such that, $\forall f, g, h \in \mathcal{K}$ and scalars $\alpha, \beta \in \mathbb{R}$, it satisfies:

1. $\langle \alpha f + \beta g, h \rangle_{\mathcal{K}} = \alpha \langle f, h \rangle_{\mathcal{K}} + \beta \langle f, h \rangle_{\mathcal{K}}$, and
2. $\langle f, \alpha g + \beta f \rangle_{\mathcal{K}} = \alpha \langle f, g \rangle_{\mathcal{K}} + \beta \langle f, h \rangle_{\mathcal{K}}$.

For $f \in \mathcal{K}$, if $\langle f, g \rangle_{\mathcal{K}} = 0$, $\forall g \in \mathcal{K}$ implies $f = 0$, then the bilinear form is called non-degenerate. The bilinear form $\langle \cdot, \cdot \rangle_{\mathcal{K}}$ is symmetric if, $\langle f, g \rangle_{\mathcal{K}} = \langle g, f \rangle_{\mathcal{K}}$, $\forall f, g \in \mathcal{K}$. The form is called indefinite if there exists $f, g \in \mathcal{K}$ such that $\langle f, f \rangle_{\mathcal{K}} > 0$ and $\langle g, g \rangle_{\mathcal{K}} < 0$. If $\langle f, f \rangle_{\mathcal{K}} \geq 0$, $\forall f \in \mathcal{K}$, then the form is called positive. A non-degenerate, symmetric and positive bilinear form on $\mathcal{K}$ is called *inner product*.

Any two elements $f, g \in \mathcal{K}$ that satisfy $\langle f, g \rangle_{\mathcal{K}} = 0$ are $\langle \cdot, \cdot \rangle_{\mathcal{K}}$-orthogonal. Similarly, any two subspaces $\mathcal{K}_1, \mathcal{K}_2 \subset \mathcal{K}$ that satisfy $\langle f_1, f_2 \rangle_{\mathcal{K}} = 0, \forall f_1 \in \mathcal{K}_1$ and $\forall f_2 \in \mathcal{K}_2$ are called $\langle \cdot, \cdot \rangle_{\mathcal{K}}$-orthogonal. Krein spaces can now be defined based on the notion of a bilinear form.

**Definition B.1. Krein Space.** A vector space $\mathcal{K}$ endowed with a non-degenerate symmetric bilinear form $\langle \cdot, \cdot \rangle_{\mathcal{K}}$ is called a Krein space if it admits a decomposition into a direct sum $\mathcal{K} = \mathcal{H}_1 \oplus \mathcal{H}_2$ of $\langle \cdot, \cdot \rangle_{\mathcal{K}}$-orthogonal Hilbert spaces $\mathcal{H}_1, \mathcal{H}_2$, endowed with inner products $\langle \cdot, \cdot \rangle_{\mathcal{H}_1}, \langle \cdot, \cdot \rangle_{\mathcal{H}_2}$, such that the bilinear form can be written as

$$\langle f, g \rangle_{\mathcal{K}} = \langle f_1, g_1 \rangle_{\mathcal{H}_1} - \langle f_2, g_2 \rangle_{\mathcal{H}_2},$$

where $f_1, g_1 \in \mathcal{H}_1, f_2, g_2 \in \mathcal{H}_2$ and $f = f_1 + f_2, g = g_1 + g_2$.

Notice that, despite the non-negativity of inner products $\langle \cdot, \cdot \rangle_{\mathcal{H}_1}$ and $\langle \cdot, \cdot \rangle_{\mathcal{H}_2}$, the bilinear form $\langle \cdot, \cdot \rangle_{\mathcal{K}}$ might be indefinite. As we describe later, this property of Krein spaces is particularly useful in developing reproducing kernel Krein spaces, which can be suitably identified with the space of indefinite reproducing kernels.

Now we define an associated Hilbert space of the Krein space, where the Hilbertian inner product structure is preserved.

**Definition B.2. Associated Hilbert Space.** Let $\mathcal{K}$ be a Krein space admitting a decomposition into Hilbert spaces $\mathcal{H}_1$ and $\mathcal{H}_2$. Then the associated Hilbert space is defined by $\mathcal{H}_{\mathcal{K}} = \mathcal{H}_1 \oplus \mathcal{H}_2$, endowed with the inner product: $\langle f, g \rangle_{\mathcal{H}_{\mathcal{K}}} = \langle f_1, g_1 \rangle_{\mathcal{H}_1} + \langle f_2, g_2 \rangle_{\mathcal{H}_2}$.

The decomposition of a Krein space $\mathcal{K} = \mathcal{H}_1 \oplus \mathcal{H}_2$ in not necessarily unique. Therefore, a Krein space in general, can be associated with infinitely many Hilbert spaces. But, for any such associated Hilbert space $\mathcal{H}_{\mathcal{K}}$, the topology introduced on $\mathcal{K}$ via the norm $\|f\|_{\mathcal{H}_{\mathcal{K}}} = \sqrt{\langle f, f \rangle_{\mathcal{H}_{\mathcal{K}}}}$ is independent of the decomposition and the associated Hilbert space. The topology on $\mathcal{K}$ defined by the norm of an associated Hilbert space is known as the strong topology on $\mathcal{K}$. The notions of continuity and convergence in a Krein space are defined with respect to the strong topology.

## B.2  Scalar-valued RKKS

Having introduced Krein spaces, we can now adapt them to aid in predictive machine learning applications which aim at learning functions of the form $f : X \to Y$, where $X$ is a suitable input space and $Y = \mathbb{R}$ is the output space. Accordingly, we define a scalar-valued reproducing kernel Krein space (RKKS) and discuss few relevant results.

**Definition B.3. Evaluation functional.** Let $X$ be a suitable space of inputs. Consider a Krein space $\mathcal{K} \subset \mathbb{R}^X$. The evaluation functional $\Xi_x$ that evaluates a function $f \in \mathcal{K}$ at $x \in X$, is defined as

$$\Xi_x : \mathcal{K} \to \mathbb{R}, \text{ where } \mathcal{K} \ni f \mapsto \Xi_x f = f(x) \in \mathbb{R}.$$

**Definition B.4. Scalar-valued RKKS.** (Alpay, 2001) A Krein space $(\mathcal{K}, \langle \cdot, \cdot \rangle_{\mathcal{K}})$ is a scalar-valued reproducing kernel Krein space if $\mathcal{K} \subset \mathbb{R}^X$ and the evaluation functional is continuous on $\mathcal{K}$ with respect to its strong topology.

By restricting the functions $f : X \to \mathbb{R}$ to be such that $f \in \mathcal{K}$, where $\mathcal{K}$ is a scalar-valued RKKS, the next result on the reproducing property of a generalized kernel $\breve{k}$ (which might be indefinite), associated with the scalar-valued RKKS, allows us to learn $f$ using $\breve{k}$.

**Theorem B.1. (Reproducing Kernel)** (Ong et al., 2004) Let $\mathcal{K}$ be a scalar-valued RKKS with decomposition into Hilbert spaces $\mathcal{H}_1$ and $\mathcal{H}_2$. Then

- $\mathcal{H}_1$ and $\mathcal{H}_2$ are scalar-valued RKHS (with kernels $k_1$ and $k_2$).
- **(Reproducing property)** There is a unique symmetric $\breve{k}(x, x')$ with $\breve{k}(x, .) \in \mathcal{K}$, such that for all $f \in \mathcal{K}$, $\langle f, \breve{k}(x, .) \rangle_{\mathcal{K}} = f(x)$.
- $\breve{k} = k_1 - k_2$.

Indeed, analogous to a scalar-valued RKHS where the availability of a reproducing positive kernel is guaranteed (Schölkopf et al., 1999), at least one generalized kernel $\breve{k}$ can be associated with a scalar-valued RKKS as described in the next result.

**Theorem B.2.** (Mary, 2003) Let $\breve{k}$ be a symmetric real valued function on $X^2$, where $X$ is the input space. Then the following are equivalent:

- There exists (at least) one scalar-valued RKKS with kernel $\breve{k}$.

- $\check{k}$ admits a positive decomposition, that is there exists two positive kernels $k_1$ and $k_2$, such that $\check{k} = k_1 - k_2$.
- $\check{k}$ is dominated by some positive kernel $p$ (i.e., $p - \check{k}$ is a positive kernel).

Notice however that unlike the bijection between the set of scalar-valued RKHS and the set of Mercer kernels, there is only a surjection between the set of scalar-valued RKKS and the set of generalized kernels defined in the vector space generated by the set of all Mercer kernels over $X$ (Ong et al., 2004).

## C  Proofs of Lemmas and Theorem in Section 2

We recall the results in Section 2 and discuss the proofs here.

**Lemma C.1.** Let $K_1$ and $K_2$ be two $\mathcal{L}(\mathcal{Y})$-valued non-negative kernels on $\mathcal{X}^2$ with corresponding Hilbert spaces $\mathcal{H}_1$ and $\mathcal{H}_2$ respectively. Then the intersection $\mathcal{H}_1 \cap \mathcal{H}_2$ with the inner product

$$\langle f, f \rangle_{\mathcal{H}_1 \cap \mathcal{H}_2} = \langle f, f \rangle_{\mathcal{H}_1} + \langle f, f \rangle_{\mathcal{H}_2} \tag{9}$$

is a reproducing kernel Hilbert space contractively included in $\mathcal{H}_1$ and $\mathcal{H}_2$.

*Proof.* The intersection $\mathcal{H} = \mathcal{H}_1 \cap \mathcal{H}_2$ endowed with the inner product $\langle \cdot, \cdot \rangle$ in (9) is a pre-Hilbert space. Let $F_n(.)$ be a Cauchy sequence in $\mathcal{H}$. Then it is also a Cauchy sequence in $\mathcal{H}_1$ and $\mathcal{H}_2$, and thus there exists $F(.)$ in $\mathcal{H}_1$ and $G(.)$ in $\mathcal{H}_2$ such that

$$\lim_{n \to \infty} F_n(.) = F(.)$$

in the $\mathcal{H}_1$ norm, and

$$\lim_{n \to \infty} F_n(.) = G(.)$$

in the $\mathcal{H}_2$ norm. For $w \in \mathcal{X}, u \in \mathcal{Y}$,

$$\lim_{n \to \infty} \langle F_n(w), u \rangle_{\mathcal{Y}} = \langle \lim_{n \to \infty} F_n(w), u \rangle_{\mathcal{Y}} = \langle F(w), u \rangle_{\mathcal{Y}}$$
$$= \langle \lim_{n \to \infty} F_n(w), u \rangle_{\mathcal{Y}} = \langle G(w), u \rangle_{\mathcal{Y}}$$
$$\langle F(w), u \rangle_{\mathcal{Y}} = \langle G(w), u \rangle_{\mathcal{Y}}$$
$$\implies F(w) = G(w).$$

Hence, $F(.) = G(.)$ and $F \in \mathcal{H}$. For a Cauchy sequence $F_n(.)$ in $\mathcal{H}$, $\lim_{n \to \infty} F_n(.) = F(.) \in \mathcal{H}$, which proves $\mathcal{H}$ is a Hilbert space. In order to prove that $\mathcal{H}$ is a reproducing kernel Hilbert space, based on (Carmeli et al., 2006, Definition 2.1) and (Carmeli et al., 2010) we use the fact that for $\mathcal{H}_1$ and $\mathcal{H}_2$ which are RKHS, for every $w \in \mathcal{X}$ there exist positive constants $M_w, G_w$ such that

$$\|F(w)\|_{\mathcal{Y}} \leq M_w \|F(.)\|_{\mathcal{H}_1}, \|F(w)\|_{\mathcal{Y}} \leq G_w \|F(.)\|_{\mathcal{H}_2}, \ \forall w \in \mathcal{X}$$
$$\implies \|F(w)\|_{\mathcal{Y}}^2 \leq M_w^2 \|F(.)\|_{\mathcal{H}_1}^2, \|F(w)\|_{\mathcal{Y}}^2 \leq G_w^2 \|F(.)\|_{\mathcal{H}_2}^2, \ \forall w \in \mathcal{X}$$
$$\implies \|F(w)\|_{\mathcal{Y}}^2 \leq P_w^2 (\|F(.)\|_{\mathcal{H}_1}^2 + \|F(.)\|_{\mathcal{H}_2}^2), \ \forall w \in \mathcal{X}, \text{ where } P_w = \min\{M_w, G_w\} \tag{10}$$
$$\implies \|F(w)\|_{\mathcal{Y}} \leq P_w \|F(.)\|_{\mathcal{H}}, \ \forall w \in \mathcal{X}. \tag{11}$$

We obtain inequality in (11) from the inequality in (10) using $\|F(.)\|_{\mathcal{H}_1 \cap \mathcal{H}_2}^2 = \|F(.)\|_{\mathcal{H}_1}^2 + \|F(.)\|_{\mathcal{H}_2}^2$ from (9). Hence, $\mathcal{H}$ is a reproducing kernel Hilbert space. $\qquad \square$

In the above proof, $\mathcal{H}_1 \cap \mathcal{H}_2$ being contractively included in $\mathcal{H}_1$ and $\mathcal{H}_2$ follows based on the definition of $\langle \cdot, \cdot \rangle_{\mathcal{H}_1 \cap \mathcal{H}_2}$ in (9) as $\|F\|_{\mathcal{H}_1} \leq \|F\|_{\mathcal{H}_1 \cap \mathcal{H}_2}$ and $\|F\|_{\mathcal{H}_2} \leq \|F\|_{\mathcal{H}_1 \cap \mathcal{H}_2}, \forall F \in \mathcal{H}_1 \cap \mathcal{H}_2$. Before proceeding to the next lemma, we recall the definition of an inductive set.

**Definition C.1. Inductive Set:** An ordered set $S$ is said to be inductive if every totally ordered subset of $S$ has an upper bound in $S$.

Notice that Lemma C.1 helps us to create a RKHS using intersection of the function-valued RKHS associated with two non-negative operator-valued kernels on $\mathcal{X}^2$. We proceed to obtain a partial order on $I(K_1, K_2)$ defined in Lemma C.2, which is also shown to be inductive.

**Lemma C.2.** Let $K_1$ and $K_2$ be two $\mathcal{L}(\mathcal{Y})$-valued non-negative kernels on $\mathcal{X}^2$ and let $I(K_1, K_2)$ denote the set of all functions $K$ non-negative on $\mathcal{X}^2$ and such that $K \leq K_1$ and $K \leq K_2$. Then $I(K_1, K_2)$ is inductive.

*Proof.* In this proof, we first consider an ordered subset of $I(K_1, K_2)$. Then, we proceed to find the limit of the ordered subset. Finally, we establish that the limit belongs to $I(K_1, K_2)$.

Let $(K_j)_{j \in J}$ be an ordered subset of $I(K_1, K_2)$, where $J$ is a suitable index set. Then, for each $x \in \mathcal{X}$ and $y \in \mathcal{Y}$,

$$\{\langle K_j(x,x)y, y \rangle_{\mathcal{Y}}\}_{j \in J}$$

is an increasing bounded sequence of non-negative numbers.

Let $x \in \mathcal{X}, y \in \mathcal{Y}$ and $i \leq j$ $(i, j \in J)$. Let $\mathcal{H}_i, \mathcal{H}_j$ be the RKHS corresponding to $K_i, K_j$ respectively. Let $K = K_j - K_i$, we see that $K$ is a non-negative $\mathcal{L}(\mathcal{Y})$-valued kernel on $\mathcal{X}^2$ using the fact that $K_i \leq K_j$. Consider $\mathcal{H}$ to be the RKHS corresponding to $K$. Now, using $K_j = K + K_i$ and the definition of Function-valued RKHS in (Kadri et al., 2016), we obtain

$$\mathcal{H}_j = \{K(x,.)y + K_i(x,.)y | x \in \mathcal{X}, y \in \mathcal{Y}\}.$$

In particular, when $K(x,.)y = 0$, then $\mathcal{H}_i \subset \mathcal{H}_j$. Now, based on Lemma C.1, we obtain $\mathcal{H}_i$ is contractively included in $\mathcal{H}_j$ with a new norm $\|.\|_{\mathcal{H}_i}$ based on (9) and the reproducing property of $K_i$ gives us

$$\|K_i(x,.)y\|_{\mathcal{H}_j}^2 \leq \|K_i(x,.)y\|_{\mathcal{H}_i}^2 = \langle K_i(x,.)y, K_i(x,.)y \rangle_{\mathcal{H}_i} = \langle K_i(x,x)y, y \rangle_{\mathcal{Y}}$$

For $x, w \in \mathcal{X}, y, g \in \mathcal{Y}$,

$$\langle K_i(x,w)g - K_j(x,w)g, y \rangle_{\mathcal{Y}} = \langle K_i(x,w)g, y \rangle_{\mathcal{Y}} - \langle K_j(x,w)g, y \rangle_{\mathcal{Y}}$$
$$= \langle K_i(x,.)g, K_j(w,.)y \rangle_{\mathcal{H}_j} - \langle K_j(x,.)g, K_j(w,.)y \rangle_{\mathcal{H}_j} \quad (12)$$
$$= \langle K_i(x,.)g - K_j(x,.)g, K_j(w,.)y \rangle_{\mathcal{H}_j} \quad (13)$$

Equations (12) and (13) follow from the reproducing property of $K_j$ and properties of $\langle \cdot, \cdot \rangle_{\mathcal{Y}}$. Using Cauchy-Schwartz inequality, we obtain from Eq. (13),

$$|\langle K_i(x,w)g - K_j(x,w)g, y \rangle_{\mathcal{Y}}|^2 \leq \|K_j(w,.)y\|_{\mathcal{H}_j}^2 \|K_i(x,.)g - K_j(x,.)g\|_{\mathcal{H}_j}^2.$$

Now using the reproducibility of $K_i$, $\langle K_i(x,.)g, K_j(x,.)g \rangle_{\mathcal{H}_j} = \langle K_j(x,x)g, g \rangle_{\mathcal{Y}}$, and we have

$$\|K_i(x,.)g - K_j(x,.)g\|_{\mathcal{H}_j}^2 = \|K_i(x,.)g\|_{\mathcal{H}_j}^2 + \|K_j(x,.)g\|_{\mathcal{H}_j}^2 - 2\langle K_i(x,.)g, K_j(x,.)g \rangle_{\mathcal{H}_j}$$
$$= \|K_i(x,.)g\|_{\mathcal{H}_j}^2 + \|K_j(x,.)g\|_{\mathcal{H}_j}^2 - 2\langle K_j(x,x)g, g \rangle_{\mathcal{Y}}$$
$$= \langle K_i(x,x)g, g \rangle_{\mathcal{Y}} + \langle K_j(x,x)g, g \rangle_{\mathcal{Y}} - 2\langle K_j(x,x)g, g \rangle_{\mathcal{Y}}$$
$$= \langle K_i(x,x)g, g \rangle_{\mathcal{Y}} - \langle K_j(x,x)g, g \rangle_{\mathcal{Y}}$$
$$= \langle K_i(x,x)g - K_j(x,x)g, g \rangle_{\mathcal{Y}}.$$

Therefore,

$$|\langle K_i(x,w)g - K_j(x,w)g, y \rangle_{\mathcal{Y}}|^2 \leq \langle K_j(w,w)y, y \rangle_{\mathcal{Y}} \langle K_i(x,x)g - K_j(x,x)g, g \rangle_{\mathcal{Y}}. \quad (14)$$

Now, using inequality in (14), when $i \to \infty$ then $\langle K_i(x,x)g - K_j(x,x)g, g \rangle_{\mathcal{Y}} \to 0$ as $\{\langle K_j(x,x)g, g \rangle_{\mathcal{Y}}\}_{j \in J}$ is an increasing bounded sequence of non-negative numbers. Thus $|\langle K_i(x,w)g - K_j(x,w)g, y \rangle_{\mathcal{Y}}| \to 0$, as $i \to \infty$. Therefore, $\exists \bar{K}(\cdot, \cdot) : \mathcal{X} \times \mathcal{X} \to \mathcal{L}(\mathcal{Y})$ such that,

$$\lim_{i \to \infty} \langle K_i(x,w)g, y \rangle_{\mathcal{Y}} = \langle \lim_{i \to \infty} K_i(x,w)g, y \rangle_{\mathcal{Y}} = \langle \bar{K}(x,w)g, y \rangle_{\mathcal{Y}}, \quad \forall x, w \in \mathcal{X}, g, y \in \mathcal{Y}.$$

Hence, $\bar{K}(x,w) = \lim_i K_i(x,w)$ exists for any $x, w \in \mathcal{X}$.

Let $y \in \mathcal{Y}, x \in \mathcal{X}$. Now we show that $\bar{K}$ is non-negative.

$\{\langle K_j(x,x)y, y \rangle_{\mathcal{Y}}\}_{j \in J}$ is an increasing bounded sequence of non-negative numbers.

$$\implies \lim_{j \to \infty} \langle K_j(x,x)g, g \rangle_{\mathcal{Y}} \geq 0$$
$$\implies \langle \lim_{j \to \infty} K_j(x,x)g, g \rangle_{\mathcal{Y}} \geq 0$$
$$\implies \langle \bar{K}(x,x)g, g \rangle_{\mathcal{Y}} \geq 0$$
$$\implies \bar{K} \text{ is non-negative.}$$

Now, since $K_j \in I(K_1, K_2), j \in J$ by our assumption,

$$\langle K_j(x,x)g, g\rangle_{\mathcal{Y}} \leq \langle K_1(x,x)g, g\rangle_{\mathcal{Y}}$$
$$\implies \lim_{j \to \infty} \langle K_j(x,x)g, g\rangle_{\mathcal{Y}} \leq \langle K_1(x,x)g, g\rangle_{\mathcal{Y}}$$
$$\implies \langle \lim_{j \to \infty} K_j(x,x)g, g\rangle_{\mathcal{Y}} \leq \langle K_1(x,x)g, g\rangle_{\mathcal{Y}}$$
$$\implies \langle \bar{K}(x,x)g, g\rangle_{\mathcal{Y}} \leq \langle K_1(x,x)g, g\rangle_{\mathcal{Y}}$$

Similarly,

$$\langle \bar{K}(x,x)g, g\rangle_{\mathcal{Y}} \leq \langle K_2(x,x)g, g\rangle_{\mathcal{Y}}.$$

Therefore, $\bar{K} \leq K_1, K \leq K_2$. Hence, $\bar{K}(x,w) = \lim_i K_i(x,w)$ exists for any $x, w \in \mathcal{X}$. $\bar{K}$ is non-negative and is in $I(K_1, K_2)$. $\qquad\square$

**Corollary C.2.1.** Let $K$ be a difference of two non-negative $\mathcal{L}(\mathcal{Y})$-valued kernels on $\mathcal{X}^2$, $K = K_1 - K_2$. Then, without loss of generality, one can choose $K_1$ and $K_2$ with corresponding Hilbert spaces $\mathcal{H}_1$ and $\mathcal{H}_2$, respectively, such that $\mathcal{H}_1 \cap \mathcal{H}_2 = \{0\}$.

*Proof.* By Zorn's lemma, the set $I(K_1, K_2)$ admits a maximum element $K_{\max}$. Based on the proof in Lemma C.2, we can ensure $K_{\max} \leq K_1, K_{\max} \leq K_2$ i.e., $K_1 - K_{\max}$ and $K_2 - K_{\max}$ are non-negative kernels on $\mathcal{X}^2$. Suppose that $\mathcal{H}_1^{\max}$ and $\mathcal{H}_2^{\max}$ be the corresponding RKHS with respect to $K_1 - K_{\max}$ and $K_2 - K_{\max}$, respectively. Suppose $\mathcal{H}_1^{\max} \cap \mathcal{H}_2^{\max} \neq \{0\}$. By Lemma C.1, the intersection is then an RKHS with a reproducing kernel $K$. Now, let $x \in \mathcal{X}, y \in \mathcal{Y}$, based on the contractive inclusion in Lemma C.1 we obtain

$$\|K(x,.)y\|_{\mathcal{H}_1^{\max}} \leq \|K(x,.)y\|_{\mathcal{H}_1^{\max} \cap \mathcal{H}_2^{\max}}$$
$$\implies \|K(x,.)y\|_{\mathcal{H}_1^{\max}} \leq \|(K_1 - K_{\max})(x,.)y\|_{\mathcal{H}_1^{\max} \cap \mathcal{H}_2^{\max}} \tag{15}$$
$$\implies \langle K(x,.)y, K(x,.)y\rangle_{\mathcal{H}_1^{\max}} \leq \langle (K_1 - K_{\max})(x,.)y, (K_1 - K_{\max})(x,.)y\rangle_{\mathcal{H}_1^{\max} \cap \mathcal{H}_2^{\max}} \tag{16}$$
$$\implies \langle K(x,x)y, y\rangle_{\mathcal{Y}} \leq \langle (K_1 - K_{\max})(x,x)y, y\rangle_{\mathcal{Y}} \tag{17}$$

The inequality in (15) can be obtained from the first inequality, since any function $K(x,.)y$ in $\mathcal{H}_1^{\max} \cap \mathcal{H}_2^{\max}$ is also a member of $\mathcal{H}_1^{\max}$ and hence can be equivalently represented as $(K_1 - K_{\max})(x,.)y$. We obtain inequality in (17) from inequality (16) using reproducing property of $\mathcal{L}(\mathcal{Y})$-valued kernels $K_1 - K_{\max}$ and $K$. From inequality (17), we can deduce that $K \leq K_1 - K_{\max}$. Similarly, we can argue that $K \leq K_2 - K_{\max}$. As $\mathcal{H}_1^{\max} \cap \mathcal{H}_2^{\max} \neq \{0\}$, $K$ is a non-zero reproducing kernel which contradicts the maximality of $K_{\max}$. This leads to $\bar{K}_1 = K_1 - K_{\max}$ and $\bar{K}_2 = K_2 - K_{\max}$ having corresponding Hilbert spaces $\mathcal{H}_1$ and $\mathcal{H}_2$, respectively, satisfying $\mathcal{H}_1 \cap \mathcal{H}_2 = \{0\}$. $\qquad\square$

The following theorem provides a characterization between a generalized $\mathcal{L}(\mathcal{Y})$-valued kernel on $\mathcal{X}^2$ and an associated function-valued RKKS.

**Theorem C.3.** Let $\breve{K}$ be a $\mathcal{L}(\mathcal{Y})$-valued kernel on $\mathcal{X}^2$. Then there is an associated reproducing kernel Krein space if and only if $\breve{K}$ is a generalized $\mathcal{L}(\mathcal{Y})$-valued kernel, that is, $\breve{K} = K_1 - K_2$, where $K_1$ and $K_2$ are non-negative $\mathcal{L}(\mathcal{Y})$-valued kernels on $\mathcal{X}^2$

*Proof.* Suppose that $\breve{K}$ is the reproducing kernel of some RKKS $(\mathcal{K}, [.,.])$ and let $\mathcal{K} = \mathcal{K}_1 \oplus \mathcal{K}_2$ be a decomposition of $\mathcal{K}$, where $(\mathcal{K}_1, \langle.,.\rangle_1)$ and $(\mathcal{K}_2, \langle.,.\rangle_2)$ are orthogonal Hilbert subspaces. Let $P_1$ (respectively, $P_2$) be the orthogonal projection from $\mathcal{K}$ onto $\mathcal{K}_1$ (respectively, $\mathcal{K}_2$). Using reproducibility property in Definition 2.3, we get

$$\langle \breve{K}(z,w)g, y\rangle_{\mathcal{Y}} = [\breve{K}(z,.)g, \breve{K}(w,.)y]$$
$$= \langle P_1 \breve{K}(z,.)g, P_1 \breve{K}(w,.)y\rangle_1 - \langle P_2 \breve{K}(z,.)g, P_2 \breve{K}(w,.)y\rangle_2$$

which exhibits $\breve{K}$ as a difference of two positive functions, that is, $\breve{K}$ is a generalized $\mathcal{L}(\mathcal{Y})$-valued kernel.

Conversely, by definition, a generalized $\mathcal{L}(\mathcal{Y})$-valued kernel $\breve{K}$ is associated with two non-negative operator-valued kernels $K_1, K_2$ such that $\breve{K} = K_1 - K_2$. Using Corollary (C.2.1), we can obtain

Hilbert spaces $\mathcal{H}_1, \mathcal{H}_2$ corresponding to $K_1, K_2$ respectively such that $\mathcal{H}_1 \cap \mathcal{H}_2 = \{0\}$. Then the space

$$\mathcal{K} = \{F = F_1 + F_2, \ F_1 \in \mathcal{H}_1, F_2 \in \mathcal{H}_2\}$$

with the inner product

$$\langle F, F \rangle_{\mathcal{K}} = \langle F_1, F_1 \rangle_{\mathcal{H}_1} + \langle F_2, F_2 \rangle_{\mathcal{H}_2}$$

is a Hilbert space. Moreover, the map $\sigma$ defined by

$$\sigma F = F_1 - F_2$$

is self-adjoint and unitary from $\mathcal{K}$ to $\mathcal{K}$, $K(x,.)y$ belongs to $\mathcal{K}$ for any $x \in \mathcal{X}$ and $y \in \mathcal{Y}$, with

$$[F, F] = \langle F, \sigma F \rangle_{\mathcal{K}},$$

whence we obtain

$$\begin{aligned}
[F, \check{K}(x,.)y] &= \langle F_1, K_1(x,.)y \rangle_{\mathcal{H}_1} + \langle F_2, K_2(x,.)y \rangle_{\mathcal{H}_2} \\
&= \langle F_1(x), y \rangle_{\mathcal{Y}} + \langle F_2(x), y \rangle_{\mathcal{Y}} \\
&= \langle F_1(x) + F_2(x), y \rangle_{\mathcal{Y}} \\
&= \langle F(x), y \rangle_{\mathcal{Y}}.
\end{aligned}$$

Therefore, $(\mathcal{K}, [.,.])$ is a reproducing kernel Krein space with the reproducing kernel $\check{K}$, where $\mathcal{K} = \mathcal{H}_1 \oplus \mathcal{H}_2$. □

Theorem C.3 ensures that for a generalized operator-valued kernel there exists an associated function-valued RKKS, which is a compromise on the bijection that exists between positive definite operator-valued kernels and associated function-valued RKHS (Kadri et al., 2016).

## D   Example (Eq. 3) in Section 2 revisited

Recall the generalized operator-valued kernel in Eq. (3):

$$(\check{K}(x_i, x_j)y)(t) = g(x_i, x_j) \int_{\Omega_y} h(s,t)y(s)ds, \tag{18}$$

where, $\Omega_x = \Omega_y = [0,1], \mathcal{X} = L^2(\Omega_x), \mathcal{Y} = L^2(\Omega_y), g$ is a scalar-valued kernel on $\mathcal{X}^2$ and $h$ is an output kernel on $(\Omega_y)^2$, and either $g$ or $h$ is indefinite. We illustrate here that the indefinite operator-valued kernel constructed in Eq. (18) satisfies the properties in Definition (2.3).

From the definition of $\check{K} = K_1 - K_2$, where $K_1, K_2$ are defined as

$$(K_1(x_i, x_j)y)(t) = g_1(x_i, x_j) \int_{\Omega_y} h_1(s,t)y(s)ds$$

$$(K_2(x_i, x_j)y)(t) = g_2(x_i, x_j) \int_{\Omega_y} h_2(s,t)y(s)ds$$

with $x_i, x_j \in \mathcal{X}, y \in \mathcal{Y}, g_1, g_2$ are scalar-valued positive kernels on $\mathcal{X}^2, h_1, h_2$ are scalar-valued kernels on $(\Omega_y)^2$.

For the Krein space $\mathcal{K}$ of functions from $\mathcal{X}$ to $\mathcal{Y}$, we can obtain $\mathcal{K} = \mathcal{H}_1 \oplus \mathcal{H}_2$, where $\mathcal{H}_1$ and $\mathcal{H}_2$ are function-valued RKHS for operator-valued kernels $K_1$ and $K_2$ respectively. Now we have

$$
\begin{aligned}
\langle F, \check{K}(w,.)y \rangle_{\mathcal{K}} &= \langle F_1, K_1(w,.)y \rangle_{\mathcal{H}_1} - \langle F_2, -K_2(w,.)y \rangle_{\mathcal{H}_2} \\
&= \langle F_1, K_1(w,.)y \rangle_{\mathcal{H}_1} + \langle F_2, K_2(w,.)y \rangle_{\mathcal{H}_2} \\
&= \left\langle F_1, g_1(w,.) \int_{\Omega_y} h_1(s,t)y(s)ds \right\rangle_{\mathcal{F}} + \left\langle F_2, g_2(w,.) \int_{\Omega_y} h_2(s,t)y(s)ds \right\rangle_{\mathcal{F}} \\
&= \int_{\Omega_y} \int_{\Omega_x} [F_1(z)](t) \left[ g_1(w,z) \int_{\Omega_y} h_1(s,t)y(s)ds \right] dzdt + \\
&\qquad \int_{\Omega_y} \int_{\Omega_x} [F_2(z)](t) \left[ g_2(w,z) \int_{\Omega_y} h_2(s,t)y(s)ds \right] dzdt \quad\quad (19) \\
&= \int_{\Omega_y} \int_{\Omega_x} [g_1(w,z)F_1(z)](t) \left[ \int_{\Omega_y} h_1(s,t)y(s)ds \right] dzdt + \\
&\qquad \int_{\Omega_y} \int_{\Omega_x} [g_2(w,z)F_2(z)](t) \left[ \int_{\Omega_y} h_2(s,t)y(s)ds \right] dzdt \quad\quad (20) \\
&= \int_{\Omega_y} [F_1(w)](t)y(t)dt + \int_{\Omega_y} [F_2(w)](t)y(t)dt \quad\quad (21) \\
&= \langle F_1(w), y \rangle_{\mathcal{Y}} + \langle F_2(w), y \rangle_{\mathcal{Y}} \\
&= \langle F(w), y \rangle_{\mathcal{Y}}.
\end{aligned}
$$

Equations (19), (20) and (21) are a result of the reproducibility property of scalar-valued kernels $g_1, g_2, h_1$ and $h_2$.

## E    Proof of Representer Theorem

In this section, we provide a proof for the Representer theorem stated in Section 3. We recall the result here. In the proof we use the Gateaux derivative in an associated function-valued reproducing kernel Krein space for a generalized operator-valued kernel, which is an extension of the Gateaux derivative in a Hilbert space.

**Theorem E.1 (Representer theorem).** Let $\check{K}$ be an indefinite operator-valued kernel and $\mathcal{K}(= \mathcal{K}_1 \oplus \mathcal{K}_2)$ be its corresponding function-valued reproducing kernel Krein space. The solution $\tilde{F}_\lambda \in \mathcal{K}$ of the regularized optimization problem.

$$
\widetilde{F}_\lambda = \arg \operatorname*{stabilize}_{F \in \mathcal{K}} \sum_{i=1}^{n} \|y_i - F(x_i)\|_{\mathcal{Y}}^2 + \lambda \langle F, F \rangle_{\mathcal{K}}, \quad\quad (22)
$$

where $\lambda > 0, F(= F_1 + F_2) \in \mathcal{K}$, has the following form

$$
\widetilde{F}_\lambda(.) = \sum_{i=1}^{n} \check{K}(x_i,.)u_i, \text{ where } u_i \in \mathcal{Y}. \quad\quad (23)
$$

*Proof.* We use the Gateaux derivative to obtain the condition for stationary points which stabilize the functional $J_\lambda(F)$, given by

$$
J_\lambda(F) = \sum_{i=1}^{n} \|y_i - F(x_i)\|_{\mathcal{Y}}^2 + \lambda \langle F, F \rangle_{\mathcal{K}}, \quad \forall F \in \mathcal{K}.
$$

In order to find the critical points in $\mathcal{K}$, we use Gateaux derivative $D_G$ of $J_\lambda$ with respect to $F$ in the direction $H$, which is defined by

$$
D_G J_\lambda(F, H) = \lim_{\tau \to 0} \frac{J_\lambda(F + \tau H) - J_\lambda(F)}{\tau}.
$$

Let $\widetilde{F}$ be the operator in $\mathcal{K}$ such that

$$\widetilde{F} = \arg \operatorname*{stabilize}_{F \in \mathcal{K}} J_\lambda(F) \implies D_G J_\lambda(F, H) = 0, \ \ \forall H \in \mathcal{K}.$$

$J_\lambda$ can be written as

$$J_\lambda(F) = \sum_{i=1}^{n} G_i(F) + \lambda L(F)$$

and as $D_G J_\lambda(F, H) = \langle D_G J_\lambda(F), H \rangle_\mathcal{K}, \ \forall F, H \in \mathcal{K}$, we obtain the following.

1. $L(F) = \langle F, F \rangle_\mathcal{K}$. Therefore we have

$$\lim_{\tau \to 0} \frac{\langle F + \tau H, F + \tau H \rangle_\mathcal{K} - \langle F, F \rangle_\mathcal{K}}{\tau} = 2 \langle F, H \rangle_\mathcal{K}$$

$$\implies D_G L(F) = 2F.$$

2. $G_i(F) = \|y_i - F(x_i)\|_\mathcal{Y}^2$. Then we have

$$\lim_{\tau \to 0} \frac{\|y_i - F(x_i) - \tau H(x_i)\|_\mathcal{Y}^2 - \|y_i - F(x_i)\|_\mathcal{Y}^2}{\tau} = -2\langle y_i - F(x_i), H(x_i) \rangle_\mathcal{Y} \quad (24)$$

$$= -2 \langle \breve{K}(x_i, .)(y_i - F(x_i)), H \rangle_\mathcal{K} \quad (25)$$

$$= -2 \langle \breve{K}(x_i, .)u_i, H \rangle_\mathcal{K}, \quad (26)$$

$$\implies D_G G_i(F) = -2\breve{K}(x_i, .)u_i.$$

We obtain Eq. (25) from Eq. (24) using the reproducibility property in Definition 2.3. In Eq. (25), we use $u_i = y_i - F(x_i)$ to get Eq. (26). Using 1, 2, and $D_G J_\lambda(\widetilde{F}) = 0$, we obtain, $\widetilde{F}(.) = \frac{1}{\lambda} \sum_{i=1}^{n} \breve{K}(x_i, .)u_i$. The constant $\frac{1}{\lambda}$ can be absorbed in functions $u_i$'s, such that $\widetilde{F}(.) = \sum_{i=1}^{n} \breve{K}(x_i, .)u_i$. $\qquad \square$

## F  Condition for Stationary Points of Problem (4)

We obtain a condition for stationary points of the optimization problem in Equation (4).

Using the representer theorem, the problem (4) can be equivalently formulated as the following problem:

$$\tilde{\mathbf{u}}_\lambda = \arg \operatorname*{stabilize}_{\mathbf{u} \in \mathcal{Y}^n} \sum_{i=1}^{n} \left\| y_i - \sum_{j=1}^{n} \breve{K}(x_i, x_j)u_j \right\|_\mathcal{Y}^2 + \lambda \left\langle \sum_{i=1}^{n} \breve{K}(x_i, .)u_i, \sum_{j=1}^{n} \breve{K}(x_j, .)u_j \right\rangle_\mathcal{K}. \quad (27)$$

We have the following simplification of the term $\left\langle \sum_{i=1}^{n} \breve{K}(x_i, .)u_i, \sum_{j=1}^{n} \breve{K}(x_j, .)u_j \right\rangle_\mathcal{K}$ in problem (27). We have

$$\left\langle \sum_{i=1}^{n} \breve{K}(x_i, .)u_i, \sum_{j=1}^{n} \breve{K}(x_j, .)u_j \right\rangle_\mathcal{K} = \sum_{i=1}^{n} \left\langle \breve{K}(x_i, .)u_i, \sum_{j=1}^{n} \breve{K}(x_j, .)u_j \right\rangle_\mathcal{K} \quad (28)$$

$$= \sum_{i=1}^{n} \sum_{j=1}^{n} \left\langle \breve{K}(x_i, .)u_i, \breve{K}(x_j, .)u_j \right\rangle_\mathcal{K} \quad (29)$$

$$= \sum_{i=1}^{n} \sum_{j=1}^{n} \left\langle \breve{K}(x_i, x_j)u_i, u_j \right\rangle_\mathcal{Y}. \quad (30)$$

Note that Eq. (28) and Eq. (29) follow from the property of bilinear forms and Eq. (30) follows from the reproducing property of $\breve{K}$. Thus we have the following simplified formulation:

$$\tilde{\mathbf{u}}_\lambda = \arg \operatorname*{stabilize}_{\mathbf{u} \in \mathcal{Y}^n} \sum_{i=1}^{n} \left\| y_i - \sum_{j=1}^{n} \breve{K}(x_i, x_j)u_j \right\|_\mathcal{Y}^2 + \lambda \sum_{i=1, j=1}^{n} \langle \breve{K}(x_i, x_j)u_i, u_j \rangle_\mathcal{Y},$$

To solve this problem, we use the directional derivative of the objective function $J_\lambda(\mathbf{u})$, given by

$$J_\lambda(\mathbf{u}) = \sum_{i=1}^{n} \left\| y_i - \sum_{j=1}^{n} \breve{K}(x_i, x_j) u_j \right\|_{\mathcal{Y}}^2 + \lambda \sum_{i=1,j=1}^{n} \langle \breve{K}(x_i, x_j) u_i, u_j \rangle_{\mathcal{Y}}, \quad \mathbf{u} \in \mathcal{Y}^n.$$

Letting $J_\lambda(\mathbf{u}) = \sum_{i=1}^{n} G_i(\mathbf{u}) + \lambda L(\mathbf{u})$, we can find the directional derivative of $J_\lambda(\mathbf{u})$ with respect to the direction $\mathbf{v}$ as $D_\mathbf{v} J_\lambda(\mathbf{u})$.

$$D_\mathbf{v} G_i(\mathbf{u}) = \lim_{\tau \to 0} \frac{G_i(u + \tau v) - G_i(u)}{\tau}$$

$$= -2 \left\langle y_i - \sum_{j=1}^{n} \breve{K}(x_i, x_j) u_j, \sum_{j=1}^{n} \breve{K}(x_i, x_j) v_j \right\rangle.$$

$$D_\mathbf{v} L(\mathbf{u}) = \lim_{\tau \to 0} \frac{L(u + \tau v) - L(u)}{\tau}$$

$$= \lambda \sum_{i,j}^{n} \langle \breve{K}(x_i, x_j) u_i, v_j \rangle + \lambda \sum_{i,j}^{n} \langle \breve{K}(x_i, x_j) v_i, u_j \rangle.$$

As $\breve{K}$ is Hermitian from the definition of operator-valued kernel, we obtain

$$\langle \breve{K}(x_i, x_j) u_i, v_j \rangle = \langle u_i, \breve{K}(x_i, x_j) v_j \rangle, \quad \forall i, j = 1, \ldots, n. \tag{31}$$

Therefore,

$$D_\mathbf{v} L(\mathbf{u}) = \lambda \sum_{i,j}^{n} \langle \breve{K}(x_i, x_j) u_i, v_j \rangle + \lambda \sum_{i,j}^{n} \langle \breve{K}(x_i, x_j) v_i, u_j \rangle$$

$$= \lambda \sum_{i,j}^{n} \langle u_i, \breve{K}(x_i, x_j) v_j \rangle + \lambda \sum_{i,j}^{n} \langle \breve{K}(x_i, x_j) v_i, u_j \rangle \tag{32}$$

$$= \lambda \sum_{i,j}^{n} \langle u_i, \breve{K}(x_i, x_j) v_j \rangle + \lambda \sum_{i,j}^{n} \langle u_j, \breve{K}(x_j, x_i) v_i \rangle \tag{33}$$

$$= 2\lambda \sum_{i,j}^{n} \langle u_i, \breve{K}(x_i, x_j) v_j \rangle \tag{34}$$

Eq. (32) follows from Eq. (31) and in Eq. (32), we use symmetry of $\langle \cdot, \cdot \rangle$ to obtain Eq. (34). In order to stabilize $J_\lambda(\mathbf{u})$, its directional derivative $D_\mathbf{v} J_\lambda(\mathbf{u}) = 0$, $\forall v \in \mathcal{Y}^n$.

$$D_\mathbf{v} J_\lambda(\mathbf{u}) = 0$$

$$\implies \sum_{i=1}^{n} D_\mathbf{v} G_i(\mathbf{u}) + \lambda D_\mathbf{v} L(\mathbf{u}) = 0$$

$$\implies -2 \sum_{i=1}^{n} \left\langle y_i - \sum_{j=1}^{n} \breve{K}(x_i, x_j) u_j, \sum_{j=1}^{n} \breve{K}(x_i, x_j) v_j \right\rangle + 2\lambda \sum_{i,j}^{n} \langle u_i, K(x_i, x_j) v_j \rangle = 0$$

$$\implies \sum_{i=1}^{n} \left\langle \sum_{j=1}^{n} \breve{K}(x_i, x_j) u_j - y_i, \sum_{j=1}^{n} \breve{K}(x_i, x_j) v_j \right\rangle + \sum_{i,j}^{n} \langle \lambda u_i, K(x_i, x_j) v_j \rangle = 0$$

$$\implies \sum_{i=1}^{n} \left\langle \sum_{j=1}^{n} \breve{K}(x_i, x_j) u_j - y_i, \sum_{j=1}^{n} \breve{K}(x_i, x_j) v_j \right\rangle + \sum_{i=1}^{n} \left\langle \lambda u_i, \sum_{j=1}^{n} \breve{K}(x_i, x_j) v_j \right\rangle = 0$$

$$\implies \sum_{i=1}^{n} \left\langle \sum_{j=1}^{n} \breve{K}(x_i, x_j) u_j - y_i + \lambda u_i, \sum_{j=1}^{n} \breve{K}(x_i, x_j) v_j \right\rangle = 0, \forall v \in \mathcal{Y}^n.$$

The above condition can be reduced to

$$(\breve{\mathbf{K}} + \lambda I)\mathbf{u} = \mathbf{y}, \tag{35}$$

where $\check{\mathbf{K}}$ is a matrix of operators formed by using $\check{K}$. For the example considered in Appendix D), we have

$$\check{\mathbf{K}} = \begin{bmatrix} K_1(x_1,x_1) - K_2(x_1,x_1) & \dots & K_1(x_1,x_n) - K_2(x_1,x_n) \\ \vdots & \ddots & \vdots \\ K_1(x_n,x_1) - K_2(x_n,x_1) & \dots & K_1(x_n,x_n) - K_2(x_n,x_n) \end{bmatrix}$$

$$= \begin{bmatrix} g_1(x_1,x_1)T_1 - g_2(x_1,x_1)T_2 & \dots & g_1(x_1,x_n)T_1 - g_2(x_1,x_n)T_2 \\ \vdots & \ddots & \vdots \\ g_1(x_n,x_1)T_1 - g_2(x_n,x_1)T_2 & \dots & g_1(x_n,x_n)T_1 - g_2(x_n,x_n)T_2 \end{bmatrix}.$$

Note that in Eq. (35), $\mathbf{y}$ is a column vector of output functions corresponding to the inputs $x_i$'s, for $i = 1, 2, \dots, n$. The $\mathbf{u}$ computed from Eq. (35) consists of a column vector of operators in $\mathcal{L}(\mathcal{Y})$ which act as basis functions for predictions made for an unseen example.

Equation (35) provides a sufficient condition for obtaining the stationary points of the stabilization problem 27.

## G  Krylov Subspace Methods

There are a number of Krylov subspace methods for solving system of a linear system of equations. For solving a linear system

$$Ax = b, A \in \mathbb{R}^{n \times n}, A^\top = A, x, b \in \mathbb{R}^n, \tag{36}$$

a Krylov subspace method is based on iteratively computing an approximation of the solution $x$. Consider the $m$-th Krylov subspace,

$$\mathcal{K}_m(A, r_0) = span\{r_0, Ar_0, \dots, A^m r_0\}, \text{ where } r_0 = b - Ax_0,$$

and $x_0$ is an initial approximation (or guess) of $x$. The solution $x$ of $Ax = b$ is obtained in $\mathcal{K}_m(A, r_0)$ for $m \leq n$, without explicitly computing $A^{-1}$.

A popular variant is the Minimal residual method (MINRES), first proposed in (Paige and Saunders, 1975). MINRES algorithm is based on solving for $x$ in Eq. (36), with a symmetric (Hermitian) matrix (possibly indefinite) $A$ by minimizing the norm residual $\|r_i\| = \|b - Ax_i\|, x_i \in \mathcal{K}_i(A, b)$ in the $i$-th iteration ($\|.\|$ is the 2-norm). MINRES is based on tridiagonalization using orthonormal vectors obtained from Lanczos algorithm (Lanczos, 1950). A detailed account of MINRES and other Krylov subspace methods can be found in (Barrett et al., 1994) and (Choi, 2006).

## H  Details of OpMINRES Algorithm

In order to solve for $\mathbf{u}$ in Eq. (35), we use an operator based Krylov subspace method, inspired by a similar construction in (Ong et al., 2004). As the matrix of operators $(\check{\mathbf{K}} + \lambda I)$ in Eq. (35) is symmetric and possibly indefinite, we based our algorithm on the minimal residual method (MINRES). The proposed OpMINRES is designed for a matrix of operators acting on a column of functions from $\mathcal{L}(\mathcal{Y})$. We illustrate the algorithm by solving for $\mathbf{u}$ in $\mathbf{Au} = \mathbf{y}$, with $\mathbf{A} = (\check{\mathbf{K}} + \lambda I)$.

### H.1  OpLanczos Step

The Lanczos method used in MINRES helps to tridiagonalize $A$ in Eq. (36). Similarly, OpLanczos in OpMINRES is used to trigiagonalize the operator matrix $\mathbf{A}$. The vectors obtained from OpLanczos form an orthonormal set. Using the OpLanczosStep Algorithm 1, we can obtain,

$$\mathbf{A}V_k = V_k T_k, \quad \text{where } T_k = \begin{bmatrix} \alpha_1 & \beta_2 & & & & & 0 \\ \beta_2 & \alpha_2 & \beta_3 & & & & \\ & \beta_3 & \alpha_3 & \ddots & & & \\ & & \ddots & \ddots & \beta_{k-2} & & \\ & & & \beta_{k-1} & \alpha_{k-1} & \beta_k \\ 0 & & & & \beta_k & \alpha_k \end{bmatrix},$$

**Algorithm 1 OpLanczosStep**$(A, v_k, v_{k-1}, \beta_k)$

---

**Input:** $A, v_k, v_{k-1}, \beta_k$
**Output:** $\alpha_k, \beta_{k+1}, v_{k+1}$
$\bar{v}_{k+1} = A v_k - \beta_k v_{k-1}$
$\alpha_k = \langle \bar{v}_{k+1}, q_k \rangle_{\mathcal{Y}^n}$
$\bar{v}_{k+1} \leftarrow \bar{v}_{k+1} - \alpha_k v_k$
$\beta_{k+1} = \|\bar{v}_{k+1}\|_{\mathcal{Y}^n}$
$v_{k+1} = \frac{1}{\beta_{k+1}} \bar{v}_{k+1}$

---

and $V_k = [v_1 \, v_2 \, \ldots \, v_k]$, where $v_i$'s are obtained using OpLanczosStep Algorithm. The columns of $V_k$ belonging to $\mathcal{Y}^n$ are orthonormal and the following equation is satisfied

$$\mathbf{A} V_k = V_{k+1} \overline{T}_k, \quad \text{where } \overline{T}_k = \begin{bmatrix} \alpha_1 & \beta_2 & & & & & 0 \\ \beta_2 & \alpha_2 & \beta_3 & & & & \\ & \beta_3 & \alpha_3 & \ddots & & & \\ & & \ddots & \ddots & \beta_{k-2} & & \\ & & & \beta_{k-1} & \alpha_{k-1} & \beta_k & \\ & & & & \beta_k & \alpha_k \\ 0 & & & & & \beta_{k+1} \end{bmatrix}.$$

We intend to solve $\mathbf{A}\mathbf{u} = \mathbf{y}$ by obtaining a solution in the Krylov space $\mathcal{K}_k(\mathbf{A}, \mathbf{y}) = \text{span}\{\mathbf{y}, \mathbf{A}\mathbf{y}, \mathbf{A}^2\mathbf{y}, \ldots, \mathbf{A}^{k-1}\mathbf{y}\}$. For each iteration $k$, we obtain the following equations using the transformation $\mathbf{x} = V_k x$, where $\mathbf{x} \in \mathcal{Y}^n, x \in \mathbb{R}^k$.

$$\min_{\mathbf{x} \in \mathcal{K}_k(\mathbf{A}, \mathbf{y})} \|\mathbf{y} - \mathbf{A}\mathbf{x}\|_{\mathcal{Y}^n} = \min_{x \in \mathbb{R}^k} \|\mathbf{y} - \mathbf{A} V_k x\|_{\mathcal{Y}^n} = \min_{x \in \mathbb{R}^k} \|\mathbf{y} - V_{k+1} \overline{T}_k x\|_{\mathcal{Y}^n}$$

$$= \min_{x \in \mathbb{R}^k} \|V_{k+1}(\beta_1 e_1 - \overline{T}_k x)\|_{\mathcal{Y}^n}, \tag{37}$$

$$(\text{where } \beta_1 = \|\mathbf{y}\|_{\mathcal{Y}^n}, e_1 = [1 \, 0 \, \ldots \, 0]^\top \text{ and } v_1 = \mathbf{y}/\|\mathbf{y}\|_{\mathcal{Y}^n})$$

$$= \min_{x \in \mathbb{R}^k} \|\beta_1 e_1 - \overline{T}_k x\|_2. \tag{38}$$

The change in norms $\|.\|_{\mathcal{Y}^n}$ in (37) to $\|.\|_2$ is obtained based on the following arguments. Let $z = [z_1, z_2, \ldots, z_{k+1}]^\top \in \mathbb{R}^{k+1}$ and $V_{k+1} = [v_1 \, v_2 \, \ldots v_{k+1}]$, where $v_i \in \mathcal{Y}^n$, for $i = 1, 2, \ldots, k+1$, then we have

$$\|V_{k+1} z_{k+1}\|_{\mathcal{Y}^n} = \|z_1 v_1 + z_2 v_2 + \cdots + z_{k+1} v_{k+1}\|_{\mathcal{Y}^n}$$

$$= \sqrt{z_1^2 \int_{\Omega_y} v_1^2(t) dt + z_2^2 \int_{\Omega_y} v_2^2(t) dt + \cdots + z_{k+1}^2 \int_{\Omega_y} v_{k+1}^2(t) dt} \tag{39}$$

$$= \sqrt{z_1^2 + z_2^2 + \cdots + z_{k+1}^2} \tag{40}$$

$$= \|z\|_2$$

Equation (39) reduces to (40) as the $v_i$'s are orthonormal in $\mathcal{Y}^n$. Solving for $x_k = \arg\min_{x \in \mathbb{R}^k} \|\beta_1 e_1 - \overline{T}_k x\|_2$ can be done using QR decomposition (Choi, 2006) which has been discussed in the next section. Now, the transformation from $\mathbb{R}^k$ back to $\mathcal{Y}^n$ to obtain $\mathbf{u}^k$ is achieved using the following:

$$\mathbf{u}^k = V_k x_k = V_k \left( \arg\min_{x \in \mathbb{R}^k} \|\beta_1 e_1 - \overline{T}_k x\|_2 \right).$$

---

**Algorithm 2 SymOrtho**$(a, b)$

---

    **Input:** $a, b$
    **Output:** $c, s, r$
    **if** $b == 0$ **then**
       $s = 0$
       $r = |a|$
       **if** $a == 0$ **then**
          $c = 1$
       **else**
          $c = \text{sgn}(a)$
       **end if**
    **else if** $a == 0$ **then**
       $c = 0$
       $s = \text{sgn}(b)$
       $r = |b|$
    **else if** $|b| > |a|$ **then**
       $\tau = a/b$
       $s = \text{sgn}(b)/\sqrt{1 + \tau^2}$
       $c = s\tau$
       $r = b/s$
    **else if** $|a| > |b|$ **then**
       $\tau = b/a$
       $c = \text{sgn}(a)/\sqrt{1 + \tau^2}$
       $s = c\tau$
       $r = a/c$
    **end if**

---

### H.1.1 QR Decomposition

In order to apply QR decomposition on symmetric $\overline{T}_k$, we use Givens rotation $Q_k$ to obtain a upper-triangular system.

$$
Q_k \overline{T}_k = \begin{bmatrix} R_k \\ 0 \end{bmatrix} = \begin{bmatrix} \gamma_1^{(1)} & \delta_2^{(1)} & \epsilon_3^{(1)} & & & & 0 \\ & \gamma_2^{(2)} & \delta_3^{(2)} & \epsilon_4^{(1)} & & & \\ & & \ddots & \ddots & \ddots & & \\ & & & \gamma_{k-2}^{(2)} & \delta_{k-1}^{(2)} & \epsilon_k^{(1)} & \\ & & & & \gamma_{k-1}^{(2)} & \delta_k^{(2)} & \\ & & & & & \gamma_k^{(2)} & \\ 0 & & & & & & 0 \end{bmatrix}, \qquad Q_k(\beta_1 e_1) = \begin{bmatrix} t_k \\ \phi_k \end{bmatrix},
$$

where $Q_k = Q_{k,k+1} \ldots Q_{2,3} Q_{1,2}$, $Q_{i,i+1}$ are Givens rotations created to annihilate the $\beta_i$'s in sub-diagonal of $\overline{T}_k$. The $Q_{i,i+1}$'s involved in the product to obtain $Q_k$ are given by,

$$
Q_{i,i+1} = \begin{bmatrix} I_{i-1} & & & \\ & c_i & s_i & \\ & s_i & -c_i & \\ & & & I_{k-i} \end{bmatrix}.
$$

The matrices $Q_{i,i+1}$ are obtained using the SymOrtho Algorithm 2. The sub-problem can be rewritten with $x_k = \arg\min_{x \in \mathbb{R}^k} \|\beta_1 e_1 - \overline{T}_k x\|_2$ as

$$
x_k = \arg\min_{x \in \mathbb{R}^k} \left\| \begin{bmatrix} t_k \\ \phi_k \end{bmatrix} - \begin{bmatrix} R_k \\ 0 \end{bmatrix} x \right\|_2, \text{ where } t_k = [\tau_1 \ \tau_2 \ \ldots \ \tau_k]^\top \text{ and }
$$

**Algorithm 3 OpMINRES**$(A, b, maxiter)$

---

**Input:** $A, b, maxiter$
**Output:** $x, \phi, \psi, \chi$
$\beta_1 = \|b\|_{\mathcal{Y}^n}$
$v_0 = 0$
$v_1 = \frac{1}{\beta_1} b$
$\phi_0 = \tau_0 = \beta_1$
$\chi_0 = 0$
$\delta_1^{(1)} = 0$
$c_0 = -1$
$s_0 = 0$
$d_0 = d_{-1} = x_0 = 0$
$k = 1$
**while** stopping criteria not satisfied **do**
    **OpLanczosStep**$(A, v_k, v_{k-1}, \beta_k) \rightarrow \alpha_k, \beta_{k+1}, v_{k+1}$
    //last left orthogonalization on middle two entries in last column of $T_{k+1,k}$
    $\delta_k^{(2)} = c_{k-1}\delta_k^{(1)} + s_{k-1}\alpha_k$
    $\gamma_k^{(1)} = s_{k-1}\delta_k^{(1)} - c_{k-1}\alpha_k$
    //last left orthogonalization to produce first two entries of $T_{k+2,k+1}e_{k+1}$
    $\epsilon_{k+1}^{(1)} = s_{k-1}\beta_{k+1}$
    $\delta_{k+1}^{(1)} = -c_{k-1}\beta_{k+1}$
    //current left orthogonalization to zero out $\beta_{k+1}$
    **SymOrtho**$(\gamma_k^{(1)}, \beta_{k+1}) \rightarrow c_k, s_k, \gamma_k^{(2)}$
    //right-hand side, residual norms
    $\tau_k = c_k\phi_{k-1}$
    $\phi_k = s_k\phi_{k-1}$
    $\psi_{k-1} = \phi_{k-1}\sqrt{(\gamma_k^{(1)})^2 + (\delta_{k+1}^{(1)})^2}$
    //update solution
    $d_k = \frac{1}{\gamma_k^{(2)}}\left(v_k - \delta_k^{(2)}d_{k-1} - \epsilon_k^{(1)}d_{k-2}\right)$
    $x_k = x_{k-1} + \tau_k d_k$
    $\chi_k = \|x_k\|_{\mathcal{Y}^n}$
    $k \leftarrow k + 1$
**end while**
$x = x_k, \phi = \phi_k, \psi = \phi_k\sqrt{(\gamma_{k+1}^{(1)})^2 + (\delta_{k+2}^{(1)})^2}, \chi = \chi_k$

---

$$
\begin{bmatrix} t_k \\ \phi_k \end{bmatrix} = \beta_1 Q_{k,k+1} \ldots Q_{2,3} \begin{bmatrix} c_1 \\ s_1 \\ 0_{k-1} \end{bmatrix} = \beta_1 Q_{k,k+1} \ldots Q_{3,4} \begin{bmatrix} c_1 \\ s_1 c_2 \\ s_1 s_2 \\ 0_{k-2} \end{bmatrix} = \beta_1 \begin{bmatrix} c_1 \\ s_1 c_2 \\ \vdots \\ s_1 \ldots s_{k-1} c_k \\ s_1 \ldots s_{k-1} s_k \end{bmatrix}.
$$

A shorthand way to represent the action of $Q_{k,k+1}$ can be described as

$$
\begin{bmatrix} c_k & s_k \\ s_k & -c_k \end{bmatrix} \left[ \begin{array}{ccc|c} \gamma_k^{(1)} & \delta_{k+1}^{(1)} & 0 & \phi_{k-1} \\ \beta_{k+1} & \alpha_{k+1} & \beta_{k+2} & 0 \end{array} \right] = \left[ \begin{array}{ccc|c} \gamma_k^{(2)} & \delta_{k+1}^{(2)} & \epsilon_{k+2}^{(1)} & \tau_k \\ 0 & \gamma_{k+1}^{(1)} & \delta_{k+2}^{(1)} & \phi_k \end{array} \right].
$$

OpMINRES computes $\mathbf{u}^k$ in $\mathcal{K}_k(\mathbf{A}, \mathbf{y})$ as an approximate solution to the problem $\mathbf{A}\mathbf{u} = \mathbf{y}$:

$$
\mathbf{u}^k = V_k x_k = V_k R_k^{-1} t_k = D_k \begin{bmatrix} t_{k-1} \\ \tau_k \end{bmatrix} = [D_{k-1} \quad d_k] \begin{bmatrix} t_{k-1} \\ \tau_k \end{bmatrix}
$$
$$
= \mathbf{u}^{k-1} + \tau_k d_k.
$$

The relation satisfied by $d_k$ is given by,

$$d_k = \frac{1}{\gamma_k^{(2)}} \left( v_k - \delta_k^{(2)} d_{k-1} - \epsilon_k^{(1)} d_{k-2} \right).$$

The details are provided in OpMINRES Algorithm 3. As OpMINRES Algorithm 3 is based on reducing the problem in Eq. (35) from an infinite-dimensional optimization problem to a finite-dimensional problem in Eq. (38), the convergence of OpMINRES follows from the convergence of MINRES (Choi, 2006). The construction of OpMINRES ensures the monotonicity of the residual norms. The stopping criteria for OpMINRES could be based on the value of relative residual norms $\phi_k/\phi_0$. Traditionally, MINRES suffers from loss of orthogonalization but the effect is not usually observed in practical applications (Choi, 2006). In our experiments, we observed that OpMINRES does not suffer from the issue of loss of orthogonalization and no extra steps were taken to ensure the orthogonality of the intermediate systems.

# I   Details on Experiments with OpMINRES

In addition to the experiments described in Section 7, we report in this section the details on two more experiments conducted using a real data set and a synthetic data set. We also provide the data set details of speech inversion data set in Section I.3.

In the following experiments two different functional regression problems have been considered. Let $\mathcal{X} = L^2(\Omega_x)$, $\mathcal{Y} = L^2(\Omega_y)$ for suitable $\Omega_x$ and $\Omega_y$ based on the datasets used. We intend to learn a function-valued function $F : \mathcal{X} \to \mathcal{Y}$. However as noted in Section 1, in practical applications, $x(s) \in \mathcal{X}$ and $y(t) \in \mathcal{Y}$ are not available $\forall s \in \Omega_x$ and $\forall t \in \Omega_y$. Instead only discrete observations $\{x_p\}_{p=1}^P \subset \Omega_x$ and $\{y_q\}_{q=1}^Q \subset \Omega_y$ are observed. However we can approximate these discrete observations as functions using FDA techniques like B-splines or Fourier bases, so that the generalized operator-valued framework introduced in the previous sections can be used. The error metric used for evaluating output functions is residual sum of squares error (RSSE) defined as $RSSE = \int \sum_i \{y_i(t) - \hat{y}_i(t)\}^2 dt$ (Kadri et al., 2016), where $y_i$ is the actual output and $\hat{y}_i$ is the predicted output function. We use total RSSE since it is suitable for the functional nature of the outputs in a functional regression problem. Numerical integration techniques (Hamming, 2012) were used to compute the integrals. For all the experiments, we used OpMINRES with maximum iteration as $10^5$ and tolerance as $10^{-3}$.

## I.1   Additional Experiments on Diffusion Tensor Imaging Data

Multiple sclerosis (MS) is a potentially long-term illness in which the immune system attacks the protective sheath (myelin) that covers nerve fibers affecting the brain and spinal cord (central nervous system) that disrupts the flow of information within the brain, and between the brain and body. Eventually, the disease can cause permanent damage or deterioration of the nerves. As fractional anisotropy (FA) tract profiles for corpus callosum (CCA) and the right corticospinal (RCS) are major indicators of demyelification, we intend to predict the FA profiles along the RCS tract from the FA profiles along the CCA. This would help us having a broader understanding of the relationship between the two for both the healthy as well as MS subjects.

**Dataset Description.**   The Diffusion Tensor Imaging (DTI) dataset available at `https://www.rdocumentation.org/packages/refund/versions/0.1-21/topics/DTI` contains the FA tract profiles along CCA and RCS inferred from DTI scans for 382 profiles from 142 subjects, where 100 subjects are found to suffer from MS and 42 are healthy controls. DTI dataset is available in Refund R package as well. The DTI data were collected at Johns Hopkins University and the Kennedy-Krieger Institute. The dataset also includes subject ID numbers, visit number, total number of scans, multiple sclerosis case status and Paced Auditory Serial Addition Test (pasat) score.

**Data Pre-processing.** As the DTI dataset contains 382 profiles from 142 subjects, we focus on the scans from first visits of all the patients in order to avoid interdependencies. The FA tract values along the CCA and RCS are taken at 93 locations and 54 locations, respectively. There are a lot of missing data with NA values especially in the FA tract values along RCS with a big chunk of the data missing in the initial block of locations. We ignore the missing blocks and refrain from using interpolation or approximations for the missing values for medical record data. Extrapolation and approximation of missing values are not performed in our experiments, considering the significance

of medical attributes and taking into account the possible implications of filling missing data with arbitrary quantities. This pre-processing results in working with 141 pairs of functions. The functions has samples from 93 locations along the CCA tract and 43 along the RCS tract (positions $12 - 54$).

We assume the locations are equally spaced in $[0, 1]$ for both CCA and RCS tract data. Both the functions are normalized to be varying in between $[0, 1]$ by scaling them with their respective maximum absolute quantities.

**Experimental Setting.** All methods were coded in Python 3.6 and all experiments were run on a Linux box with 182 Gigabytes main memory and 28 CPU cores. The experiments performed used 112 samples for training and 29 samples for testing. For hyperparameter tuning, we used 3-fold multi-grid cross validation for all the methods. For encoding of the output functions, we cross-validated the $n_b$ parameter from the set $\{10, 20, 30, 40, 50\}$ for all methods except 3BE with random kitchen sink features.

We consider the following methods for comparison.

**OpMINRES.** We considered the generalized operator-valued kernel in Eq. (3), where we used the following choices for output kernel $h(s, t)$: $e^{-\gamma|t-s|}$ (ABS), $e^{-\gamma(t-s)^2}$ (SQ), $e^{-\gamma_1|t-s|} - e^{-\gamma_2|t-s|}$ (DIFFABS), $e^{-\gamma_1(t-s)^2} - e^{-\gamma_2(t-s)^2}$ (DIFFSQ), $e^{-\gamma_1|t-s|} - e^{-\gamma_2(t-s)^2}$ (DIFFABSSQ) and $e^{-\gamma_1(t-s)^2} - e^{-\gamma_2|t-s|}$ (DIFFSQABS). The following choices for the input kernel $g(x, z)$ were used: $e^{-\eta\|x-z\|^2}$ (RBF), $e^{-\eta_1\|x-z\|^2} - e^{-\eta_2\|x-z\|^2}$ (DIFF-GAUSS) and $\max(0, 1 - \eta\|x - z\|^2)$ (EPAN), where EPAN denotes the Epanechnikov kernel. $\lambda$ was chosen from $\{10^{-3}, 10^{-2}, 0.1, 1, 10, 100\}$. $\gamma, \gamma_1, \gamma_2, \eta, \eta_1, \eta_2$ were chosen from $\{0.001, 0.002, \ldots, 0.009, 0.01, 0.02, \ldots, 0.09, 0.1, 0.2, \ldots, 0.9, 1, 2, \ldots, 10, 20, \ldots, 100\}$.

**3BE.** (Oliva et al., 2015) For this approach, we used two different encodings for the inputs. In the first case, the data set of random kitchen sink features was generated using the input and output bases to be orthogonal trigonometric bases each of size 150, and by setting $\sigma = 0.1$, $D = 3000$ Oliva et al. (2015). Hence the input kernel is computed in this case using the projection coefficients of the inputs onto the bases and then using a transformation $z$ onto a $D$-dimensional space. We denote the input kernel as RKS-DOTPROD in Table 2.

In the second case, the encoding was done only for the output functions using a trigonometric basis of $n_b$ elements and the input functions were considered in their vector form. An RBF kernel $e^{-\eta\|x-z\|^2}$ for inputs was considered and range for $\eta$ was chosen similar to OpMINRES. The regularization parameter $\lambda$ of 3BE was chosen from $\{10^{-3}, 10^{-2}, 0.1, 1, 10, 100\}$.

**KPL.** (Bouche et al., 2020) The dictionary for output functions was an orthonormal basis of $n_b$ trigonometric functions. A separable kernel of the type $K(x_i, x_j) = g(x_i, x_j)B$ was chosen where $B$ is a $n \times n$ diagonal matrix with $B_{ii} = 1/b^{n-i}$. An RBF kernel $e^{-\eta\|x-z\|^2}$ for the inputs was chosen where $\eta$ was chosen similar to OpMINRES. For matrix $B$, the value of $b$ was chosen from $\{0.1, 1, 10, 20, 50, 100\}$. Computing the $\boldsymbol{\eta}^k$ parameter using sample average did not yield good results, hence we chose $\boldsymbol{\eta}^k = \Phi_{(n)}^\# \mathbf{y}$ (Bouche et al., 2020). The regularization parameter $\lambda$ of KPL was chosen from $\{10^{-3}, 10^{-2}, 0.1, 1, 10, 100\}$.

**Non-negative Operator-valued kernel approach (NOVK).** (Kadri et al., 2016) Note that the resultant matrix operator equation in (Kadri et al., 2016) is similar to Eq. (6). Hence OpMINRES was used for obtaining the solution. ABS and SQ were used as output kernels. RBF was used as input kernel. All parameters were cross-validated similar to OpMINRES.

The results given in Table 2 show that some indefinite kernel choices used in OpMINRES achieve comparable performance, while others achieve slightly deteriorated performance, indicating that some applications might benefit from particular choices of kernels. Also, 3BE with random kitchen sink features was comparably worse than all other methods. However considering non-encoded inputs in 3BE gave better performance. In terms of runtime, 3BE with non-encoded inputs was faster than all methods. KPL was slower than 3BE with non-encoded inputs and relatively faster than OpMINRES for our approach and for NOVK and 3BE with random kitchen sink features. The time taken for KBE with random kitchen sink features, OpMINRES for NOVK and OpMINRES for our approach were comparable.

| Method | Input Kernel | Output kernel | Best Test RSSE |
|---|---|---|---|
| NOVK | RBF | ABS | 0.1916 |
| NOVK | RBF | SQ | 0.1916 |
| 3BE | RBF | – | 0.1905 |
| 3BE | RKS-DOTPROD | – | 3.1294 |
| KPL | RBF | – | 0.1924 |
| OpMINRES | RBF | DIFFABS | 0.2032 |
| | RBF | DIFFSQ | 0.2035 |
| | RBF | DIFFABSSQ | 0.2034 |
| | RBF | DIFFSQABS | 0.2035 |
| | DIFFGAUSS | ABS | 0.2164 |
| | DIFFGAUSS | SQ | 0.2414 |
| | EPAN | ABS | 0.1903 |
| | EPAN | SQ | 0.1916 |

Table 2: Test RSSE Comparison Results for DTI data

## I.2 Additional Experiments on Toy Problem

We now discuss a few experiments conducted on a synthetic data set.

**Data Generation.** We generate input functions using weighted cosine function on $[-1, 1]$ and the output functions are weighted sixth order Chebychev polynomials of the first kind. In order to generate the toy dataset, we create the input and output functions with $N = 5$, using $c_n \in U([-1, 1]), w_n \in U([0, 1]), \forall n = 1, 2, \ldots, N$ as

$$x(t) = \sum_{n=1}^{N} c_n \cos(w_n t), \ t \in [0, 2\pi], \ y(t) = \sum_{n=1}^{N} c_n T_6(w_n t), \ t \in [-1, 1].$$

The functions $x$ and $y$ have been sampled at 100 points, with Gaussian noise being introduced for both. In order to illustrate the learning capabilities of OpMINRES algorithm, we consider 80 training samples with $\sigma_x = 0.02$ and 20 test samples with $\sigma_y = 0.02$.

**Experimental Setting.** All methods were coded in Python 3.6 and all experiments were run on a Linux box with 182 Gigabytes main memory and 28 CPU cores. The experiments performed used 160 samples for training and 40 samples for testing. For hyperparameter tuning, we used 3-fold multi-grid cross validation for all the methods. For encoding of the output functions, we cross-validated the $n_b$ parameter from the set $\{10, 20, 30, 40, 50\}$ for all methods. The following results are obtained based on different methods used for comparison.

We consider the following methods for comparison.

**OpMINRES.** We considered the generalized operator-valued kernel in Eq. (3), where we used the following choices for output kernel $h(s, t)$: $e^{-\gamma|t-s|}$ (ABS), $e^{-\gamma(t-s)^2}$ (SQ), $e^{-\gamma_1|t-s|} - e^{-\gamma_2|t-s|}$ (DIFFABS), $e^{-\gamma_1(t-s)^2} - e^{-\gamma_2(t-s)^2}$ (DIFFSQ), $e^{-\gamma_1|t-s|} - e^{-\gamma_2(t-s)^2}$ (DIFFABSSQ) and $e^{-\gamma_1(t-s)^2} - e^{-\gamma_2|t-s|}$ (DIFFSQABS). The following choices for the input kernel $g(x, z)$ were used: $e^{-\eta\|x-z\|^2}$ (RBF), $e^{-\eta_1\|x-z\|^2} - e^{-\eta_2\|x-z\|^2}$ (DIFFGAUSS) and $\max(0, 1 - \eta\|x - z\|^2)$ (EPAN). $\lambda$ was chosen from $\{10^{-3}, 10^{-2}, 0.1, 1, 10, 100\}$. $\gamma, \gamma_1, \gamma_2, \eta, \eta_1, \eta_2$ were chosen from $\{0.001, 0.01, 0.1, 1, 10, 100\}$.

**3BE.** (Oliva et al., 2015) Here, the encoding was done only for the output functions using a trigonometric basis of $n_b$ elements and the input functions were considered in their vector form. An RBF kernel $e^{-\eta\|x-z\|^2}$ for inputs was considered and range for $\eta$ was chosen similar to OpMINRES. The regularization parameter $\lambda$ of 3BE was chosen from $\{10^{-3}, 10^{-2}, 0.1, 1, 10, 100\}$.

**KPL.** (Bouche et al., 2020) The dictionary for output functions was an orthonormal basis of $n_b$ trigonometric functions. A separable kernel of the type $K(x_i, x_j) = g(x_i, x_j)B$ was chosen where $B$ is a $n \times n$ diagonal matrix with $B_{ii} = 1/b^{n-i}$. An RBF kernel $e^{-\eta\|x-z\|^2}$ for the inputs was chosen where $\eta$ was chosen similar to OpMINRES. For matrix $B$, the value of $b$ was chosen from

$\{0.1, 1, 10, 20, 50, 100\}$. Computing the $\boldsymbol{\eta}^k$ parameter using sample average did not yield good results, hence we chose $\boldsymbol{\eta}^k = \Phi_{(n)}^{\#}\mathbf{y}$ (Bouche et al., 2020). The regularization parameter $\lambda$ of KPL was chosen from $\{10^{-3}, 10^{-2}, 0.1, 1, 10, 100\}$.

**Non-negative Operator-valued kernel approach (NOVK).** (Kadri et al., 2016) Since the resultant matrix operator equation in (Kadri et al., 2016) is similar to Eq. (6), we used OpMINRES for obtaining the solution. ABS and SQ were used as output kernels. RBF was used as input kernel. All parameters were cross-validated similar to OpMINRES.

The results obtained were almost similar for all the methods (the differences arose only in the seventh digit after the decimal point). During the cross-validation, we could compare the predictions to the noisy outputs. However at the end we could compute the RSSE against the noiseless outputs as well. Accordingly all methods resulted in RSSE of 30.0512 against the noisy outputs and RSSE of 31.6249 against the noiseless outputs. Through these experiments, we see that the results obtained using indefinite kernels are comparable (almost same in this case) to the existing methods using positive definite kernels and algorithms using other techniques.

### I.3 Additional Information on Speech Inversion Dataset

We use the dataset *Haskins IEEE Rate Comparison DB* available at `https://yale.app.box.com/s/cfn8hj2puveo65fq54rp1ml2mk7moj3h/`. The data set contains recordings from 4 female and 4 male subjects reciting 720 phonetically balanced sentences at normal and fast production rates (Tiede et al., 2017). The recordings were done using an electromagnetic articulometry (EMA) system. Each sentence was first produced at speaker's normal speaking rate and then by producing a *fast* repetition of the same, without making errors. Five sensors were placed on the tongue (tip (TT), body (TB), root (TR)), lips (upper (UL) and lower (LL)) and mandible, together with reference sensors on the left and right mastoids, and upper and lower incisors (UI, LI). These EMA trajectories were obtained at 100 Hz and then were low-pass filtered at 5 Hz for references and 20 Hz for articulator sensors. Synchronized audio was recorded at 44100 Hz. The VT variables (namely Lip Aperture (LA), Lip Protrusion (LP), Jaw Angle (JA)) were computed using the EMA trajectories as in (Seneviratne et al., 2019). The experiments were performed for F01 female speaker at normal speaking rate to estimate LA function.