[Reviews · NeurIPS 2020]

Review 1

Summary and Contributions: ====== Post-rebuttal comments Thank you for the comments. I am happy with the response and would recommend including the paragraph (stabilization vs ERM) from the rebuttal into the final version of the paper. It might be interesting as an open problem for future work. ====== Operator valued kernels provide a theoretical framework for modelling learning problems that map functions to functions. A potential shortcoming of this framework is the fact that kernels need to be positive definite. This work addresses that shortcoming and provides a theoretical background for learning with indefinite operator valued kernels. The work mimics presentation and theoretical contributions from [3] by extending that work via previous theory for learning with vector-valued functions [1-2]. The first part of Section 2 reviews (positive) operator valued kernels and provides a concise and clear insight into the specifics of this class of learning algorithms. Definition 2.1 introduces positive definite operator valued kernels and Definition 2.3 gives its indefinite counterpart via reproducing kernel Krein spaces. Theorem 2.4 builds on [4] and provides a characterization of an indefinite operator valued kernel defined as a difference between two positive definite operator valued kernels. Section 3 translates the learning problem from [3] from scalar reproducing kernel Krein spaces to vector valued ones modelled via operator valued kernels. In Section 4, there is a procedure for tackling the formulated learning problem, again quite closely following derivations for the scalar case (see [3]). Section 5 provides a generalization bound via Rademacher complexities and a ball of fixed radius in the strong topology of the operator valued kernel. Experiments have been performed with several kernels on problems where a functional dependence between sets of observations needs to be learned. References: [1] Kadri et al. (2016). Operator-valued Kernels for Learning from Functional Response Data. [2] Micchelli & Pontil (2005). On learning vector-valued functions. [3] Ong et al. (2004). Learning with non-positive kernels. [4] Alpay (1991). Some remarks on reproducing kernel Krein spaces.

Strengths: It is a quite neat extension of scalar valued indefinite kernels to the case of operator valued kernels. Overall the presentation is clean and not difficult to follow. I would say that the paper is well organized and nicely embeds prior work into the presentation while maintaining the focus on its own contributions.

Weaknesses: The experiments seem to be inconclusive as it is unclear whether there is any difference compared to modelling function-to-function learning problems with positive definite kernels. There is also a discrepancy between the sample complexity bound and the learning problem. The former is defined over a ball of constant radius in the strong topology but it is unclear whether this directly translates to the stabilization problem with stabilization term that is not positive definite. A few sentences along these lines might be helpful. The second point that would benefit from further improvement is motivation. Is there any example where indefinite operator valued kernels occur naturally? For instance, indefinite kernel matrices can occur with user defined similarities between items of interest.

Correctness: I have not checked the appendix in details but the results seem to be correct.

Clarity: I find that the paper is clear and rather easy to follow. The structure closely follows prior work and this makes it easy to compare notes when following the theoretical derivations.

Relation to Prior Work: I think that the prior work is adequately covered.

Reproducibility: No

Additional Feedback: Please see the weaknesses.


Review 2

Summary and Contributions: Thanks to the author for the response, I have nothing to add there ============================================================ The paper present the extension of OVK framework to RKKS. Its main contribution is to adapt RKKS tools in a neat manner. A solver is proposed and there is an application to speech inversion, showing that learning in RKKS enables to achieve comparable performances with much more flexibility in term of kernel choice.

Strengths: The theoretical part is sound and presented in a straight forward way (formulation of the combined notions, representer theorem, solver, bound on generalization error). One can easily go through the paper. As far as I know this extension to RKKS was not done before.

Weaknesses: There is no real weakness in the paper. I'm not comfortable with the fact that supplementary material is bigger than the paper itself, but it seems to me that the paper is almost self-sufficient.

Correctness: I did not see any flaws, each part follows some rather established techniques. The experimental part is quite detailed, or at least it provides many information on parameter settings.

Clarity: Yes, the paper can be read quite easily and I did not see typos.

Relation to Prior Work: As the paper is a mix of two worlds, the way it differs from previous work is quite obvious. However it seems that related work from RKKS literature is loosely reported (a low-impact non published paper - Gheondea - in which I did not find the reason why it is mentioned, and only one type of recent algorithm).

Reproducibility: Yes

Additional Feedback: Details on bibliography on arxiv citations: - Gheondea, A. (2013) : not published - Maurer, A. (2016) is published in (LNCS, volume 9925) - Oglic, D. and T. Gärtner (2018b) is published in ICML 2019


Review 3

Summary and Contributions: This paper combines results of RKKS (induced by some indefinite kernels) and functional RKHS, and thus studies the functional space, representer theorem, and generalization properties in functional RKKS. Regularized least squares stabilization problem is given and an iteration algorithm is used to solve the corresponding linear equations.

Strengths: -extend results in functional RKHS to functional RKKS -utilize MINRES to solve a possibly indefinite linear systerm

Weaknesses: I have read all reviews and the author feedback. The rebuttal partially addressed my issues, and thus I increase my score to 6. Nevertheless, the weak motivation and the marginally improvement on experiments would degrade the value of this paper. I suggest the authors improve the motivation for a broader impact in ML community. ============================================================== My feeling is that this paper directly combines results in RKKS and functional RKHS without further improvement. I'm quite doubting about the claim "To the best of our knowledge, most of the technical results in this work are new." For example, 1) in terms of the used techniques, it's unclear to me what's the difference between the presented results and (Alpay 1991) in Lemma 2.2 and 2.3. I think it would be better if you can clarify the relation to prior work. 2) The proof of the representer theorem in functional RKKS (Thm 3.1) appears similar to that of Thm 11 in (Ong 2004). 3) The generalization result in Section 5 is acutally an application of (Maurer, 2016), which is indeed not new. Regarding to Assumption 5.1 that doesn't hold for all non-negative operator valued kernels in functional RKKS, I suggest the following reference could be useful. [A1] De Vito E, Caponnetto A. Risk bounds for regularized least-squares algorithm with operator-value kernels, 2005. Besides, statistical test is needed to evaluate experimental results in Table 1 as the perofmance is indeed not promising enough. Overall, I understand the effort on the study of functional RKKS. But I think it is better to clarify what's the difference, what's the critical gap in technical parts. Minor issues: -I'm familiar with RKKS and understand the importance of indefinite kernels. But I suggest the authors take up some space on the motivation of indefinite kernels for a broader impact in ML community. Maybe, the following references are useful. [A2] Indefinite proximity learning: a review. Neural Computation, 2015. [A3] Learning SVM in Krein spaces. TPAMI, 2016.

Correctness: This paper is technically sound though I just make a high level check on the technical parts.

Clarity: This paper is well written and organized.

Relation to Prior Work: This paper appears not clear to highlight the difference between technical parts in this paper and previous work. I have detailed this in Weaknesses.

Reproducibility: Yes

Additional Feedback:


Review 4

Summary and Contributions: This paper builds the theory of kernel machines with non-positive operator-valued kernels and proposes a suitable algorithm for least squares regression.

Strengths: The work presented in this paper is definitely relevant for the NeurIPS community and very valuable since it builds the theory of non-positive operator-valued kernel machines. In particular, it firmly states the mathematical basis of learning with that kind of kernels, derives a suitable algorithm, gives insights for potential statistical results and provides a numerical evaluation of the proposed method.

Weaknesses: My main objection is that the generalization bound in Equation (8) does not easily translate to a bound on the excess of risk. It results that there is a clear lack of statistical evidence that the proposed estimator does minimize the population version of the risk. Thus, except empirically, what proves that the method is sound from a statistical point of view? That is, that the estimator achieves low error on average? As a more general question: is empirical risk stabilization as sound as empirical risk minimization? Minor comments: - Page 2, Line 86: "operaor" is a typo. - Page 3, Line 97: use \citep and \citet respectively. - Page 3, Line 126: sentence has to be refactored. - Page 4, Line 149: inequality for kernels has to be defined. - Page 4, Line 159: "a an" is a typo. - Page 5, Line 186: symbol \oplus has to be defined. - Page 7, Line 287: "for \forall" is a typo.

Correctness: Everything seems correct.

Clarity: The paper is quite clear (or at least effort has been done in this direction).

Relation to Prior Work: Relation to prior work is adequately addressed.

Reproducibility: Yes

Additional Feedback: I thank the authors for addressing my comments in their rebuttal.

[Author Response · NeurIPS 2020]

**Reviewer 1**: Thank you for your valuable comments. Our replies follow. **(1)** Our experiments show comparable results with other competing methods, while offering an advantage of choice of operator-valued kernels (which might not be non-negative). The results for each dataset are provided for a single test dataset. More replications might be needed to infer statistical significance of our results, which we can add. **(2)** Our stabilization problem (4) in Section 3, inspired from (Ong et al., 2004) helps in deriving the result in Representer Theorem 3.1. On the other hand, when the stabilizer $\widetilde{F}_\lambda$ from Eq. (4) belongs to the ball $\mathcal{B}_\mathcal{K}$ of fixed radius $r$ (defined in Section 5 with $r = 1$), it enjoys the generalization bounds in Eq. (8). At least to us it is not very clear how the stabilizer behaves when it does not belong to $\mathcal{B}_\mathcal{K}$. One might suppose that the formulation similar to (Oglic and Gärtner, 2018) can be used here. However, adapting the minimization problem formulation in (Oglic and Gärtner, 2018) would lead to integral variance constraints in our case. Further, using a Gateaux derivative approach for the constrained or unconstrained minimization problem similar to that in (Oglic and Gärtner, 2018), leads to difficulties in obtaining the Representer Theorem 3.1 in our paper. As a consequence of these facts, we can only resort to an empirical cross-validation approach which we have used in our experiments to ensure that the stabilizer of problem (4) is not far away from $\mathcal{B}_\mathcal{K}$. **(3)** In contrast to scalar-valued indefinite kernels which arise naturally in many scenarios, we are not aware of natural occurrences of operator valued kernels (both positive and indefinite) in existing literature. Hence we motivate the use of generalized operator valued kernels from a function estimation and learning methodology viewpoint, which allows us to relax the requirement of positive definite kernels in learning function-valued functions. However, we will strive to improve the motivation.

**Reviewer 2**: Thank you for your encouraging comments. We will refine the references and include other related works.

**Reviewer 3**: Thank you for your inquisitive comments. Our replies follow. **(1)** Lemma 2.2 and 2.3 presented in our paper are for function-valued RKHS and $\mathcal{L}(\mathcal{Y})$-valued kernels, whereas the similar lemmas in (Alpay, 1991) are for $\mathbb{C}^{n \times n}$-valued kernels. Though our results are extensions of similar results in (Alpay, 1991), we point to the important differences here. In the proof of Lemma 2.2, we require the results in (Carmeli et al., 2006) and (Carmeli et al., 2010) to prove that $\mathcal{H}$ is a function-valued RKHS, which are not required in Alpay's proof. In deriving Corollary 2.3.1 using Lemma 2.2 and Lemma 2.3 we needed to establish arguments for operator valued kernels which were not obvious based on the arguments in (Alpay, 1991). **(2)** The derivation of representer theorem in our case requires using the definition of generalized operator-valued kernel (in Section 2) to obtain Equations (24), (25) and (26) in Appendix E which yield the required representer theorem in our setting. The derivation in our case uses Gateaux derivative with variational function approach to obtain necessary condition for stationary points for the stabilization problem (4), whereas the result in (Ong et al., 2004) uses subdifferential with respect to a vector $[f(x_1), \ldots, f(x_m)]^\top$ to obtain the representer theorem. **(3)** However, the bound on Rademacher average in Section 5 is a natural extension of the result in (Maurer, 2016). **(4)** We have cited published version of ref. [A1] in Section 5; on careful reading of ref. [A1], we found that trace class condition (Assumption 5.1) is used in [A1] as well. **(5)** The results for each dataset are provided for a single test dataset. More replications might be needed to infer statistical significance of our results, which we can add. **(6)** Thank you for the additional references. In contrast to scalar-valued indefinite kernels which arise naturally in many scenarios, we are not aware of natural occurrences of operator valued kernels (positive and indefinite) in existing literature. Hence we motivate the use of generalized operator valued kernels from a function estimation and learning methodology viewpoint, which allows us to relax the requirement of positive definite kernels in learning function-valued functions. However, we will strive to improve the motivation.

**Reviewer 4**: Thank you for your enlightening comments. Our replies follow. **(1)** Our stabilization problem (4) in Section 3, inspired from (Ong et al., 2004) helps in deriving the result in Representer Theorem 3.1. On the other hand, when the stabilizer $\widetilde{F}_\lambda$ from Eq. (4) belongs to the ball $\mathcal{B}_\mathcal{K}$ of fixed radius $r$ (defined in Section 5 with $r = 1$), it enjoys the generalization bounds in Eq. (8). At least to us it is not very clear how the stabilizer behaves when it does not belong to $\mathcal{B}_\mathcal{K}$. One might suppose that the formulation similar to (Oglic and Gärtner, 2018) can be used here. However, adapting the minimization problem formulation in (Oglic and Gärtner, 2018) would lead to integral variance constraints in our case. Further, using a Gateaux derivative approach for the constrained or unconstrained minimization problem similar to that in (Oglic and Gärtner, 2018), leads to difficulties in obtaining the Representer Theorem 3.1 in our paper. As a consequence of these facts, we can only resort to an empirical cross-validation approach which we have used in our experiments to ensure that the stabilizer of problem (4) is not far away from $\mathcal{B}_\mathcal{K}$. **(2)** We will correct the typos.

# References

Alpay, D. (1991). Some remarks on reproducing kernel krein spaces. *Journal of Mathematics 21*(4).

Carmeli, C., E. De Vito, and A. Toigo (2006). Vector valued reproducing kernel hilbert spaces of integrable functions and mercer theorem. *Analysis and Applications 4*(4), 377–408.

Carmeli, C., E. De Vito, A. Toigo, and V. Umanitá (2010). Vector valued reproducing kernel hilbert spaces and universality. *Analysis and Applications 8*(1), 19–61.

Maurer, A. (2016). A vector-contraction inequality for rademacher complexities. In *ALT*.

Oglic, D. and T. Gärtner (2018). Learning in reproducing kernel kreın spaces. In *ICML*.

Ong, C. S., X. Mary, S. Canu, and A. J. Smola (2004). Learning with non-positive kernels. In *ICML*.


[Meta-Review · NeurIPS 2020]

The paper considers the problem of operator valued indefinite kernels. It presents theoretical analysis, optimization method, and empirical results. Four knowledgeable reviewers carefully considered the paper, and all of them recommended accept. Therefore I also recommend to accept the paper at NeurIPS 2020. The authors are encouraged to address all the issues raised by the reviewers for the final version. For example the comparison of stabilization and minimization by Reviewer #1 and #4, and overall to improve the motivation of using operator valued indefinite kernels.